# RoFt-Mol: Benchmarking Robust Fine-Tuning with Molecular Graph Foundation Models

**Shikun Liu**[*1]   **Deyu Zou**[*2]   **Nima Shoghi**[1]   **Victor Fung**[1]   **Kai Liu**[3]   **Pan Li**[1]

[1] Georgia Institute of Technology [2] The Chinese University of Hong Kong
[3] SES AI

## Abstract

In the era of foundation models, fine-tuning pre-trained models for specific downstream tasks has become crucial. This drives the need for robust fine-tuning methods to address challenges such as model overfitting and sparse labeling. Molecular graph foundation models (MGFMs) face unique difficulties that complicate fine-tuning. These models are limited by smaller pre-training datasets and more severe data scarcity for downstream tasks, both of which require enhanced model generalization. Moreover, MGFMs must accommodate diverse objectives, including both regression and classification tasks. To better understand and improve fine-tuning techniques under these conditions, we classify eight fine-tuning methods into three mechanisms: weight-based, representation-based, and partial fine-tuning. We benchmark these methods on downstream regression and classification tasks across supervised and self-supervised pre-trained models in diverse labeling settings. This extensive evaluation provides valuable insights and informs the design of a refined robust fine-tuning method, ROFT-MOL. This approach combines the strengths of simple post-hoc weight interpolation with more complex weight ensemble fine-tuning methods, delivering improved performance across both task types while maintaining the ease of use inherent in post-hoc weight interpolation. [2]

## 1 Introduction

In recent years, foundation models [1, 2] have achieved success in learning high-quality, general-purpose representations of images and text through pre-training on diverse datasets [3, 4, 5, 6, 7, 8]. To adapt these pre-trained models for downstream applications, additional training on task-specific data, known as fine-tuning, is often required. However, vanilla fine-tuning frequently encounters challenges, including model overfitting [9, 10, 11], catastrophic forgetting of pre-trained knowledge [12, 13, 14, 15], and distribution shifts between fine-tuned and test samples, which can lead to negative transfer [16, 17]. These challenges highlight the need for robust fine-tuning strategies [18, 19, 20, 21, 22, 23].

Recently, the advantages of foundation models have been extended to various scientific applications [24, 25, 26]. Among these, molecular graph foundation models (MGFMs) have gained significant attention for their promising potential in biochemistry [27, 28, 29, 30, 31, 32, 33, 34, 35, 36]. While MGFMs exhibit scaling behaviors similar to foundation models in other domains [37], they face unique challenges related to data and tasks.

A primary challenge stems from the significantly smaller pre-training datasets in this domain, typically consisting of at most $O(100M)$ molecular samples, compared to the billions of samples used in other domains [38]. This limitation restricts the parameter scale of MGFMs ($O(100M)$ parameters) and their generalization capacity [39, 40]. Furthermore, downstream tasks in this domain often involve limited data for fine-tuning, with datasets containing only tens or a few hundred labeled samples [41],

---

[*]Equal contribution.

[2]Our code and data are available at: `https://github.com/Graph-COM/RoFt-Mol`

exacerbating the difficulty of achieving robust model generalization. In addition to data constraints, many downstream tasks, such as molecular property prediction, are regression-based [42, 43]. These tasks require models to capture fine-grained numerical patterns, which presents a distinct requirement compared to the coarse-grained feature reliance typical in classification tasks in CV and NLP. These factors collectively highlight the need for a careful examination of fine-tuning strategies for MGFMs and their appropriate improvement.

To answer this question, we introduce RoFt-Mol, a benchmark that evaluates existing fine-tuning methods across diverse molecular property prediction tasks. To explore factors influencing the fine-tuning (FT) performance of MGFMs, we categorize 8 FT methods into 3 distinct mechanisms: 1) *weight-based FT*, which ensembles the weights from both pre-trained and fine-tuned models, 2) *representation-based FT*, which regularizes the proximity between pre-trained and fine-tuned latent data representations, and 3) *partial FT*, which optimizes only a subset of the pre-trained model's weights while keeping the rest frozen. To derive generalizable insights into how different fine-tuning mechanisms interact with pre-training strategies and downstream task types, we evaluate six diverse pre-trained models, spanning self-supervised and supervised learning, with pure graph-based, graph transformer based and multi-modal models in varying scales, then evaluate on a broad set of molecular property prediction tasks, including 8 classification and 4 regression tasks. To simulate the challenges encountered during the fine-tuning stages of MGFMs, we further consider the few-shot and out-of-distribution settings. Drawing from the broad range of pre-trained models and downstream tasks, we indeed find that the choice of best fine-tuning mechanism is highly determined by the *pre-training objective* and the *downstream task type*. We summarize high-level insights as follows, with further detailed results presented in Sec. 4. The bold text within brackets indicates the corresponding support in the experiment sections for clear cross-referencing:

- **Impact from Supervised vs. Self-supervised pre-trained models:** Supervised pre-training learns domain-specific information with task supervision, while self-supervised pre-training captures general-purpose knowledge through training on generic synthetic tasks. We observe that, in few shot fine-tuning, supervised pre-training generally yields better fine-tuning performance than self-supervised pre-training even when the pre-training tasks do not align well with the fine-tuning tasks. In contrast, for non-few-shot settings, supervised pre-training performs better only when the supervised pre-training tasks closely align with the downstream tasks [**Q2**].
- **Impact from Classification vs. Regression tasks:** Regression tasks need more precise numerical labels and finer molecule modeling. Therefore, MGFMs face less risk of overfitting in regression tasks compared to classification tasks, particularly in the few-shot setting [**Q1**].
- **Correspondence with different fine-tuning methods:** For self-supervised pre-trained models, *weight-based fine-tuning* often results in better performance by effectively integrating general knowledge from pre-training with task-specific knowledge from fine-tuning [**Finding 1**]. On the other hand, *partial fine-tuning* typically leads to underfitted molecular representations in few-shot fine-tuning, particularly for regression tasks [**Finding 2**]. For supervised pre-trained models, *representation-based fine-tuning* performs well due to the preservation of domain-relevant pre-trained representations [**Finding 3**].

Based on the findings, we argue that the first step in selecting or designing an effective fine-tuning strategy is to consider the pre-training strategies. Then after finding the suitable fine-tuning mechanisms, we need to take the type of downstream tasks into account. For instance, weight-based fine-tuning methods generally work the best under self-supervised pre-trained model, while simple post-hoc weight interpolation between pre-trained and fine-tuned model weights (WiSE-FT) performs well for classification tasks but struggles with regression tasks. In contrast, a more complex weight ensemble approach ($L^2$-SP) achieves better performance in regression tasks, though it comes with the cost of increased tuning complexity. Therefore, inspired by the rule, we propose a **new method, DWiSE-FT** that achieves strong performance for both regression and classification tasks as a weight-based solution for self-supervised pre-trained model. DWiSE-FT combines the strengths of WiSE-FT and $L^2$-SP, providing strong performance for both task types while maintaining the plug-and-play ease of post-hoc interpolation. The success of DWiSE-FT showcases that our benchmark identifies valuable insights in improving fine-tuning strategies given distinct MGFMs.

## 2 Finetuning Methods for Evaluation

In this section, we briefly introduce representative methodologies used in pre-training and fine-tuning for MGFMs.

**Self-supervised Pre-training** strategies have been proven to be effective in generating transferable molecular representations for downstream tasks [44]. In a high level, they can be divided into *reconstruction* methods and *contrastive* methods. The generative-based strategies adopt mask-based graph reconstruction by utilizing graph autoencoders [28, 45, 46, 47], context predictions [27, 35] and generative language model pre-training [48, 49]. On the other hand, contrastive-based methods aim for maximizing the similarity between perturbed instance pairs [50, 30, 51, 52, 53, 54, 55, 56, 57, 58]. Moreover, the advancement of language models has prompted numerous studies to employ multi-modal frameworks. These approaches harness language models to enhance molecular understanding through techniques such as cross-modal contrastive learning and alignment [59, 60, 61, 62].

In this work, we select *GraphMAE* [28] as the representative of the recontruction-based pre-trained model, which focuses on masked feature reconstruction with scaled cosine error that enabled robust training. Regarding the contrastive pre-trained model, we choose *Mole-BERT* [52] that combines the node-level masked atom modeling to predict the masked atom tokens and the graph-level contrastive learning through triplet loss and contrastive loss. Lastly, we choose *MoleculeSTM* [60] as the representative of multi-modal molecule structure-text model that jointly learning molecules' chemical structures and textual descriptions via a contrastive learning strategy.

**Supervised Pre-training.** Recently, to leverage more diverse datasets and tasks, researchers started exploring the ability of supervised pre-training with multi-task learning for molecular representations [63, 31, 32]. We adopt pre-trained models trained on multi-task labeled samples in a supervised manner from the *Graphium* library [32]. In addition to the GNN-based backbone, more expressive architectures like Graph Transformer [64, 65, 66] have been proposed and can be used as the pre-trained backbone with supervised labels, which we adopt *GraphGPS* [65] as a representative.

**Fine-tuning**'s overall goal is to adapt the pre-trained model to downstream applications. Specifically, given a pre-trained GNN encoder $f_{\boldsymbol{\theta}}$ with parameters $\boldsymbol{\theta}$ initialized from the pretrained parameters $\boldsymbol{\theta}_{\mathrm{pre}}$, fine-tuning optimizes the encoder $f_{\boldsymbol{\theta}}$ and an additional prediction head $g_{\boldsymbol{\phi}}$ with parameters $\boldsymbol{\phi}$ over downstream molecules $\{(\mathcal{G}_i, y_i)\}_{i=1}^{N}$. The vanilla version, **full-FT**, optimizes the entire model weights following:

$$\min_{\{\boldsymbol{\theta}, \boldsymbol{\phi}\}} \sum_{i=1}^{N} \mathcal{L}(g_{\boldsymbol{\phi}} \circ f_{\boldsymbol{\theta}}(\mathcal{G}_i), y_i), \tag{1}$$

where $\boldsymbol{\theta}$ is initialized as $\boldsymbol{\theta}_{\mathrm{pre}}$ and $\mathcal{L}$ denotes the loss function for prediction tasks. As discussed, there are advanced fine-tuning strategies proposed on top of the full-FT framework. As shown in Fig. 1, we group them into three categories based on their mechanisms and benchmark representative methods for each category. More FT methods that fall into each category or others will be discussed in Appendix C.

• **Partial FT** strategies only optimizes partial weights of the pre-trained model, *i.e.*, a subset of weights within $\{\boldsymbol{\theta}, \boldsymbol{\phi}\}$ will be updated following the same objective as Eq. 1. *Linear Probing (LP)* only trains the additional prediction head $g$ during the FT. *Surgical FT* [12] updates only partial layers within the encoder. For instance, we can update the weights for $k$-th layer of the GNN encoder as $\min_{\{[\boldsymbol{\theta}]_k, \boldsymbol{\phi}\}} \sum_{i=1}^{N} \mathcal{L}(g_{\boldsymbol{\phi}} \circ f_{\boldsymbol{\theta}}(\mathcal{G}_i), y_i)$, where $k$ is the hyperparameter that can be tuned. *LP-FT* [20] aims to address the issue of pre-trained feature distortion during the full-FT process. It first performs the LP step to the prediction head $g_{\boldsymbol{\phi}}$ while keeping the encoder $f_{\boldsymbol{\theta}}$ with fixed pre-trained parameters $\boldsymbol{\theta}_{\mathrm{pre}}$, followed by applying full-FT with the tuned prediction head.

• **Weight-based FT** strategies mainly update the entire model weights through combining pre-trained model weights and fine-tuned model weights. *WiSE-FT* [19] linearly interpolates between pre-training parameters $\boldsymbol{\theta}_{\mathrm{pre}}$ and fine-tuning parameters $\boldsymbol{\theta}_{\mathrm{ft}}$ using a mixing coefficient $\alpha$, to get the interpolated GNN $f_{\boldsymbol{\theta}_{\mathrm{int}}}$ with weights $\boldsymbol{\theta}_{\mathrm{int}} = (1 - \alpha) \cdot \boldsymbol{\theta}_{\mathrm{pre}} + \alpha \cdot \boldsymbol{\theta}_{\mathrm{ft}}$. We first perform full-FT to obtain the adapted encoder $f_{\boldsymbol{\theta}_{\mathrm{ft}}}$ and classifier $g_{\boldsymbol{\phi}}$, then apply post-hoc weight ensembling to get $f_{\boldsymbol{\theta}_{\mathrm{int}}}$, with final predictions given by $g_{\boldsymbol{\phi}} \circ f_{\boldsymbol{\theta}_{\mathrm{int}}}(\mathcal{G}_i)$. $\alpha$, as a hyperparameter, controls the weight ensemble. $L^2$-*SP* [14] regularizes the fine-tuning model weights $\boldsymbol{\theta}$ closer to the pre-trained weights $\boldsymbol{\theta}_{\mathrm{pre}}$ by $\Omega(\boldsymbol{\theta}, \boldsymbol{\phi}) = \frac{\delta}{2} \|\boldsymbol{\theta} - \boldsymbol{\theta}_{\mathrm{pre}}\|_2^2$. We optimize for $\boldsymbol{\theta}$ and $\boldsymbol{\phi}$ by combining the prediction loss from Eq. 1 and $\Omega(\boldsymbol{\theta}, \boldsymbol{\phi})$ with tunable trade-off coefficient $\delta$.

• **Representation-based FT** methods mainly regulate the latent representation space during FT. *Feature-map* [13] adds distance regularization between the latent representations of pre-trained and fine-tuned models to the Full-FT loss. The regularization is defined as $\Omega(\boldsymbol{\theta}) = \delta \sum_{i=1}^{N} \frac{1}{2} \|f_{\boldsymbol{\theta}}(\mathcal{G}_i) - f_{\boldsymbol{\theta}_{\mathrm{pre}}}(\mathcal{G}_i)\|_2^2$, where $\delta$ controls the regularization strength. *BSS* [17] aims at

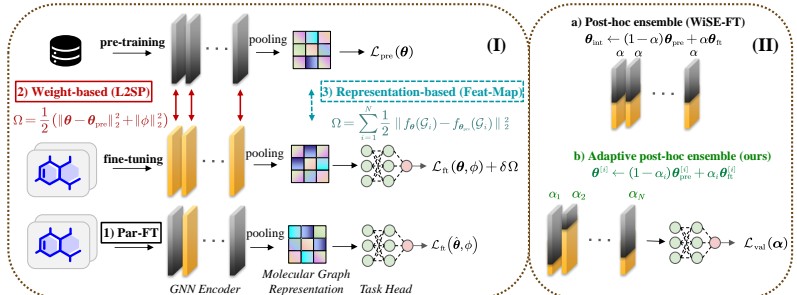

Figure 1: The overall framework of fine-tuning strategies evaluated in our benchmark, ROFT-MOL, and the proposed novel method, DWiSE-FT. **(I)** The GNN encoder is pre-trained on a large database by the objective $\mathcal{L}_{\text{pre}}$, and fine-tuned on the downstream dataset by $\mathcal{L}_{\text{ft}}$ (*c.f.*, Eq. 1). 1) Partial-FT, 2) Weight-based FT, and 3) Representation-based FT achieve robust fine-tuning by freezing partial pre-trained model weights, regularizing model weights and latent representations, respectively. **(II)** DWiSE-FT combines the strength of simple post-hoc weight interpolation with more elaborate weight ensemble, showing the improved performance while maintaining easy usage.

resolving the negative transfer issue through eliminating the spectral components corresponding to small singular values that are less transferable. The regularization is done as $\Omega(\boldsymbol{F}) = \delta \sum_{i=1}^{k} \sigma_{-i}^2$, where $\boldsymbol{F} = [f_{\boldsymbol{\theta}}(\mathcal{G}_0), \cdots, f_{\boldsymbol{\theta}}(\mathcal{G}_b)]$ is the feature matrix of a batch of graphs and $\sigma_{-i}$ are the $i$-th smallest singular values obtained from the SVD of $\boldsymbol{F}$. We can tune $k$ and $\delta$ to determine the number of singular values to penalize and the degree of penalty.

## 3 Experimental Settings in the Benchmark

In this section, we briefly introduce the experimental settings in this work. More detailed experimental settings can be found in Appendix F.

**Foundation Models**. For self-supervised pre-training, we adopt three open-source pre-trained checkpoints: *Mole-BERT*, *GraphMAE*, and *MoleculeSTM*. For supervised pre-training, we use models from the *Graphium* [32] library, which get pre-trained on the Toymix and Largemix datasets provided in this library. To differentiate between them, we refer to these models as *Graphium-Toy* and *Graphium-Large*. For larger graph transformer based model, we adopt the pre-trained checkpoint of *GraphGPS* [65] pre-trained on the PCQM4MV2 [67]. For details of datasets used in pre-training are in Appendix D. Furthermore, we include the traditional baseline XGBoost [68] for Fewshot scenarios to better compare with the foundation model in Appendix G.2.

**Downstream Datasets**. We use 8 classification and 4 regression datasets for downstream task evaluation. Detailed statistics and references for these tasks are in Appendix E.

† *Classification.* The BBBP dataset measures if a molecule will penetrate blood-brain barrier. The Tox21, ToxCast, and ClinTox datasets are related to toxicity qualitative measurements. The Sider dataset stores qualitative results of different types of adverse drug reactions. The MUV dataset is specifically designed for validation of virtual screening techniques. The HIV dataset provides qualitative activity results of the molecular ability to inhibit HIV replication. The BACE dataset contains qualitative binding results for a set of inhibitors of human $\beta$-secretase 1 (BACE-1).

† *Regression.* Esol is a dataset which measures aqueous solubility of molecules. The Lipo dataset measures the octanol-water partition coefficient. Cep is a subset of the Havard Clean Energy Project (CEP), which estimates the organic photovoltaic efficiency. Malaria measures the drug efficacy against the parasite that causes malaria.

**Dataset Splits**. For each downstream dataset, we experiment with *random, scaffold*, and *size* splits to create the Train/Val/Test subsets. Specifically, the random splitting shuffles the data, maintaining the Train/Val/Test sets as in-distribution (ID). The other two splitting methods simulate out-of-distribution (OOD) challenges in real-world applications. For scaffold splitting, we follow prior works [69], ensuring structural differences in molecular scaffolds across splits. Size splitting, following Zou et al. [70], arranges molecules in ascending order by size, evaluating model generalization across different molecule sizes.

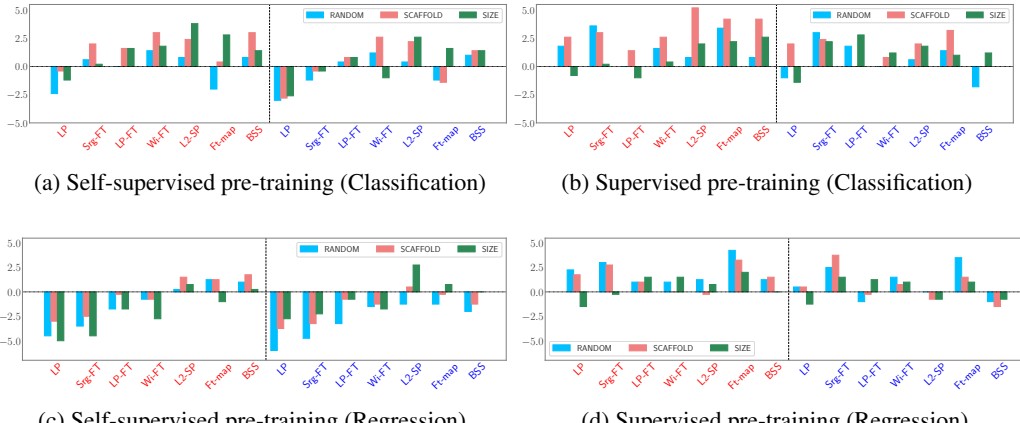

(a) Self-supervised pre-training (Classification)    (b) Supervised pre-training (Classification)

(c) Self-supervised pre-training (Regression)    (d) Supervised pre-training (Regression)

Figure 2: Average Rank improvements over Full-fine-tuning for 7 robust fine-tuning methods in self-supervised and supervised pre-training across 8 *classification* (*a, b*) datasets and across 4 *regression* (*c, d*) datasets. Each subfigure presents few-shot-50 (left of the dashed line) and few-shot-100 (right of the dashed line) settings, with **random**, **scaffold**, and **size** splits.

**Size of fine-tuning samples**. In practice, molecular property prediction tasks can have very limited experimentally-validated data, e.g., with less than 100 samples [41]. Thus, we consider both *Non-Fewshot* and *Fewshot* settings to better simulate the label scarcity issue. In the Non-Fewshot setting, we use all available samples from the splitted train set. In the Fewshot settings, we sample subsets of 50, 100, and 500 molecules from the Train set for fine-tuning, while keeping the Val/Test sets unchanged to ensure a fair comparison. Note that we exclude MUV, Tox21, and ToxCast datasets for the Fewshot settings, as we cannot *randomly* select training samples while ensuring that all tasks have a specified number of labels simultaneously, due to the severe label scarcity issues in these datasets.

**Evaluation Metrics**. We use AUC to evaluate the performance for classification datasets and RMSE for regression datasets. We report the model performance over 5 random seeds and the test performance are reported based on the best validation performance. The AVG, AVG-F, AVG-R denote the average metrics, average metrics without max and min values, and average rank over all the datasets for each evaluated method, respectively.

Table 1: A summary of evaluated pre-trained models and their corresponding result tables for reference. "CLF" and "RGS" represent classification and regression tasks, respectively, while "NON" and "FEW" denote Non-Fewshot and Fewshot settings.

| Objectives | Models | Reference Tables of Experimental Results | | | |
| --- | --- | --- | --- | --- | --- |
| | | CLF-NON | CLF-FEW | RGS-NON | RGS-FEW |
| | Mole-BERT | 2 | 6 | 3 | 7 |
| Self-Supervised | GraphMAE | 12 | 14 | 13 | 15 |
| | MoleculeSTM | 8 | 10 | 9 | 11 |
| | Graphium-Toy | 2 | 6 | 3 | 7 |
| Supervised | Graphium-Large | 8 | 10 | 9 | 11 |
| | GraphGPS | 12 | 14 | 13 | 15 |

# 4 Results and Analysis

This section mainly analyzes the experimental results from Mole-BERT and Graphium-Toy models as representatives of self-supervised and supervised pre-training. Table 1 is a summary of all pre-trained models we test on and their corresponding result tables for reference. Since we observe similar trends from pre-trained models of the same category, we will refer to them in our result analysis and compare over different pre-trained models in Sec. 4.3. Due to limited space, more findings with different fine-tuning methods and pre-trained models comparison can be found in Appendix G.

## 4.1 Self-supervised Pre-trained Models

*Q1: How does self-supervised pre-training influence downstream prediction tasks?*

**(1a) Regression tasks require more task-specific knowledge from downstream fine-tuning compared to classification tasks.**

Table 2: Robust fine-tuning performance on 8 **Classification** datasets (AUC metrics) in the **Non-Fewshot** setting, evaluated across 3 dataset splits (RANDOM, SCAFFOLD, SIZE), over **MOLE-BERT** and **GRAPHIUM-TOY** models. AVG, AVG-F, AVG-R denote the average AUC, average AUC without max and min values, and average rank over all the datasets for each method, respectively. Standard deviations across five replicates are shown. We **bold** and underline the best and second-best performances in each scenario.

| SPLIT | METHODS | CLINTOX | BBBP | BACE | HIV | MUV | SIDER | TOX21 | TOXCAST | AVG | AVG-F | AVG-R |
|---|---|---|---|---|---|---|---|---|---|---|---|---|
| | | | | | SELF-SUPERVISED PRE-TRAINING (MOLE-BERT) | | | | | | | |
| SCAFFOLD | FULL-FT | 77.70 ± 1.50 | 67.93 ± 3.85 | 80.12 ± 1.07 | 77.00 ± 0.80 | 80.50 ± 0.81 | 63.47 ± 0.77 | 78.31 ± 0.28 | 65.18 ± 0.35 | 73.78 | 74.37 | 3.75 |
| | LP | 66.49 ± 0.46 | 65.42 ± 0.26 | 78.70 ± 0.27 | 77.15 ± 0.12 | 79.27 ± 0.48 | 62.01 ± 0.60 | 78.12 ± 0.15 | 64.75 ± 0.17 | 71.49 | 71.77 | 6.12 |
| | SURGICAL-FT | 68.19 ± 1.58 | 67.70 ± 0.54 | 84.24 ± 0.37 | 76.65 ± 0.46 | 81.60 ± 1.02 | 64.61 ± 0.31 | 78.34 ± 0.10 | 65.21 ± 0.28 | 73.32 | 72.95 | 3.62 |
| | LP-FT | 70.35 ± 0.99 | 68.30 ± 0.65 | 81.90 ± 0.70 | 76.69 ± 0.40 | 77.65 ± 1.15 | 63.38 ± 0.67 | 77.60 ± 0.19 | 65.32 ± 0.24 | 72.65 | 72.65 | 4.88 |
| | WiSE-FT | 73.59 ± 3.74 | 66.52 ± 3.29 | 82.73 ± 0.87 | 77.21 ± 0.69 | 81.92 ± 0.94 | 63.62 ± 0.62 | 78.05 ± 0.28 | 65.41 ± 0.25 | 73.63 | 73.78 | 3.38 |
| | $L^2$-SP | 73.95 ± 1.86 | 67.86 ± 1.68 | 81.47 ± 0.80 | 76.63 ± 0.56 | 77.21 ± 0.72 | 65.27 ± 0.45 | 78.66 ± 0.17 | 63.55 ± 0.16 | 73.07 | 73.26 | 4.50 |
| | FEATURE-MAP | 70.65 ± 0.76 | 65.41 ± 2.37 | 73.44 ± 0.23 | 76.71 ± 0.26 | 80.03 ± 0.47 | 64.35 ± 0.17 | 76.61 ± 0.39 | 65.77 ± 0.15 | 71.62 | 71.43 | 5.25 |
| | BSS | 76.07 ± 3.23 | 67.47 ± 3.80 | 80.98 ± 1.27 | 77.12 ± 0.86 | 77.35 ± 1.76 | 63.88 ± 0.80 | 78.19 ± 0.40 | 65.00 ± 0.27 | 73.26 | 73.53 | 4.50 |
| SIZE | FULL-FT | 72.78 ± 1.74 | 87.37 ± 0.82 | 66.00 ± 1.99 | 79.85 ± 0.64 | 77.02 ± 2.15 | 52.46 ± 0.29 | 75.74 ± 0.48 | 63.13 ± 0.32 | 71.79 | 72.42 | 4.88 |
| | LP | 76.07 ± 0.32 | 82.73 ± 0.76 | 47.18 ± 0.45 | 78.16 ± 0.24 | 78.52 ± 1.60 | 51.25 ± 0.22 | 74.92 ± 0.22 | 63.33 ± 0.20 | 69.02 | 70.37 | 6.00 |
| | SURGICAL-FT | 73.55 ± 0.81 | 88.82 ± 0.53 | 66.43 ± 0.88 | 79.30 ± 0.87 | 80.52 ± 1.47 | 51.87 ± 0.23 | 76.32 ± 0.16 | 64.51 ± 0.20 | 72.66 | 73.44 | 3.50 |
| | LP-FT | 75.32 ± 0.93 | 83.42 ± 1.67 | 64.84 ± 1.38 | 79.10 ± 1.14 | 79.38 ± 1.86 | 52.82 ± 0.32 | 76.40 ± 0.28 | 63.37 ± 0.29 | 71.83 | 73.07 | 3.88 |
| | WiSE-FT | 73.45 ± 1.08 | 87.79 ± 1.53 | 66.58 ± 1.11 | 79.89 ± 1.75 | 78.41 ± 1.88 | 52.46 ± 0.49 | 76.46 ± 0.46 | 63.53 ± 0.65 | 72.32 | 73.05 | 3.00 |
| | $L^2$-SP | 73.97 ± 0.88 | 87.15 ± 0.68 | 64.58 ± 1.93 | 80.05 ± 0.53 | 74.83 ± 1.06 | 52.37 ± 0.22 | 75.84 ± 0.28 | 60.63 ± 0.36 | 71.18 | 71.65 | 5.12 |
| | FEATURE-MAP | 74.61 ± 0.53 | 85.42 ± 0.31 | 51.23 ± 0.46 | 76.39 ± 0.91 | 75.20 ± 2.27 | 51.96 ± 0.26 | 76.81 ± 0.25 | 63.42 ± 0.76 | 69.38 | 69.73 | 5.00 |
| | BSS | 73.99 ± 0.77 | 86.84 ± 1.00 | 66.97 ± 1.58 | 79.64 ± 1.44 | 73.42 ± 2.60 | 53.50 ± 0.66 | 75.69 ± 0.26 | 62.41 ± 0.69 | 71.56 | 72.02 | 4.62 |
| | | | | | SUPERVISED PRE-TRAINING (GRAPHIUM-TOY) | | | | | | | |
| SCAFFOLD | FULL-FT | 81.27 ± 3.88 | 69.17 ± 1.32 | 79.75 ± 1.07 | 76.42 ± 0.72 | 76.84 ± 1.80 | 63.63 ± 0.06 | 78.12 ± 0.46 | 66.37 ± 0.26 | 73.95 | 74.45 | 3.75 |
| | LP | 80.48 ± 0.00 | 66.90 ± 0.00 | 80.44 ± 0.00 | 75.83 ± 0.00 | 73.35 ± 0.00 | 62.03 ± 0.00 | 79.02 ± 0.00 | 66.09 ± 0.00 | 73.02 | 73.61 | 5.12 |
| | SURGICAL-FT | 86.17 ± 0.00 | 73.71 ± 0.00 | 84.16 ± 0.00 | 77.47 ± 0.00 | 78.87 ± 0.00 | 64.02 ± 0.00 | 78.23 ± 0.00 | 67.34 ± 0.00 | 76.25 | 76.63 | 1.38 |
| | LP-FT | 83.67 ± 3.53 | 69.98 ± 0.83 | 79.28 ± 0.32 | 76.17 ± 2.01 | 77.82 ± 1.15 | 61.20 ± 0.00 | 76.94 ± 0.00 | 66.28 ± 0.00 | 73.92 | 74.41 | 4.62 |
| | WiSE-FT | 85.40 ± 1.61 | 71.89 ± 1.79 | 78.13 ± 2.92 | 76.69 ± 1.76 | 74.37 ± 1.79 | 63.58 ± 0.00 | 77.98 ± 0.33 | 66.48 ± 0.43 | 74.31 | 74.26 | 3.62 |
| | $L^2$-SP | 76.83 ± 8.87 | 67.35 ± 0.82 | 78.17 ± 0.02 | 73.69 ± 0.03 | 62.35 ± 0.15 | 62.21 ± 0.45 | 76.27 ± 0.32 | 62.75 ± 0.88 | 69.95 | 69.87 | 6.62 |
| | FEATURE-MAP | 90.13 ± 2.12 | 70.99 ± 0.27 | 83.17 ± 0.49 | 73.61 ± 0.03 | 78.74 ± 0.76 | 62.12 ± 0.02 | 79.99 ± 0.12 | 65.03 ± 0.08 | 75.47 | 75.25 | 3.50 |
| | BSS | 79.99 ± 5.89 | 67.10 ± 0.93 | 78.12 ± 2.32 | 72.50 ± 0.51 | 61.20 ± 0.08 | 61.13 ± 0.95 | 76.69 ± 0.64 | 65.45 ± 0.89 | 70.27 | 70.18 | 7.38 |
| SIZE | FULL-FT | 85.96 ± 4.28 | 87.62 ± 0.90 | 67.41 ± 2.44 | 81.47 ± 1.94 | 72.03 ± 2.55 | 54.72 ± 0.01 | 69.71 ± 0.37 | 61.31 ± 0.37 | 72.53 | 72.98 | 3.88 |
| | LP | 81.84 ± 0.02 | 78.09 ± 0.00 | 58.08 ± 0.01 | 77.48 ± 0.00 | 69.46 ± 0.00 | 53.59 ± 0.00 | 73.65 ± 0.00 | 61.25 ± 0.00 | 69.18 | 69.67 | 5.38 |
| | SURGICAL-FT | 86.59 ± 0.01 | 89.07 ± 0.00 | 70.94 ± 0.01 | 82.50 ± 0.00 | 74.47 ± 0.00 | 56.24 ± 0.00 | 72.30 ± 0.00 | 62.74 ± 0.00 | 74.36 | 74.92 | 1.62 |
| | LP-FT | 86.78 ± 2.69 | 88.02 ± 1.50 | 63.72 ± 1.85 | 82.57 ± 0.46 | 73.51 ± 1.77 | 52.40 ± 0.00 | 68.23 ± 0.87 | 60.85 ± 0.00 | 72.01 | 72.61 | 4.00 |
| | WiSE-FT | 82.44 ± 3.02 | 87.76 ± 0.5 | 72.89 ± 0.66 | 81.37 ± 1.07 | 73.67 ± 3.44 | 55.87 ± 0.01 | 68.85 ± 0.84 | 60.61 ± 0.53 | 72.93 | 73.31 | 3.62 |
| | $L^2$-SP | 71.03 ± 3.67 | 81.32 ± 1.51 | 68.82 ± 0.06 | 70.66 ± 0.00 | 64.69 ± 0.32 | 52.08 ± 0.84 | 70.91 ± 0.34 | 56.50 ± 0.01 | 67.00 | 67.10 | 6.88 |
| | FEATURE-MAP | 82.48 ± 3.25 | 87.70 ± 0.64 | 69.56 ± 0.20 | 67.23 ± 1.93 | 71.49 ± 0.13 | 54.43 ± 0.03 | 74.12 ± 0.09 | 58.73 ± 0.04 | 70.72 | 70.60 | 4.38 |
| | BSS | 72.42 ± 0.03 | 82.92 ± 1.60 | 62.76 ± 4.23 | 72.81 ± 0.66 | 65.79 ± 5.31 | 52.89 ± 1.12 | 71.91 ± 0.44 | 57.79 ± 1.80 | 67.41 | 67.25 | 6.25 |

Table 3: Robust fine-tuning performance on 4 **Regression** datasets (RMSE metrics) in the **Non-Fewshot** setting, evaluated across 3 dataset splits (RANDOM, SCAFFOLD, SIZE), over **MOLE-BERT** and **GRAPHIUM-TOY** models. AVG-R, AVG-R$^*$ denote the average rank and the rank based on the average normalized performance over all the datasets for each method, respectively. Standard deviations across five replicates are shown. We **bold** and underline the best and second-best performances in each scenario.

| SPLIT | METHODS | SELF-SUPERVISED PRE-TRAINING (MOLE-BERT) | | | | | | SUPERVISED PRE-TRAINING (GRAPHIUM-TOY) | | | | | |
|---|---|---|---|---|---|---|---|---|---|---|---|---|---|
| | | ESOL | LIPO | MALARIA | CEP | AVG-R | AVG-R$^*$ | ESOL | LIPO | MALARIA | CEP | AVG-R | AVG-R$^*$ |
| SCAFFOLD | FULL-FT | 1.126 ± 0.014 | 0.728 ± 0.011 | 1.152 ± 0.015 | 1.377 ± 0.015 | 3.75 | 5 | 0.911 ± 0.041 | 0.709 ± 0.009 | 1.110 ± 0.009 | 1.419 ± 0.014 | 4.00 | 4 |
| | LP | 1.614 ± 0.010 | 0.870 ± 0.003 | 1.110 ± 0.002 | 2.006 ± 0.002 | 7.00 | 8 | 0.973 ± 0.000 | 0.881 ± 0.000 | 1.105 ± 0.000 | 1.826 ± 0.009 | 6.75 | 8 |
| | SURGICAL-FT | 1.166 ± 0.017 | 0.783 ± 0.003 | 1.120 ± 0.014 | 1.601 ± 0.006 | 5.25 | 6 | 0.892 ± 0.000 | 0.709 ± 0.000 | 1.105 ± 0.000 | 1.419 ± 0.000 | 3.50 | 2 |
| | LP-FT | 1.070 ± 0.021 | 0.730 ± 0.002 | 1.144 ± 0.022 | 1.397 ± 0.013 | 3.50 | 4 | 0.922 ± 0.004 | 0.735 ± 0.019 | 1.080 ± 0.005 | 1.368 ± 0.037 | 4.00 | 3 |
| | WiSE-FT | 1.264 ± 0.055 | 0.768 ± 0.010 | 1.072 ± 0.001 | 1.470 ± 0.029 | 4.00 | 2 | 0.888 ± 0.014 | 0.708 ± 0.008 | 1.128 ± 0.021 | 1.490 ± 0.024 | 3.75 | 6 |
| | $L^2$-SP | 1.099 ± 0.030 | 0.742 ± 0.008 | 1.101 ± 0.001 | 1.631 ± 0.006 | 3.75 | 3 | 0.948 ± 0.022 | 0.729 ± 0.015 | 1.141 ± 0.015 | 1.606 ± 0.013 | 7.00 | 7 |
| | FEATURE-MAP | 1.403 ± 0.012 | 0.842 ± 0.004 | 1.083 ± 0.002 | 1.787 ± 0.003 | 5.75 | 7 | 0.895 ± 0.016 | 0.688 ± 0.018 | 1.074 ± 0.000 | 1.472 ± 0.010 | 2.50 | 1 |
| | BSS | 1.110 ± 0.022 | 0.726 ± 0.004 | 1.125 ± 0.018 | 1.385 ± 0.018 | 3.00 | 1 | 0.896 ± 0.018 | 0.718 ± 0.018 | 1.130 ± 0.005 | 1.408 ± 0.039 | 4.50 | 5 |
| SIZE | FULL-FT | 1.419 ± 0.044 | 0.745 ± 0.008 | 0.896 ± 0.007 | 1.893 ± 0.035 | 3.25 | 3 | 1.070 ± 0.082 | 0.719 ± 0.010 | 0.886 ± 0.007 | 1.906 ± 0.006 | 4.00 | 4 |
| | LP | 2.073 ± 0.012 | 0.912 ± 0.004 | 0.921 ± 0.008 | 2.381 ± 0.006 | 8.00 | 8 | 1.115 ± 0.000 | 0.829 ± 0.000 | 0.907 ± 0.000 | 2.246 ± 0.000 | 8.00 | 8 |
| | SURGICAL-FT | 1.685 ± 0.060 | 0.775 ± 0.007 | 0.890 ± 0.005 | 2.145 ± 0.022 | 5.00 | 6 | 0.993 ± 0.000 | 0.719 ± 0.000 | 0.860 ± 0.000 | 1.906 ± 0.000 | 2.50 | 1 |
| | LP-FT | 1.440 ± 0.081 | 0.735 ± 0.013 | 0.893 ± 0.007 | 1.905 ± 0.016 | 3.50 | 2 | 1.038 ± 0.038 | 0.694 ± 0.012 | 0.883 ± 0.005 | 1.913 ± 0.031 | 2.75 | 2 |
| | WiSE-FT | 1.814 ± 0.092 | 0.831 ± 0.007 | 0.873 ± 0.005 | 1.951 ± 0.024 | 4.50 | 5 | 1.100 ± 0.005 | 0.691 ± 0.015 | 0.894 ± 0.007 | 1.943 ± 0.039 | 4.50 | 6 |
| | $L^2$-SP | 1.438 ± 0.046 | 0.799 ± 0.002 | 0.888 ± 0.005 | 2.101 ± 0.016 | 4.00 | 4 | 1.053 ± 0.026 | 0.720 ± 0.015 | 0.904 ± 0.002 | 2.122 ± 0.018 | 6.00 | 7 |
| | FEATURE-MAP | 1.656 ± 0.025 | 0.880 ± 0.011 | 0.893 ± 0.002 | 2.252 ± 0.008 | 6.25 | 7 | 0.993 ± 0.034 | 0.724 ± 0.009 | 0.884 ± 0.001 | 1.970 ± 0.013 | 4.50 | 3 |
| | BSS | 1.375 ± 0.019 | 0.731 ± 0.007 | 0.887 ± 0.010 | 1.900 ± 0.016 | 1.50 | 1 | 1.043 ± 0.022 | 0.703 ± 0.016 | 0.905 ± 0.005 | 1.890 ± 0.071 | 3.75 | 5 |

When checking the few-shot results in Fig. 2a and 2c, full fine-tuning ranks the highest for regression tasks but only achieves mid-tier performance for classification tasks. This disparity likely arises from the distinct nature of these tasks. Classification tasks typically require coarser-grained features, as exemplified by the Tox21 dataset. In this case, determining toxicity may largely rely on recognizing certain functional groups, such as toxicophores or structural alerts [71]. In contrast, regression tasks demand finer-grained features. For example, predicting precise solubility involves factors such as partial charge distribution, conformational flexibility, and hydrogen bond patterns, among others [72]. Consequently, models fine-tuned for regression tasks must acquire more downstream knowledge during the fine-tuning process and are generally less prone to overfitting compared to those used for classification tasks.

**(1b) Molecular representations learned from self-supervised pre-training are not informative enough for downstream tasks.**

As shown in Tables 2 and 3, LP is consistently the worst performing method for self-supervised pre-trained models across all data splits, even under the few-shot fine-tuning in Fig. 2a and 2c. Furthermore, this behavior is widely observed across all tested self-supervised models as GraphMAE and MoleculeSTM, which contrasts the observations in CV where LP demonstrates robust OOD performance by preserving high quality and generalizable features from pre-trained embeddings [19,

20]. We attribute this to the misalignment between general-purpose representations produced by self-supervised pre-training and the features required by the specific molecular tasks. Consequently, relying solely on tuning the classifier $g_\phi$ is insufficient to extract meaningful predictions from these non-informative representations.

Below, we summarize insightful findings from the performance of different fine-tuning strategies.

● **Finding 1. Under few-shot fine-tuning, weight-based fine-tuning strategies stand out with WiSE-FT for classification tasks and $L^2$-SP for regression tasks.**

Among various fine-tuning methods, weight-based approaches consistently outperform others across a wide range of experiments, regardless of the few-shot sample sizes (Fig. 2a and 2c). Self-supervised models are known to capture general-purpose knowledge for substructure discovery[39]. During fine-tuning, combining pre-trained and fine-tuned weights proves effective in extracting molecular patterns relevant to downstream tasks. Notably, WiSE-FT demonstrates superior performance on classification datasets, whereas $L^2$-SP excels in regression tasks. This finding is also supported by MoleculeSTM in table 11 where $L^2$-SP remains as top method under all few-shot regression tasks and WiSE-FT excels under Fewshot-50 classification. Essentially, WiSE-FT applies a straightforward post-hoc linear interpolation between pre-trained and fine-tuned models, governed by a single coefficient. In contrast, $L^2$-SP implicitly determines the weight combination through the training loss [15, 14], aligning with statement (1a) that regression tasks typically demand more nuanced modeling.

● **Finding 2. Partial FT results in underfitted molecular representations under Fewshot settings, which is more severe for regression tasks compared to classification.**

For the non-few-shot fine-tuning (Tables 2 and 3), surgical FT and LP-FT improve over full FT in both classification and regression tasks. However, in few-shot fine-tuning, both methods rank as the worst methods. This is likely because partial fine-tuning underfits and bias towards the the limited samples. This issue is more pronounced in regression tasks.

### 4.2 Supervised Pre-trained Models

*Q2: How does supervised pre-training influence downstream tasks?*

We first discuss the **task similarity** between the datasets used in the pre-training and downstream fine-tuning process. As introduced in Appendix. D, the ToyMix dataset used for supervised pre-training contains QM9, Tox21 and Zinc12K. The predictions from QM9 are not directly related to our downstream tasks, but may involve indirect correlations, as the quantum chemical properties provided by QM9 are highly valuable for characterizing molecular features. **Tox21** is an overlapping dataset that also exists as one of the downstream datasets. Its tasks in predicting qualitative toxicity measurements are *highly related* to the downstream **ClinTox** and **ToxCast** datasets, and also *correlate* to the **Sider** dataset which contains evaluation in drug side effects. Lastly, Zinc12K, which is to predict the constrained solubility, is relevant to the **Esol** and **Lipo** datasets that involve solubility predictions. Other downstream tasks *do not share* the same tasks with pre-training *directly*. Then we observe the following rules.

**(2a) Under few-shot fine-tuning, supervised pre-training models generally yield higher fine-tuning performance compared to self-supervised pre-training, regardless of the pre-training and fine-tuning task correlations.**

Supervised pre-training brings more benefits to downstream tasks than self-supervised pre-training in few-shot situations when checking Tables 6 and 7. Besides, the benefits are less relevant to the task similarity in contrast to the non-few-shot cases. For example, the improvements are also observed in HIV and Cep datasets even their tasks do not share with pre-training tasks directly. This implies that learned domain-specific knowledge still offer better insights than generic knowledge when fine-tuning guidance is minimal.

**(2b) Under non-few-shot fine-tuning, fine-tuning performance given supervised pre-training outperforms self-supervised pre-training when its objectives closely align with downstream tasks, while task misalignment may harm performance.**

From Tables 2 and 3, we observe consistent fine-tuning performance improvements over self-supervised pre-training on highly task-correlated downstream datasets including ClinTox, Esol, Lipo and Tox21. Even when pre-training involves regression tasks and downstream tasks are classification, performance gains occur if the physical meanings align. For datasets that do not directly share

Table 4: DWiSE-FT performance on 4 **Regression** datasets (RMSE metrics) in the **Fewshot** setting with 50, 100 samples, evaluated across 3 dataset splits (RANDOM, SCAFFOLD, SIZE) given **MOLE-BERT** model. AVG-R denote the average rank. Standard deviations across five replicates are shown. We **bold** and underline the best and second-best performances in each scenario.

| SPLIT | METHODS | FEWSHOT 50 | | | | | FEWSHOT 100 | | | | |
|---|---|---|---|---|---|---|---|---|---|---|---|
| | | ESOL | LIPO | MALARIA | CEP | AVG | ESOL | LIPO | MALARIA | CEP | AVG |
| RANDOM | WiSE-FT | $1.384 \pm 0.047$ | $1.212 \pm 0.020$ | $1.276 \pm 0.007$ | $2.410 \pm 0.051$ | 3.75 | $1.189 \pm 0.030$ | $1.142 \pm 0.025$ | $\mathbf{1.256 \pm 0.006}$ | $2.211 \pm 0.028$ | 3.00 |
| | $L^2$-SP | $\underline{1.372 \pm 0.029}$ | $1.196 \pm 0.019$ | $1.277 \pm 0.006$ | $2.280 \pm 0.031$ | 3.00 | $1.161 \pm 0.016$ | $1.149 \pm 0.007$ | $1.260 \pm 0.004$ | $\underline{2.131 \pm 0.014}$ | 3.25 |
| | TOP | $\mathbf{1.329 \pm 0.021}$ | $\mathbf{1.164 \pm 0.010}$ | $1.271 \pm 0.007$ | $\underline{2.275 \pm 0.022}$ | 1.25 | $\mathbf{1.120 \pm 0.038}$ | $\underline{1.139 \pm 0.017}$ | $\mathbf{1.256 \pm 0.006}$ | $2.131 \pm 0.014$ | 1.50 |
| | DWiSE-FT | $1.378 \pm 0.055$ | $\underline{1.189 \pm 0.020}$ | $\underline{1.273 \pm 0.009}$ | $\mathbf{2.222 \pm 0.059}$ | 2.00 | $\underline{1.132 \pm 0.025}$ | $\mathbf{1.138 \pm 0.028}$ | $1.256 \pm 0.004$ | $\mathbf{2.129 \pm 0.020}$ | 1.25 |
| SCAFFOLD | WiSE-FT | $1.842 \pm 0.056$ | $1.177 \pm 0.009$ | $\underline{1.162 \pm 0.004}$ | $2.454 \pm 0.043$ | 3.50 | $1.544 \pm 0.063$ | $1.041 \pm 0.017$ | $\underline{1.151 \pm 0.007}$ | $2.301 \pm 0.042$ | 3.50 |
| | $L^2$-SP | $1.699 \pm 0.049$ | $\underline{1.086 \pm 0.009}$ | $\underline{1.162 \pm 0.002}$ | $2.331 \pm 0.024$ | 2.50 | $1.473 \pm 0.009$ | $\underline{0.961 \pm 0.003}$ | $1.153 \pm 0.002$ | $2.201 \pm 0.038$ | 2.50 |
| | TOP | $1.680 \pm 0.042$ | $\mathbf{1.036 \pm 0.007}$ | $\mathbf{1.159 \pm 0.000}$ | $2.292 \pm 0.026$ | 1.25 | $1.436 \pm 0.054$ | $\mathbf{0.937 \pm 0.008}$ | $1.149 \pm 0.003$ | $2.187 \pm 0.034$ | 1.25 |
| | DWiSE-FT | $\mathbf{1.616 \pm 0.047}$ | $1.110 \pm 0.013$ | $1.173 \pm 0.005$ | $\underline{2.306 \pm 0.030}$ | 2.50 | $1.485 \pm 0.041$ | $0.979 \pm 0.014$ | $1.158 \pm 0.009$ | $\mathbf{2.149 \pm 0.040}$ | 2.75 |
| SIZE | WiSE-FT | $2.615 \pm 0.072$ | $1.391 \pm 0.042$ | $0.929 \pm 0.004$ | $2.762 \pm 0.053$ | 4.00 | $2.216 \pm 0.056$ | $1.124 \pm 0.031$ | $0.917 \pm 0.004$ | $2.543 \pm 0.027$ | 3.75 |
| | $L^2$-SP | $2.393 \pm 0.068$ | $1.306 \pm 0.037$ | $0.915 \pm 0.002$ | $\mathbf{2.497 \pm 0.019}$ | 2.50 | $\underline{1.731 \pm 0.071}$ | $1.025 \pm 0.028$ | $0.905 \pm 0.002$ | $2.424 \pm 0.024$ | 1.75 |
| | TOP | $\underline{2.369 \pm 0.075}$ | $\underline{1.297 \pm 0.040}$ | $\mathbf{0.911 \pm 0.002}$ | $2.497 \pm 0.019$ | 1.50 | $1.731 \pm 0.071$ | $1.025 \pm 0.028$ | $\mathbf{0.898 \pm 0.003}$ | $\underline{2.424 \pm 0.024}$ | 1.50 |
| | DWiSE-FT | $\mathbf{1.488 \pm 0.101}$ | $\mathbf{1.113 \pm 0.021}$ | $\underline{0.913 \pm 0.007}$ | $2.539 \pm 0.023$ | 1.75 | $\mathbf{1.469 \pm 0.052}$ | $1.031 \pm 0.022$ | $0.920 \pm 0.006$ | $\mathbf{2.390 \pm 0.025}$ | 2.25 |

tasks with pre-training, we observe mixed performance on Sider, Malaria, and Cep datasets, and even worse performance on HIV and MUV datasets. This observation contrasts to few-shot cases in (2a), which entails that downstream task specific knowledge can be learned given sufficient guidance on top of generic knowledge from self-supervised pre-training.

Below are some detailed findings with different fine-tuning methods given supervised pre-training.

● **Finding 3. Fine-tuning strategies that regularizes towards pre-trained molecular representations rank top, while weight-based methods are suboptimal.**

From non-few-shot (Tables 2 and 3) and few-shot fine-tuning (Figs. 2b and 2d) in both supervised models with ToyMix and LargeMix, surgical FT and Feature-map tend to be the top-ranking methods. However, best performing weight-based methods for self-supervised pre-training, only show mediocre performance here. This can also be observed in the larger-scale GraphGPS model as discussed in Appendix G.1. In addition, the other representation-based method BSS shows limited performance compared to Feature-map, which directly regularizes the distance to pre-trained representations. These observations suggest that given the task alignment between supervised pre-training and downstream fine-tuning, pre-trained representations tend to contain transferable features for downstream tasks. Consequently, controlling the degree to preserve pre-trained representations is the key to downstream fine-tuning performance.

## 4.3 Discussions over Pre-trained Models

Our extensive evaluation shows that the ranking of fine-tuning techniques remains consistent across pre-trained models within the same category, either supervised or self-supervised, regardless of model architecture, scale, or pre-training dataset. This suggests that our guidance for selecting fine-tuning methods based on the pre-training paradigm is broadly applicable and generalizable across diverse model designs. For instance, self-supervised models such as Mole-BERT and MoleculeSTM tend to benefit more from weight-based fine-tuning, while supervised models like Graphium and GraphGPS perform better with feature-based approaches.

## 5 Methodology Exploration

Based on findings in Sec. 4, we observe that weight-based fine-tuning generally performs well under self-supervised pre-training. However, the top strategy varies: WiSE-FT excels in classification tasks, while $L^2$-SP is more effective for regression tasks. This motivates us to further explore the connections and trade-offs between these methods to identify potential improvements. In this section, we introduce DWiSE-FT, an extension of the weight ensemble method unifying the strengths from WiSE-FT and $L^2$-SP. DWiSE-FT demonstrates top-ranking results through efficient post-processing that better suits the practical fine-tuning needs.

### 5.1 Motivation

As introduced in Sec. 2, WiSE-FT adopts the post-hoc linear interpolation between the pre-trained and fine-tuned model weights as $(1 - \alpha) \cdot \boldsymbol{\theta}_{\mathrm{pre}} + \alpha \cdot \boldsymbol{\theta}_{\mathrm{ft}}$. Although $L^2$-SP does not explicitly have weight interpolation in the form, the optimal weight $\tilde{\boldsymbol{\theta}}_{\mathrm{ft}}$ from the weight-regularized loss $\tilde{\mathcal{L}}(\boldsymbol{\theta})$ is indeed the linear interpolation of the optimal model from full FT $\boldsymbol{\theta}_{\mathrm{ft}}^*$ and the pre-trained model $\boldsymbol{\theta}_{\mathrm{pre}}$.

**Proposition 1.** *Given* $\tilde{\mathcal{L}}(\boldsymbol{\theta}) = \mathcal{L}(\boldsymbol{\theta}) + \frac{\delta}{2}\|\boldsymbol{\theta} - \boldsymbol{\theta}_{pre}\|_2^2$, *we define the optimal weights as* $\tilde{\boldsymbol{\theta}}_{ft} = \arg\min_{\boldsymbol{\theta}}\tilde{\mathcal{L}}(\boldsymbol{\theta})$ *and* $\boldsymbol{\theta}_{ft}^* = \arg\min_{\boldsymbol{\theta}}\mathcal{L}(\boldsymbol{\theta})$.

$$\mathbf{Q}^T\tilde{\boldsymbol{\theta}}_{ft} = (\boldsymbol{\Lambda} + \delta\mathbf{I})^{-1}\boldsymbol{\Lambda}\mathbf{Q}^T\boldsymbol{\theta}_{ft}^* + \delta(\boldsymbol{\Lambda} + \delta\mathbf{I})^{-1}\mathbf{Q}^T\boldsymbol{\theta}_{pre} \quad . \tag{2}$$

*where* $\boldsymbol{H}$ *is the hessian matrix of* $\mathcal{L}$ *evaluated at* $\boldsymbol{\theta}_{ft}^*$ *and* $\boldsymbol{H} = \boldsymbol{Q}\boldsymbol{\Lambda}\boldsymbol{Q}^T$.

Namely, $L^2$-SP can be seen as a more tailored weight ensemble method, employing variable mixing coefficients for different weights. This approach balances the influence of the prediction loss and the degree of weight regularization, unlike the fixed interpolation controlled by $\alpha$ across all weights in WiSE-FT. By accounting for subtle differences in loss values, $L^2$-SP is better suited for regression tasks, which are more sensitive to numerical variations.

While $L^2$-SP excels on regression datasets, its regularization coefficient is less interpretable and necessitates retraining when experimenting with different values. In contrast, WiSE-FT offers a simpler and more flexible approach, performing post-hoc interpolation without additional training once the model is fine-tuned once. Furthermore, the mixing coefficient $\alpha$ is both easy to adjust and straightforward to interpret. Therefore, our goal is to find a method that benefits from both WiSE-FT and $L^2$-SP to accommodate regression and classification tasks at the same time.

### 5.2 Algorithm

We propose DWiSE-FT that shares the framework of using the $\alpha$ to control the weight ensemble between the pre-trained model and fine-tuned model. The key idea, inspired by Eq. 4 is to enable different $\alpha$ values when ensembling the weights for different encoder layers as shown in Fig. 1. Given the pre-trained model with parameters $\boldsymbol{\theta}_{\mathrm{pre}}$ and model after full fine-tuning with parameters $\boldsymbol{\theta}_{\mathrm{ft}}$, The interpolated model has weights $\boldsymbol{\theta}^{[i]}$ with mixing coefficient $\alpha_i$ for the $i$-th layer as:

$$\boldsymbol{\theta}^{[i]} = (1 - \alpha_i) \cdot \boldsymbol{\theta}_{\mathrm{pre}}^{[i]} + \alpha_i \cdot \boldsymbol{\theta}_{\mathrm{ft}}^{[i]} \tag{3}$$

This approach naturally incorporates the characteristics of $L^2$-SP and even surgical FT: The weight ensemble in DWiSE-FT offers the flexibility through varying mixing layer-wise coefficients between the pre-trained and fine-tuned models, addressing the limitations of WiSE-FT. Additionally, we enable the selection of $\boldsymbol{\alpha}$ through optimization via validation loss gradient inspired by the Gradient-based Neural Architecture Search (NAS) [73].

### 5.3 Experiment results

Regarding the classification datasets, DWiSE-FT should have the performance at least as good as WiSE-FT since WiSE-FT is the special case of DWiSE-FT with one fixed mixing coefficient. We evaluate DWiSE-FT to see how it improves upon WiSE-FT and matches the superior performance of $L^2$-SP for regression tasks under few-shot fine-tuning. Please note that, due to space constraints, we only present the experiments for few-shot fine-tuning with 50 and 100 samples in the main text. The complete table is available in Appendix E, Table 16. In Table 4, we compare DWiSE-FT's performance against WiSE-FT, $L^2$-SP, and the best-performing method in each setting. Specifically, we find that DWiSE-FT consistently outperforms WiSE-FT. Furthermore, DWiSE-FT often surpasses $L^2$-SP or at least maintains comparable results in most scenarios. Additionally, in some cases, DWiSE-FT even exceeds the performance of the best-performing methods. Therefore, DWiSE-FT can be a great candidate for fine-tuning on regression datasets in practice since it guarantees top performance with easier usage.

## 6 Conclusion

This work benchmarks totally 8 fine-tuning methods, categorizing them into three groups, and evaluate them across 12 downstream datasets under 36 different experimental settings covering 3 dataset splits, 4 training sample sizes, and 6 molecular pre-trained models. The design of these settings reflects practical demands of molecular representation fine-tuning under 1) diversified foundation model with both supervised and self-supervised pre-training, 2) wide range of downstream tasks in both classification and regression that has not been widely studied by previous literature and 3) scarcely labeled molecules for fine-tuning. The study analyzes what is needed when facing classification vs. regression tasks and when given supervised vs. self-supervised pre-training. Then,

we provide insights in best performing fine-tuning methods accordingly under aforementioned scenarios. Additionally, we propose an extended fine-tuning method DWiSE-FT, driven by our observations, that maintains top-ranking results through a more efficient and automated design for certain fine-tuning scenarios. This highlights the value of our benchmark in offering valuable insights for both fine-tuning methodology design and practical guidance in molecular representation learning.

## 7 Acknowledgments

S. Liu, D. Zou, N. Shoghi and P. Li are partially supported by NSF awards PHY-2117997, IIS-2435957, IIS-2428777 and IIS-2239565; the JP-Morgan Chase Faculty Award, the NVIDIA Academic Grant Program and the IDEaS Cyberinfrastructure Awards.

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

## A    Proof of proposition 1

**Proposition 2.** *Given $\tilde{\mathcal{L}}(\boldsymbol{\theta}) = \mathcal{L}(\boldsymbol{\theta}) + \frac{\delta}{2}\|\boldsymbol{\theta} - \boldsymbol{\theta}_{pre}\|_2^2$, we define the optimal weights as $\tilde{\boldsymbol{\theta}}_{ft} = \arg\min_{\boldsymbol{\theta}} \tilde{\mathcal{L}}(\boldsymbol{\theta})$ and $\boldsymbol{\theta}_{ft}^* = \arg\min_{\boldsymbol{\theta}} \mathcal{L}(\boldsymbol{\theta})$.*

$$\mathbf{Q}^T \tilde{\boldsymbol{\theta}}_{ft} = (\boldsymbol{\Lambda} + \delta \mathbf{I})^{-1} \boldsymbol{\Lambda} \mathbf{Q}^T \boldsymbol{\theta}_{ft}^* + \delta(\boldsymbol{\Lambda} + \delta \mathbf{I})^{-1} \mathbf{Q}^T \boldsymbol{\theta}_{pre} \ . \tag{4}$$

*where $\boldsymbol{H}$ is the hessian matrix of $\mathcal{L}$ evaluated at $\boldsymbol{\theta}_{ft}^*$ and $\boldsymbol{H} = \boldsymbol{Q}\boldsymbol{\Lambda}\boldsymbol{Q}^T$.*

*Proof.* Based on the quadratic approximation, we can approximate $\mathcal{L}(\boldsymbol{\theta})$ as follows:

$$\mathcal{L}(\boldsymbol{\theta}) = \mathcal{L}(\boldsymbol{\theta}_{\text{ft}}^*) + \mathcal{L}'(\boldsymbol{\theta}_{\text{ft}}^*)(\boldsymbol{\theta} - \boldsymbol{\theta}_{\text{ft}}^*) + \frac{1}{2}(\boldsymbol{\theta} - \boldsymbol{\theta}_{\text{ft}}^*)^T \boldsymbol{H}(\boldsymbol{\theta} - \boldsymbol{\theta}_{\text{ft}}^*)$$

$$= \mathcal{L}(\boldsymbol{\theta}_{\text{ft}}^*) + \frac{1}{2}(\boldsymbol{\theta} - \boldsymbol{\theta}_{\text{ft}}^*)^T \boldsymbol{H}(\boldsymbol{\theta} - \boldsymbol{\theta}_{\text{ft}}^*)$$

since $\mathcal{L}'(\boldsymbol{\theta}_{\text{ft}}^*) = 0$ as $\boldsymbol{\theta}_{\text{ft}}^*$ is the minimum. Then, we add the weight regularization term, such that

$$\tilde{\mathcal{L}}(\boldsymbol{\theta}) = \mathcal{L}(\boldsymbol{\theta}_{\text{ft}}^*) + \frac{1}{2}(\boldsymbol{\theta} - \boldsymbol{\theta}_{\text{ft}}^*)^T \boldsymbol{H}(\boldsymbol{\theta} - \boldsymbol{\theta}_{\text{ft}}^*) + \delta\|\boldsymbol{\theta}_{\text{ft}} - \boldsymbol{\theta}_{\text{pre}}\|_2^2$$

Then, we solve for $\tilde{\boldsymbol{\theta}}_{\text{ft}}$ by setting $\nabla\tilde{\mathcal{L}}(\boldsymbol{\theta}) = 0$

$$\boldsymbol{H}(\tilde{\boldsymbol{\theta}}_{\text{ft}} - \boldsymbol{\theta}_{\text{ft}}^*) + \delta(\tilde{\boldsymbol{\theta}}_{\text{ft}} - \boldsymbol{\theta}_{\text{pre}}) = 0$$

$$(\boldsymbol{H} + \delta\boldsymbol{I})\tilde{\boldsymbol{\theta}}_{\text{ft}} = \boldsymbol{H}\boldsymbol{\theta}_{\text{ft}}^* + \delta\boldsymbol{\theta}_{\text{pre}}$$

$$\tilde{\boldsymbol{\theta}}_{\text{ft}} = (\boldsymbol{H} + \delta\boldsymbol{I})^{-1}(\boldsymbol{H}\boldsymbol{\theta}_{\text{ft}}^* + \delta\boldsymbol{\theta}_{\text{pre}})$$

$$\tilde{\boldsymbol{\theta}}_{\text{ft}} = (\boldsymbol{Q}\boldsymbol{\Lambda}\boldsymbol{Q}^T + \delta\boldsymbol{I})^{-1}(\boldsymbol{Q}\boldsymbol{\Lambda}\boldsymbol{Q}^T\boldsymbol{\theta}_{\text{ft}}^* + \delta\boldsymbol{\theta}_{\text{pre}})$$

$$\tilde{\boldsymbol{\theta}}_{\text{ft}} = (\boldsymbol{Q}(\boldsymbol{\Lambda} + \delta\boldsymbol{I})\boldsymbol{Q}^T)^{-1}(\boldsymbol{Q}\boldsymbol{\Lambda}\boldsymbol{Q}^T\boldsymbol{\theta}_{\text{ft}}^* + \delta\boldsymbol{\theta}_{\text{pre}})$$

$$\mathbf{Q}^T\tilde{\boldsymbol{\theta}}_{\text{ft}} = (\boldsymbol{\Lambda} + \delta\mathbf{I})^{-1}\boldsymbol{\Lambda}\mathbf{Q}^T\boldsymbol{\theta}_{\text{ft}}^* + \delta(\boldsymbol{\Lambda} + \delta\mathbf{I})^{-1}\mathbf{Q}^T\boldsymbol{\theta}_{\text{pre}}$$

$\square$

## B    Limitations and Future Works

We acknowledge certain limitations in this current work and highlight potential improvements for future research. Firstly, this study primarily focuses on the *property prediction tasks* of *small molecules* using *2D-graph* based foundation models. Exploring a broader array of foundation models across a wider range of applications–such as covering more areas like DNA, proteins, and materials, addressing various scientific tasks like linker design and chemical reactions, and incorporating diverse data formats like 3D geometric data–is highly worthwhile. Secondly, although we attempt to include many representative fine-tuning methods from various categories in this study, additional fine-tuning methods from different categories, as discussed in Appendix C, deserve investigation. For instance, future research could explore whether graph-specific fine-tuning methods offer additional benefits over non-graph fine-tuning approaches across various settings we design. Thirdly, the method DWiSE-FT introduced here is an extension and combination of existing methods directly motivated by our benchmark findings for specific fine-tuning scenarios. Future work may involve more thorough exploration into fine-tuning methodology design inspired by our current findings, and aiming to develop approaches effective across a broader range of fine-tuning scenarios.

Regarding the broader impact, we recognize our work can be beneficial to the drug discovery and material science, but people should be aware of the misuse of molecular property prediction tasks to harmful chemical production.

## C    Additional Discussions of Related Works

In this section, we additionally discuss more related works about fine-tuning (FT) techniques. Designing advanced fine-tuning strategies first gained attention in the computer vision (CV) and natural

language processing (NLP) domains, leading to the development of various research directions. We categorize the mainstream approaches into the following groups.

**Partial model FT.** Numerous studies demonstrate that freezing certain parameters while fine-tuning only specific components of the pre-trained model can help mitigate overfitting during the fine-tuning process [74, 75, 76, 77, 78, 79]. Specifically, Linear Probing (LP) only trains the additional prediction head during FT. Surgical FT [12] selectively fine-tunes a subset of layers based on the specific mechanism of distribution shifts. Partial FT is similar to the concept of parameter efficient fine-tuning methods like LoRA [80], Prefix tuning [81] and IA3 [82]. We also include an additional study on LoRA performance in App. G.3.

**Weight-based FT** strategies mainly control the model weights during the FT. Specifically, WiSE-FT [19], grounded on the linear mode connectivity [83], linearly interpolates between pre-training parameters and fine-tuning parameters by a mixing coefficient. $L^2$-SP [14] regularizes the fine-tuning model weights using $L^2$ distance to constrain the parameters around pre-trained ones. REGSL [84] further introduces a layer-wise parameter regularization, where the constraint strength gradually reduces from the top to bottom layers. MARS-SP [85] adopts the projected gradient method (PGM) to constrain the fine-tuning model weights within a small sphere centered on the pre-trained ones. More recently, TPGM [86] further incorporates trainable weight projection radii constraint for each layer, inspired by MARS-SP, to support layer-wise regularization optimization.

**Representation-based FT** methods mainly regulate the latent representation space during FT. Feature-map [13] adds distance regularization between the latent representations of pre-trained and fine-tuned models to the Full-FT loss. DELTA [87] specifically constrains feature maps with the pre-trained activations selected by channel-wise attention. BSS [17] penalizes the spectral components corresponding to small singular values that are less transferable to prevent negative transfer. Li et al. [88] proposes to transfer representations by encouraging small deviations from the reference one through an regularizer based on optimal transport. Inspired by this, GTOT-Tuning [89] presents optimal transport-based fine-tuning framework. LP-FT [20] first performs LP to prediction head while keeping the pre-trained encoder fixed, followed by applying full-FT with the tuned prediction head.

**Architecture Refinement.** Besides the weight and representation based FT, StochNorm [90] refactors the widely used Batch Normalization (BN) module and proposes Stochastic Normalization, to transfer more pre-trained knowledge during the fine-tuning process and mitigate over-fitting.

**Contrastive-based FT.** As discussed in Sec. 2, contrastive-based strategies have been widely demonstrated to be effective in the pre-training stage. There are other works which explore its effectiveness in the fine-tuning process. Gunel et al. [91], Bi-tuning [92], Core-tuning [93] and COIN [94] introduce supervised contrastive learning [95] to better leverage the label information in the target datasets with more discriminative representations as a result. More recently, FLYP [96] shows that simply finetuning a classifier via the same contrastive loss as pre-training leads to superior performance in finetuning image-text models. Oh et al. [97] fine-tunes the model with contrastive loss on additional hard negative samples, which are generated by geodesic multi-modal Mixup, for robust fine-tuning in multi-modal models.

**Graph-specific fine-tuning techniques.** Apart from the CV and NLP domains, several fine-tuning techniques specifically designed for the Graph-ML domain have recently been proposed. GTOT-Tuning [89] achieves efficient knowledge transfer from the pre-trained models by an optimal transport-based FT framework. Bridge-Tune [98] introduces an intermediate step that bridges pre-training and downstream tasks by considering the task similarity between them. G-tuning [99] tunes the pre-trained GNN so that it can reconstruct the generative patterns (graphons) of the downstream graphs. Li et al. [100] leverages expressive adapters for GNNs, to boost adaptation to the downstream tasks.

# D   Pre-training Datasets Detail

For self-supervised pre-training, *Mole-BERT* and *GraphMAE* are pre-trained over 2M molecules sampled from the ZINC15 database [101], following previous works [102]. *MoleculeSTM* is initially trained on PubChemSTM, a large multimodal dataset comprising over 280,000 chemical structure–text pairs contructed from the PubChem database [103].

For supervised pre-training, we use the models from the *Graphium* [32] library, which get pre-trained on the Toymix and Largemix datasets provided in this library. The ToyMix dataset [32], totally 2.61M graph-level data points, contains QM9 [104], Tox21 [42] and Zinc12K [105]. Specifically, QM9 consists of 19 graph-level quantum properties associated to an energy-minimized 3D conformation of the molecules. Zinc12K is to predict the constrained solubility which is the term $logP - SA - cycle$ (octanol-water partition coefficients, logP, penalized by the synthetic accessibility score, SA, and number of long cycles, cycle). The Largemix dataset, totally 343.4M graph-level data points and 197.7M node-level data points, contains four different datasets with tasks taken from quantum chemistry (PCQM4M_G25_N4), bio-assays (PCBA1328) and transcriptomics (L1000 VCAP and MCF7). Specifically, L1000 VCAP and MCF7 are from the LINCS L1000 database [106], which is generated using high-throughput transcriptomics. VCAP and MCF7 are, respectively, prostate cancer and human breast cancer cell lines. The PCQM4M_G25_N4 dataset is sourced from the PubChemQC project [107] that computed DFT properties on the energy-minimized conformation of 3.8M small molecules from PubChem. The PCBA1328 dataset, originally sourced from Wang et al. [108], comprises 1,328 assays and 1.56M molecules and contains information about a molecule's biological activity across various assay settings. The pretraining dataset for GraphGPS is PCQM4Mv2, which is a large-scale molecular dataset containing 3.75M graphs curated from PubChemQC. The task is to regress the HOMO-LUMO gap, a quantum physical property originally calculated using Density Functional Theory.

## E    Dataset Statistics

The statistics and references of the downstream datasets included in this work are shown in Table 5.

Table 5: Summary for the molecular datasets used for downstream FT, where "# TASKS" and "# MOLECULES" denote the number of tasks and molecules of each dataset, respectively.

| DATASET | EVALUATION METRICS | TASK | # TASKS | # MOLECULES |
|---|---|---|---|---|
| BBBP [109] | AUC | CLASSIFICATION | 1 | 2,039 |
| TOX21 | AUC | CLASSIFICATION | 12 | 7,831 |
| TOXCAST [110] | AUC | CLASSIFICATION | 617 | 8,576 |
| SIDER [111] | AUC | CLASSIFICATION | 27 | 1,427 |
| CLINTOX [112] | AUC | CLASSIFICATION | 2 | 1,478 |
| MUV [113] | AUC | CLASSIFICATION | 17 | 93,087 |
| HIV ZAHAREVITZ [114] | AUC | CLASSIFICATION | 1 | 41,127 |
| BACE [115] | AUC | CLASSIFICATION | 1 | 1,513 |
| ESOL [116] | RMSE | REGRESSION | 1 | 1,128 |
| LIPO [117] | RMSE | REGRESSION | 1 | 4,200 |
| MALARIA [118] | RMSE | REGRESSION | 1 | 9,999 |
| CEP [119] | RMSE | REGRESSION | 1 | 29,978 |

## F    Details of Experimental Implementation

**Pre-training Implementations.** For self-supervised pre-training, we use the open-source pre-trained checkpoints of Mole-BERT[3] and GraphMAE[4]. For supervised pre-training, we follow the same training pipeline as proposed in the Graphium[5]. We drop out the task head MLPs used for supervised pre-training during the downstream fine-tuning process, keeping only the graph encoder component. Note that we keep the architecture of the GNN encoder and the graph pooling strategy the same across the three pre-training models. Specifically, we use a 5-layer Graph Isomorphism Networks (GINs) with 300 hidden dimension and mean pooling as the readout function.

**Fine-tuning Implementations.** We keep the same training configurations across all the downstream datasets, pre-training models, and fine-tuning strategies, following Hu et al. [27]. Specifically, for

---

[3]https://github.com/junxia97/Mole-BERT
[4]https://github.com/THUDM/GraphMAE
[5]https://github.com/datamol-io/graphium

each distinct setting, we fine-tune the pre-training models with 5 random seeds (0-4). We use a batch size of 32 and a dropout rate of 0.5. For each dataset, We train models for 100 epochs and report the test performance when the optimal validation performance is achieved.

**Hyperparameter Tuning.** We set learning rate to be 0.001 for all the methods and train for 100 epochs. Below is the detailed sets of hyperparameters we tuned for each fine-tuning strategy.

- *Surgical FT:* We tune $k$ as which layer in GNN encoder to be updated from $\{0, 1, 2, 3, 4\}$ since our backbone architecture is a 5-layer GIN.

- *WiSE-FT:* We tune the mixing coefficient $\alpha$ from $\{0.1, 0.2, 0.3, 0.4, 0.5, 0.6, 0.7, 0.8, 0.9\}$ to control the weight ensemble from pre-trained model and fine-tuned model. A larger $\alpha$ indicates the weights are adopted more from the fine-tuned model.

- $L^2$-*SP/ BSS/ Feature-map:* For these three methods that involve an additional regularization term in the loss, we tune the regularization coefficient $\delta$ from $\{1, 0.1, 0.01, 0.001, 0.0001\}$ to control the degree of regularization. For BSS, we follow the original paper and set $k$ to be 1 meaning that we are regularizing the smallest singular value.

- *LP-FT:* We train the LP step before full fine-tuning for 100 epochs and then use the updated prediction head as initilization for the full-FT afterwards for 100 epochs. The training all use the default learning rate 0.001.

- *Full FT/ LP:* There is no additional hyperparameter tuning, where we use the default fine-tuning setting.

- *DWiSE-FT:* We tune the initialization of $\alpha_i$ for each layer $i$, where we use the same value to initialize for all layers from $\{0.9, 0.7, 0.5\}$ and the learning rate for validation loss descent from $\{0.001, 0.005, 0.01\}$. We tune $\alpha$ over validation sets over 200 epochs.

Indeed, from the DWiSE-FT experiments with different starting points of mixing coefficients, the variance of final results is small since it will converge towards the optimal value of mixing coefficients regardless of the initial starting point given a reasonable training time.

**Computing Resources** The experiments are run on NVIDIA RTX A6000 with 48G memory.

## G   Further Result Discussions

### G.1   Comparisons over pre-trained models

We mainly select the pretrained models based on their pre-training objective divided as supervised and self-supervised learning as discussed in 2. Then, among each category of pretrained models, we diversify with different architecture, model size and detailed training objective or pretraining dataset to discover the effect to the downstream finetuning method selection.

In the following, we will briefly discuss some more results that are not included in the main text with more pretrained models we tried. Detailed tables can be found in Appendix H

In general, we found the trend discussed in the main text about the difference of supervised pretrained model and self-supervised pretrained model hold in most cases. Especially, how they prefer over the representation based finetuning techniques or the weight based finetuning techniques remain consistent. However, some small variations may happen regarding the model size and architecture. For instance, for smaller model like 5 layer base GIN model, it is less likely to overfit on fewshot dataset compared to the larger scale graph transformer model. Also, the model expressiveness and capability will vary with different model scale. Therefore, we can compare the rank of different finetuning methods under pretrained models with the same scale, while it is not directly comparable if the model scale is significantly different.

For instance, both the Graphium model and the GraphGPS demonstrate superior performance from the representation based method like feature-map and BSS compared to other techniques. However, in contrast to the Graphium-Toy model results in the main text that feature-map perform better than BSS especially under the very few shot scenarios. In the GraphGPS results, we find that feature-map tend to be better with more finetuning samples and BSS tends to be better than feature-map in the fewshot cases. This might be due to the variation in the model size that leads to more overfitting, where BSS regularize over noisy feature space through penalizing smaller eigenvalues can be more

crucial in reducing overfitting compared to feature-map. Also, we experience a change in pretrained dataset compared to the ToyMix and LargeMix in the Graphium model, where the PCQM4Mv2 is less diversed. This might also cause the degraded performance of feature-map under GraphGPS with fewshot scenario since the learned representation from pretraining might not directly fit the downstream task. When there are more samples available, there might be a larger overlap with the learned representation space. Furthermore, we also observe the worse performance of LP and LP-FT under the larger model which coincides with findings in the main text from Graphium models.

The conclusions presented in Section 4 generalize well to models pre-trained on large-scale datasets, such as GraphGPS (pre-trained on PCQM4MV2) and Graphium-Large (trained on the LargeMix dataset containing hundreds of millions of labeled molecular graphs). In Section 4.3 and Appendix G.1, we analyze the consistency of trends across all six pre-trained models in our benchmark. Below, we summarize key observations that hold true for models trained on large-scale data:

- Supervised pre-training on large datasets leads to stronger downstream performance, particularly in few-shot settings. This aligns with our main conclusion in Section 4.2 (Q2), where we compare supervised and self-supervised pre-training. Models like GraphGPS and Graphium-Large consistently outperform self-supervised models such as Mole-BERT and GraphMAE under the same fine-tuning protocols.

- Representation-based fine-tuning methods (e.g., Feature-map and BSS) remain top-performing strategies for supervised pre-trained models on large-scale datasets, consistent with Finding 3. This trend holds across both classification and regression tasks, and across different dataset splits.

- Partial fine-tuning methods (e.g., LP and Surgical-FT) continue to underperform in few-shot settings. This observation supports Finding 2, and reflects their tendency to underfit in data-scarce regimes, even when the underlying pre-trained model is strong.

### G.2 Comparisons over traditional method

To further understand the effect from foundation model pre-training and fine-tuning process, we include the XGBoost algorithm as a representative for the traditional method. Specifically, we tested the XGBoost algorithm under the Fewshot setting with 50, 100 and 500 samples to see whether it can surpass the performance of foundation model when the training data is scarce. The featurizer being used for the XGBoost model is the Extended Connectivity Circular Fingerprints adopted from the MoleculeNet paper. Then, we keep the exact same splits with the other experiments under random, scaffold and size split. From the result in table 18, we can conclude that foundation model result (e.g.) from Mole-BERT surpass the performance in XGBoost on almost all the settings. This indicates the benefit from the pretraining and finetuning framework and the value of our work in selecting the best finetuning technique given different pretraining situation.

### G.3 Study on parameter efficient fine-tuning methods

As an additional study over parameter efficient fine-tuning method, we incorporate the LoRA [80] results for GraphGPS under the scaffold split across three regimes: Fewshot-50, Fewshot-500, and non-fewshot. The results are shown in the table 19.

Across both classification and regression tasks, LoRA falls short of full fine-tuning in roughly two-thirds of cases, with the gap widening for regression when more samples are available. This pattern is unsurprising since more challenging tasks and larger downstream datasets generally require updating a greater number of parameters. In the instances where LoRA does outperform full-FT, its results typically lie between standard full fine-tuning and the strongest fine-tuning baselines. Notably, under the Fewshot-50 regression setting, LoRA occasionally matches or even exceeds the best benchmarked fine-tuning methods, highlighting its potential in extremely low-data scenarios.

### G.4 Additional study of DWiSE-FT on other pretrained models

We additionally test DWiSE-FT on other pretrained model like GraphGPS. As shown in Table 17, we report the results of fewshot fine-tuning with 50 samples under scaffold and size splits. These results show that DWiSE-FT not only significantly improves over WiSE-FT and $L^2$-SP, but also matches or exceeds the best-performing method (TOP) in some cases. This demonstrates that DWiSE-FT

remains effective even under supervised pre-training, including on models like GraphGPS where traditional weight-based methods struggle.

### G.5 Additional findings

• **Finding 4. LP with pre-trained molecular representations from supervised pre-training surpasses full FT under few-shot fine-tuning, except for size splits.**

For few-shot fine-tuning with 50 and 100 samples (*c.f.*, Fig. 2b and 2d), LP surpasses full FT in random and scaffold splits, differing from self-supervised pre-training discussed in (**1a**). This again supports the claim that directly adopting molecular representations from supervised pre-training retain useful knowledge for downstream tasks. But interestingly, this does not hold for size splits. We believe it is due to the susceptibility of graph level tasks under size shift, as noted in prior OOD studies [70]. Namely, the prediction head tends to overfit to the mapping from representations to output labels with molecules in a specific range of sizes, and thus cannot generalize to OOD molecules of different sizes.

• **Finding 5. Regulating feature representations brings significant benefits under few-shot fine-tuning but has only a marginal impact in non-few-shot fine-tuning.**

Representation-based methods incorporates additional representation regularization in addition to full FT. BSS aims to eliminate noisy or non-transferable dimensions by regularizing small singular values of representations and Feature-map enforces a close distance of the fine-tuned representations to the pre-trained representations. Since the baseline full FT performs well under non-few-shot settings (*c.f.*, Tables 2 and 3), and pre-trained molecular representations are unsatisfying as discussed in **Q1**, having fine-tuned representations to unsatisfying pre-trained representations does not lead to any benefits. While under few-shot fine-tuning, representation regularization prevents overfitting with limited samples on top of full FT to some extend.

## H Additional Experimental Results

In this section, we present complementary baseline results over all pretrained models that are not shown in the main text due to space limit. Table 1 is a summary of all pre-trained models we test on and their corresponding result tables for reference. Also, a complete table including all few-shot fine-tuning results for DWiSE-FT (including Fewshot 500 case omitted in the main text) are in Table 16.

Table 6: Robust fine-tuning performance on 5 **Classification** datasets (AUC metrics) in the **Fewshot** setting (covering FEWSHOT-50, FEWSHOT-100, FEWSHOT-500), evaluated across 3 dataset splits (RANDOM, SCAFFOLD, SIZE), over **MOLE-BERT** and **GRAPHIUM-TOY** models. We **bold** and underline the best and second-best performances in each scenario.

| SPLIT | METHODS | Self-supervised Pre-training (Mole-BERT) | | | | | | | | Supervised Pre-training (Graphium-Toy) | | | | | | | |
|---|---|---|---|---|---|---|---|---|---|---|---|---|---|---|---|---|---|
| | | ClinTox | BBBP | BACE | HIV | SIDER | AVG | AVG-F | AVG-R | ClinTox | BBBP | BACE | HIV | SIDER | AVG | AVG-F | AVG-R |
| **FEWSHOT-50** | | | | | | | | | | | | | | | | | |
| RANDOM | FULL-FT | 74.45±2.10 | 88.56±0.83 | 75.80±0.43 | 57.41±0.69 | 52.22±0.48 | 69.69 | 69.22 | 4.40 | 70.14±0.52 | 77.57±0.01 | 80.45±0.00 | 63.57±0.00 | 55.57±0.00 | 69.46 | 70.43 | 6.00 |
| | LP | 77.50±1.31 | 82.05±0.37 | 75.04±0.58 | 53.34±2.39 | 51.40±0.11 | 67.87 | 68.63 | 6.80 | 84.09±0.00 | 81.04±0.00 | 81.57±0.00 | 49.05±0.00 | 55.62±0.00 | 70.27 | 72.74 | 4.20 |
| | SURGICAL-FT | 77.91±1.25 | 85.41±0.66 | 75.94±0.40 | 57.90±0.40 | 51.99±0.18 | 69.83 | 70.58 | 3.80 | 77.64±0.00 | 84.99±0.00 | 81.93±0.00 | 64.72±0.00 | 56.40±0.00 | 73.14 | 74.76 | 2.40 |
| | LP-FT | 77.66±0.74 | 88.99±0.14 | 75.18±0.48 | 57.38±0.37 | 51.68±0.16 | 70.18 | 70.07 | 4.40 | 69.84±0.00 | 80.15±0.00 | 78.64±0.00 | 65.82±0.00 | 53.56±0.00 | 69.60 | 71.43 | 6.00 |
| | WISE-FT | 76.12±3.87 | 88.72±1.05 | 75.59±0.51 | 58.59±0.77 | 52.23±0.50 | 70.25 | 70.10 | 3.00 | 81.94±0.03 | 83.74±0.00 | 78.47±0.00 | 63.17±0.00 | 56.44±0.00 | 72.75 | 74.53 | 4.40 |
| | L2-SP | 76.27±1.05 | 88.50±1.25 | 75.17±0.90 | 59.09±1.33 | 52.27±0.32 | 70.26 | 70.18 | 3.60 | 72.26±1.46 | 81.07±0.13 | 79.75±0.50 | 63.68±0.92 | 55.48±0.00 | 70.45 | 71.90 | 5.20 |
| | FEATURE-MAP | 74.43±2.07 | 88.40±0.84 | 73.84±0.66 | 57.93±1.13 | 51.82±0.31 | 69.28 | 68.73 | 6.40 | 84.80±0.129 | 85.33±0.021 | 81.53±0.194 | 60.64±0.016 | 56.49±0.005 | 73.76 | 75.66 | 2.60 |
| | BSS | 75.31±3.21 | 88.69±0.54 | 75.50±0.38 | 59.19±1.58 | 52.13±0.37 | 70.16 | 70.00 | 3.60 | 74.14±2.15 | 77.94±0.35 | 78.82±1.14 | 64.45±1.10 | 55.57±0.05 | 70.18 | 72.18 | 5.20 |
| SCAFFOLD | FULL-FT | 60.18±1.70 | 59.68±1.79 | 68.88±2.31 | 55.47±6.57 | 53.12±0.45 | 59.47 | 58.44 | 6.00 | 61.94±0.00 | 62.14±0.00 | 76.51±0.94 | 63.74±0.00 | 54.02±0.00 | 63.67 | 62.61 | 7.40 |
| | LP | 60.36±0.84 | 57.58±0.82 | 70.25±1.28 | 57.45±5.76 | 51.76±0.37 | 59.48 | 58.46 | 6.40 | 79.10±0.00 | 57.74±0.00 | 76.54±0.00 | 65.43±0.00 | 55.88±0.00 | 66.94 | 66.57 | 4.80 |
| | SURGICAL-FT | 60.80±1.05 | 60.86±0.98 | 71.16±0.84 | 58.60±6.33 | 52.24±0.21 | 60.73 | 60.09 | 4.00 | 71.30±0.00 | 63.24±0.00 | 76.34±0.00 | 66.81±0.00 | 56.56±0.00 | 66.85 | 67.12 | 4.40 |
| | LP-FT | 59.59±1.11 | 60.36±1.20 | 71.57±0.37 | 56.18±2.07 | 53.31±0.29 | 60.20 | 58.71 | 4.40 | 65.30±0.00 | 63.16±0.00 | 77.15±0.00 | 66.60±0.00 | 53.65±0.00 | 65.17 | 65.02 | 6.00 |
| | WISE-FT | 67.60±3.67 | 60.51±1.64 | 72.25±1.25 | 63.65±2.09 | 50.66±0.93 | 62.93 | 63.92 | 3.00 | 67.34±0.00 | 65.55±0.00 | 78.66±0.00 | 65.28±0.00 | 55.17±0.00 | 66.40 | 66.06 | 4.80 |
| | L2-SP | 61.76±1.23 | 59.53±2.09 | 70.81±0.79 | 64.76±2.40 | 52.95±0.45 | 61.96 | 62.02 | 3.60 | 83.15±0.03 | 66.76±0.00 | 78.75±0.74 | 68.22±0.02 | 55.86±0.00 | 70.55 | 71.24 | 2.00 |
| | FEATURE-MAP | 61.30±1.94 | 55.91±2.04 | 65.37±0.99 | 61.18±2.35 | 52.64±1.03 | 59.28 | 59.46 | 5.60 | 77.49±0.04 | 67.13±0.01 | 78.57±0.03 | 64.39±0.01 | 56.74±0.00 | 68.86 | 69.67 | 3.20 |
| | BSS | 67.94±2.58 | 60.40±2.18 | 70.51±1.82 | 60.39±2.23 | 53.18±0.46 | 62.48 | 62.91 | 3.00 | 69.74±0.02 | 65.64±0.00 | 79.10±0.00 | 68.47±0.01 | 54.97±0.03 | 67.58 | 67.95 | 3.20 |
| SIZE | FULL-FT | 66.75±0.92 | 80.03±0.54 | 43.23±1.52 | 62.00±3.04 | 47.81±0.77 | 59.96 | 58.85 | 5.80 | 67.61±0.01 | 71.89±5.76 | 48.57±0.01 | 52.54±0.00 | 53.48±0.00 | 58.82 | 57.88 | 5.20 |
| | LP | 69.17±0.41 | 78.19±0.32 | 39.81±0.34 | 48.97±1.66 | 46.13±0.24 | 56.45 | 54.76 | 7.00 | 71.21±0.01 | 57.79±0.00 | 40.44±0.01 | 48.13±0.00 | 55.62±0.00 | 54.64 | 53.85 | 6.00 |
| | SURGICAL-FT | 68.76±0.63 | 82.19±0.86 | 42.26±2.37 | 56.73±1.32 | 46.77±0.14 | 59.34 | 57.42 | 5.60 | 71.70±0.01 | 68.21±0.00 | 46.06±0.01 | 53.09±0.00 | 54.86±0.00 | 58.78 | 58.72 | 5.00 |
| | LP-FT | 69.43±0.30 | 82.00±0.83 | 42.83±1.39 | 61.12±1.15 | 48.77±0.32 | 60.83 | 59.77 | 4.20 | 68.90±0.01 | 65.03±0.01 | 47.57±0.00 | 47.28±0.00 | 54.15±0.00 | 56.59 | 55.58 | 6.20 |
| | WISE-FT | 70.76±1.31 | 81.92±3.19 | 65.58±2.49 | 56.58±10.19 | 47.24±0.57 | 64.42 | 64.31 | 4.00 | 72.03±0.01 | 70.14±5.65 | 45.24±0.01 | 53.43±0.00 | 53.59±0.00 | 58.89 | 59.05 | 4.80 |
| | L2-SP | 69.09±1.06 | 83.98±1.98 | 52.70±1.51 | 63.68±3.16 | 50.80±2.97 | 64.05 | 61.82 | 2.00 | 72.95±0.73 | 63.38±5.27 | 63.46±3.90 | 66.83±0.03 | 54.89±0.01 | 64.30 | 64.56 | 3.20 |
| | FEATURE-MAP | 67.57±1.45 | 82.52±0.74 | 51.61±1.25 | 66.37±3.56 | 49.65±0.57 | 63.54 | 61.85 | 3.00 | 77.65±0.06 | 71.39±0.05 | 65.20±0.00 | 57.29±0.43 | 53.01±0.01 | 64.71 | 64.63 | 3.00 |
| | BSS | 67.65±1.32 | 80.29±3.12 | 50.73±6.35 | 62.56±2.53 | 49.05±0.64 | 62.06 | 60.31 | 4.40 | 72.26±0.16 | 68.79±6.08 | 66.98±0.01 | 55.61±0.00 | 55.40±0.01 | 63.81 | 63.79 | 2.60 |
| **FEWSHOT-100** | | | | | | | | | | | | | | | | | |
| RANDOM | FULL-FT | 78.70±5.25 | 86.87±0.80 | 79.91±0.70 | 60.88±1.37 | 53.88±0.69 | 72.05 | 73.16 | 4.20 | 69.31±1.27 | 82.85±0.00 | 83.76±0.44 | 64.82±2.36 | 56.88±0.00 | 71.52 | 72.33 | 5.00 |
| | LP | 79.45±0.85 | 84.18±0.62 | 73.16±0.46 | 51.26±1.30 | 52.78±0.31 | 68.17 | 68.46 | 7.20 | 81.85±0.00 | 80.80±0.00 | 79.25±0.00 | 51.60±0.00 | 57.78±0.00 | 70.26 | 72.61 | 6.00 |
| | SURGICAL-FT | 81.54±1.62 | 85.66±0.52 | 77.00±0.74 | 59.34±0.42 | 53.63±0.44 | 71.43 | 72.63 | 5.40 | 75.51±0.00 | 86.37±0.00 | 84.51±0.00 | 66.28±0.00 | 58.87±0.00 | 74.31 | 75.43 | 2.00 |
| | LP-FT | 79.86±1.12 | 87.26±0.81 | 78.86±0.48 | 59.37±0.51 | 54.31±0.32 | 71.93 | 72.70 | 3.80 | 81.73±0.32 | 83.54±0.02 | 81.91±0.04 | 65.46±0.62 | 58.74±0.00 | 74.28 | 76.37 | 3.20 |
| | WISE-FT | 85.55±1.43 | 86.76±0.42 | 74.53±0.97 | 61.90±1.36 | 56.41±0.69 | 73.03 | 73.99 | 3.00 | 71.90±1.49 | 83.18±0.83 | 83.63±0.95 | 63.80±0.36 | 57.66±0.00 | 72.03 | 72.96 | 5.00 |
| | L2-SP | 79.13±3.68 | 86.89±0.40 | 79.66±0.35 | 59.92±1.04 | 54.64±0.35 | 72.05 | 72.90 | 3.80 | 76.28±0.02 | 81.15±1.52 | 80.71±1.44 | 64.00±0.98 | 59.02±0.54 | 72.23 | 73.66 | 4.40 |
| | FEATURE-MAP | 78.12±3.01 | 87.80±0.62 | 73.50±0.69 | 59.97±0.75 | 53.50±0.24 | 70.58 | 70.53 | 5.40 | 82.51±0.15 | 85.94±0.56 | 82.09±1.02 | 63.34±0.11 | 57.82±0.05 | 74.34 | 75.98 | 3.60 |
| | BSS | 79.00±4.62 | 87.38±0.52 | 80.12±0.33 | 60.22±1.07 | 53.88±0.72 | 72.12 | 73.11 | 3.20 | 72.38±1.42 | 80.11±0.78 | 81.64±0.64 | 63.65±0.65 | 56.85±0.81 | 70.93 | 72.05 | 6.80 |
| SCAFFOLD | FULL-FT | 70.51±0.51 | 62.11±1.32 | 68.39±3.19 | 61.60±1.74 | 52.20±0.26 | 62.96 | 64.03 | 4.80 | 70.75±0.00 | 65.39±0.25 | 77.66±0.30 | 59.73±0.00 | 54.53±0.00 | 65.61 | 65.29 | 5.80 |
| | LP | 60.68±0.68 | 58.10±0.99 | 69.41±1.69 | 57.12±4.63 | 52.11±0.51 | 59.48 | 58.63 | 7.60 | 80.09±0.00 | 53.89±0.00 | 78.39±0.00 | 64.11±0.00 | 56.03±0.00 | 66.50 | 66.19 | 3.80 |
| | SURGICAL-FT | 65.93±0.93 | 61.45±1.01 | 70.20±1.91 | 59.62±0.64 | 52.49±0.67 | 61.94 | 62.33 | 5.20 | 75.08±0.00 | 64.49±0.00 | 78.42±0.00 | 67.41±0.00 | 54.87±0.00 | 68.05 | 68.99 | 3.40 |
| | LP-FT | 66.18±2.14 | 61.52±0.91 | 71.48±0.58 | 60.76±1.04 | 53.68±0.46 | 62.72 | 62.82 | 4.00 | 67.42±0.00 | 66.33±0.00 | 74.91±0.44 | 64.40±0.00 | 53.25±0.00 | 65.26 | 66.05 | 5.80 |
| | WISE-FT | 64.71±2.82 | 62.88±2.30 | 75.95±1.63 | 62.67±2.42 | 54.27±0.82 | 64.10 | 63.42 | 2.20 | 74.35±0.00 | 64.90±0.06 | 78.06±0.96 | 62.56±0.00 | 54.55±0.00 | 66.88 | 67.27 | 5.00 |
| | L2-SP | 70.98±2.49 | 61.93±2.03 | 72.49±0.86 | 66.43±0.76 | 52.51±0.93 | 64.87 | 66.45 | 2.60 | 74.06±0.20 | 66.11±0.00 | 77.15±0.00 | 72.98±1.69 | 54.82±0.78 | 69.03 | 71.06 | 3.00 |
| | FEATURE-MAP | 63.83±1.60 | 58.78±1.66 | 67.61±0.30 | 58.27±3.68 | 53.97±1.51 | 60.49 | 60.29 | 6.20 | 79.79±0.36 | 63.60±0.03 | 78.91±0.38 | 69.71±0.32 | 56.33±0.63 | 69.67 | 70.74 | 2.60 |
| | BSS | 70.99±1.94 | 62.47±0.62 | 69.47±2.49 | 62.09±0.93 | 52.22±0.33 | 63.45 | 64.68 | 3.40 | 68.24±1.75 | 65.35±0.00 | 78.61±0.00 | 61.43±0.16 | 53.73±0.45 | 65.45 | 65.01 | 5.80 |
| SIZE | FULL-FT | 72.17±2.23 | 80.54±1.53 | 59.53±0.71 | 61.90±2.19 | 48.97±0.30 | 64.62 | 64.53 | 4.80 | 73.66±0.01 | 81.77±0.00 | 60.31±4.27 | 59.36±4.03 | 54.37±0.00 | 65.89 | 64.44 | 5.60 |
| | LP | 68.13±0.43 | 81.53±0.52 | 49.67±2.12 | 46.66±3.40 | 47.08±0.22 | 58.61 | 54.96 | 7.40 | 72.12±0.01 | 52.13±0.00 | 47.81±0.07 | 47.18±0.00 | 55.11±0.00 | 54.87 | 51.68 | 7.00 |
| | SURGICAL-FT | 70.80±0.56 | 83.61±0.40 | 58.55±3.14 | 55.86±1.29 | 47.75±0.49 | 63.31 | 61.74 | 5.20 | 78.60±0.01 | 80.76±0.00 | 56.62±0.01 | 66.14±0.00 | 55.12±0.00 | 67.45 | 67.12 | 3.40 |
| | LP-FT | 68.05±0.12 | 83.02±0.40 | 59.92±1.08 | 60.87±1.57 | 50.40±0.29 | 64.57 | 62.95 | 4.00 | 76.90±2.09 | 85.29±0.00 | 66.72±0.02 | 51.80±0.00 | 56.61±0.00 | 67.46 | 66.74 | 2.80 |
| | WISE-FT | 71.91±1.19 | 81.89±5.23 | 55.66±2.06 | 53.27±8.19 | 48.26±0.31 | 62.20 | 60.28 | 5.80 | 73.22±0.01 | 82.39±0.00 | 62.81±1.46 | 61.23±0.03 | 54.99±0.00 | 66.93 | 65.75 | 4.40 |
| | L2-SP | 73.25±1.91 | 83.39±0.71 | 60.46±1.08 | 63.14±2.17 | 50.74±2.54 | 66.20 | 65.62 | 2.20 | 76.11±2.63 | 75.35±0.41 | 66.17±0.04 | 74.02±1.42 | 54.76±0.88 | 69.28 | 71.85 | 3.80 |
| | FEATURE-MAP | 69.78±2.65 | 83.55±1.25 | 62.51±1.38 | 57.64±3.25 | 51.26±0.38 | 64.95 | 63.31 | 3.20 | 76.90±0.04 | 76.51±0.06 | 61.49±3.16 | 62.51±1.43 | 54.57±0.09 | 66.40 | 66.84 | 4.60 |
| | BSS | 73.74±2.81 | 80.91±1.12 | 60.12±1.15 | 63.05±2.33 | 50.20±0.94 | 65.60 | 65.64 | 3.40 | 78.11±7.47 | 73.92±0.09 | 64.84±0.40 | 68.42±0.08 | 53.54±1.60 | 67.77 | 69.06 | 4.40 |
| **FEWSHOT-500** | | | | | | | | | | | | | | | | | |
| RANDOM | FULL-FT | 86.07±1.80 | 92.76±0.54 | 85.99±0.40 | 67.49±0.86 | 61.33±0.24 | 78.73 | 79.85 | 3.40 | 88.53±1.79 | 91.44±1.06 | 83.72±0.59 | 70.25±1.76 | 58.51±0.00 | 78.49 | 80.83 | 4.20 |
| | LP | 84.85±0.40 | 87.91±0.20 | 73.59±0.24 | 55.25±0.21 | 59.54±0.14 | 72.23 | 72.66 | 7.60 | 91.56±0.00 | 85.15±0.00 | 83.18±0.00 | 66.82±0.00 | 58.78±0.00 | 77.10 | 78.38 | 4.20 |
| | SURGICAL-FT | 87.77±0.56 | 92.14±0.57 | 92.20±0.29 | 85.79±0.37 | 68.44±0.80 | 78.28 | 79.87 | 4.40 | 91.31±0.00 | 92.11±0.00 | 84.49±0.00 | 69.71±0.00 | 59.93±0.00 | 79.51 | 81.84 | 2.40 |
| | LP-FT | 85.55±0.75 | 92.20±0.29 | 85.79±0.37 | 68.44±0.80 | 61.06±0.55 | 78.61 | 79.93 | 3.60 | 88.82±1.84 | 91.07±0.99 | 83.89±0.00 | 66.62±0.69 | 57.89±0.00 | 77.66 | 79.78 | 5.20 |
| | WISE-FT | 87.70±1.47 | 91.02±0.72 | 85.36±0.44 | 62.00±2.20 | 64.11±0.55 | 78.04 | 79.06 | 4.00 | 89.75±1.06 | 92.30±0.39 | 83.58±0.00 | 66.27±2.15 | 58.65±0.00 | 78.11 | 79.87 | 4.20 |
| | L2-SP | 85.46±1.06 | 92.44±0.82 | 85.11±0.32 | 68.42±0.77 | 59.37±0.56 | 78.16 | 79.66 | 5.00 | 85.29±4.89 | 82.38±1.17 | 80.83±0.91 | 66.64±1.36 | 57.95±0.76 | 74.62 | 76.62 | 6.60 |
| | FEATURE-MAP | 83.42±3.42 | 90.57±0.49 | 76.69±0.41 | 68.24±0.93 | 59.62±0.36 | 75.71 | 76.12 | 6.40 | 91.58±0.23 | 91.80±0.46 | 85.29±0.81 | 72.78±0.13 | 60.19±0.04 | 80.33 | 83.22 | 1.40 |
| | BSS | 86.17±1.34 | 92.76±0.38 | 86.04±0.32 | 69.34±0.40 | 61.45±0.51 | 79.15 | 80.52 | 1.60 | 82.20±1.72 | 81.21±1.30 | 83.13±1.36 | 64.65±1.05 | 57.16±0.83 | 73.67 | 76.02 | 7.80 |
| SCAFFOLD | FULL-FT | 69.18±2.51 | 69.56±0.99 | 79.14±0.95 | 69.86±1.35 | 56.92±0.20 | 68.93 | 69.53 | 4.20 | 77.16±1.95 | 67.79±0.50 | 74.30±3.48 | 64.63±2.67 | 57.97±0.00 | 68.37 | 68.91 | 6.00 |
| | LP | 61.91±0.52 | 64.03±0.55 | 77.67±0.10 | 66.13±1.48 | 59.60±0.30 | 65.87 | 64.02 | 6.60 | 81.39±0.00 | 65.24±0.00 | 80.66±0.00 | 67.92±0.00 | 56.93±0.00 | 70.83 | 71.27 | 4.20 |
| | SURGICAL-FT | 66.75±0.43 | 67.11±0.80 | 80.66±0.43 | 72.20±0.83 | 58.92±0.38 | 69.13 | 68.69 | 4.00 | 80.56±0.00 | 70.07±0.00 | 80.77±0.00 | 72.03±0.00 | 54.85±0.00 | 71.74 | 74.35 | 3.40 |
| | LP-FT | 69.91±1.83 | 68.58±0.18 | 78.46±0.74 | 69.38±0.59 | 58.07±0.20 | 68.88 | 69.29 | 4.20 | 85.20±1.39 | 68.48±0.55 | 77.44±0.32 | 66.97±0.52 | 54.41±0.00 | 70.50 | 70.96 | 5.20 |
| | WISE-FT | 68.66±1.86 | 64.82±1.71 | 82.01±0.60 | 72.95±0.97 | 60.35±1.11 | 69.76 | 68.81 | 3.20 | 80.96±1.12 | 68.94±0.8 | 80.28±0.18 | 64.84±3.83 | 57.45±0.02 | 70.49 | 71.35 | 4.40 |
| | L2-SP | 69.22±2.59 | 68.11±0.95 | 77.74±1.08 | 73.06±0.43 | 58.86±0.63 | 69.40 | 70.13 | 3.80 | 71.73±4.37 | 66.66±0.75 | 77.77±0.03 | 69.70±0.04 | 56.84±1.27 | 68.74 | 69.70 | 6.00 |
| | FEATURE-MAP | 66.14±1.79 | 64.83±2.23 | 72.50±0.52 | 71.49±1.13 | 59.56±0.29 | 66.90 | 67.49 | 5.60 | 83.65±0.24 | 70.95±0.40 | 82.56±0.05 | 73.09±0.29 | 59.58±0.07 | 73.97 | 75.53 | 1.40 |
| | BSS | 69.65±1.86 | 69.04±0.33 | 78.20±1.39 | 70.85±0.75 | 56.75±0.46 | 68.90 | 69.85 | 4.40 | 74.20±5.33 | 69.90±0.30 | 80.08±0.35 | 73.95±0.94 | 57.05±0.91 | 69.94 | 71.42 | 5.00 |
| SIZE | FULL-FT | 74.96±1.19 | 87.81±1.32 | 54.53±1.81 | 65.86±0.67 | 51.08±0.59 | 66.85 | 65.12 | 3.60 | 70.32±4.85 | 82.67±0.65 | 59.41±0.01 | 71.78±4.10 | 53.99±0.00 | 67.63 | 67.17 | 5.00 |
| | LP | 67.80±0.62 | 82.24±0.47 | 48.77±0.42 | 52.20±3.32 | 50.51±0.37 | 60.30 | 56.84 | 7.20 | 75.60±0.01 | 75.14±0.00 | 50.85±0.10 | 58.39±0.00 | 54.81±0.00 | 62.96 | 62.74 | 6.20 |
| | SURGICAL-FT | 70.35±0.30 | 88.56±0.70 | 60.12±1.38 | 61.09±0.81 | 51.85±0.40 | 66.39 | 63.65 | 3.20 | 77.94±0.01 | 88.47±0.00 | 52.64±0.01 | 69.53±0.00 | 54.82±0.00 | 68.72 | 67.49 | 4.00 |
| | LP-FT | 71.38±0.64 | 86.43±0.68 | 53.50±1.98 | 65.30±0.73 | 49.99±0.30 | 65.32 | 63.39 | 6.20 | 75.59±1.96 | 83.51±1.98 | 49.10±3.02 | 71.61±4.67 | 55.43±0.00 | 67.05 | 67.54 | 4.20 |
| | WISE-FT | 73.53±1.46 | 86.56±1.25 | 65.74±1.37 | 51.55±9.46 | 48.62±0.38 | 65.20 | 63.61 | 5.20 | 68.48±2.42 | 83.96±1.05 | 59.04±0.00 | 60.53±3.71 | 55.22±0.00 | 66.36 | 66.31 | 4.20 |
| | L2-SP | 73.43±1.31 | 86.82±1.64 | 56.73±3.41 | 67.80±1.83 | 51.01±0.60 | 67.16 | 65.99 | 3.80 | 74.24±5.74 | 78.60±2.29 | 59.94±0.02 | 73.61±1.82 | 55.14±1.49 | 68.31 | 69.26 | 3.60 |
| | FEATURE-MAP | 76.06±0.62 | 81.83±0.64 | 58.42±0.90 | 67.94±1.41 | 50.84±0.30 | 67.02 | 67.47 | 3.60 | 80.69±0.11 | 88.49±0.80 | 58.95±0.13 | 67.62±2.74 | 54.76±0.09 | 70.10 | 69.09 | 4.00 |
| | BSS | 74.26±1.07 | 88.06±0.96 | 56.71±1.82 | 66.29±1.10 | 52.91±0.65 | 67.65 | 65.75 | 2.80 | 68.01±0.70 | 79.45±2.68 | 59.39±6.07 | 71.78±1.54 | 54.88±1.50 | 66.70 | 66.39 | 4.80 |

Table 7: Robust fine-tuning performance on 4 **Regression** datasets (RMSE metrics) in the **Fewshot** setting (covering FEWSHOT-50, FEWSHOT-100, and FEWSHOT-500), evaluated across 3 dataset splits (RANDOM, SCAFFOLD, SIZE), over **MOLE-BERT** and **GRAPHIUM-TOY** models. AVG-R, AVG-R* denote the average rank and the rank based on the average normalized performance over all the datasets for each evavluated method, respectively. Standard deviations across five replicates are shown in parentheses. We **bold** and underline the best and second-best performances in each scenario.

| Split | Methods | Esol | Lipo | Malaria | CEP | Avg-R | Avg-R* | Esol | Lipo | Malaria | CEP | Avg-R | Avg-R* |
|---|---|---|---|---|---|---|---|---|---|---|---|---|---|
| | | SELF-SUPERVISED PRE-TRAINING (MOLE-BERT) | | | | | | SUPERVISED PRE-TRAINING (GRAPHIUM-TOY) | | | | | |
| | | **FEWSHOT-50** | | | | | | | | | | | |
| RANDOM | FULL-FT | 1.390±0.051 | 1.189±0.016 | 1.276±0.006 | 2.383±0.046 | 3.50 | 4 | 1.223±0.000 | 1.062±0.000 | 1.284±0.000 | 2.359±0.000 | 6.25 | 7 |
| | LP | 2.654±0.016 | 1.825±0.011 | 1.296±0.005 | 3.736±0.020 | 8.00 | 8 | **1.085±0.000** | 1.072±0.000 | **1.272±0.000** | 2.571±0.000 | 4.00 | 3 |
| | SURGICAL-FT | 2.647±0.022 | 1.618±0.014 | 1.295±0.004 | 3.596±0.037 | 7.00 | 7 | 1.174±0.000 | **1.009±0.000** | 1.277±0.000 | 2.355±0.000 | 3.25 | 2 |
| | LP-FT | 1.422±0.027 | 1.237±0.027 | 1.291±0.005 | 2.296±0.012 | 5.25 | 6 | 1.386±0.000 | 1.019±0.000 | 1.286±0.000 | 2.287±0.000 | 5.25 | 8 |
| | WiSE-FT | 1.384±0.047 | 1.212±0.020 | 1.276±0.007 | 2.410±0.051 | 4.25 | 5 | 1.219±0.000 | 1.060±0.000 | 1.280±0.000 | 2.366±0.000 | 5.25 | 4 |
| | L2-SP | 1.372±0.029 | 1.196±0.019 | 1.277±0.006 | 2.280±0.031 | 3.25 | 3 | 1.147±0.026 | 1.092±0.001 | 1.283±0.000 | 2.312±0.020 | 5.00 | 5 |
| | FEATURE-MAP | **1.329±0.021** | **1.164±0.010** | **1.271±0.007** | 2.448±0.010 | 2.25 | 1 | 1.089±0.001 | 1.046±0.000 | 1.276±0.000 | **2.191±0.017** | 2.00 | 1 |
| | BSS | 1.365±0.028 | 1.186±0.017 | 1.277±0.006 | **2.275±0.022** | 2.50 | 2 | 1.175±0.011 | 1.128±0.035 | 1.281±0.000 | 2.262±0.064 | 5.00 | 6 |
| SCAFFOLD | FULL-FT | 1.696±0.058 | 1.124±0.006 | 1.178±0.005 | 2.356±0.033 | 4.25 | 5 | 1.353±0.000 | 1.071±0.000 | 1.168±0.000 | 2.001±0.000 | 5.75 | 8 |
| | LP | 3.754±0.020 | 1.858±0.005 | 1.167±0.002 | 3.849±0.009 | 7.25 | 8 | 1.226±0.000 | 1.013±0.000 | 1.166±0.000 | 2.450±0.000 | 4.00 | 6 |
| | SURGICAL-FT | 3.599±0.039 | 1.843±0.006 | 1.167±0.003 | 3.819±0.017 | 6.75 | 7 | 1.239±0.000 | 1.019±0.000 | **1.162±0.000** | 2.083±0.000 | 3.00 | 2 |
| | LP-FT | 1.822±0.014 | 1.134±0.012 | 1.184±0.004 | **2.292±0.026** | 4.50 | 6 | 1.283±0.000 | 1.033±0.000 | 1.169±0.000 | **1.949±0.000** | 4.75 | 5 |
| | WiSE-FT | 1.842±0.056 | 1.177±0.009 | 1.162±0.004 | 2.454±0.043 | 5.00 | 4 | 1.320±0.000 | 1.071±0.000 | 1.168±0.000 | 1.992±0.000 | 5.75 | 7 |
| | L2-SP | 1.699±0.049 | 1.086±0.009 | 1.162±0.002 | 2.331±0.024 | 2.75 | 2 | 1.273±0.047 | 1.015±0.007 | 1.166±0.000 | 2.132±0.048 | 6.00 | 4 |
| | FEATURE-MAP | 1.823±0.028 | **1.036±0.007** | **1.159±0.000** | 2.425±0.012 | 3.00 | 1 | **1.213±0.001** | **0.991±0.000** | 1.164±0.000 | 2.128±0.006 | 2.50 | 1 |
| | BSS | **1.680±0.042** | 1.114±0.008 | 1.165±0.001 | 2.319±0.025 | 2.50 | 3 | 1.222±0.012 | 1.039±0.000 | 1.166±0.000 | 2.121±0.029 | 4.25 | 3 |
| SIZE | FULL-FT | 2.382±0.079 | **1.297±0.040** | 0.929±0.004 | 2.656±0.039 | 2.75 | 4 | 1.441±0.000 | 1.055±0.000 | 0.914±0.000 | 2.329±0.000 | 5.00 | 7 |
| | LP | 4.534±0.021 | 2.157±0.012 | 0.941±0.004 | 4.706±0.022 | 7.75 | 8 | 1.443±0.000 | 1.003±0.000 | 0.936±0.000 | 2.688±0.000 | 6.50 | 8 |
| | SURGICAL-FT | 4.344±0.026 | 2.111±0.021 | 0.943±0.004 | 4.265±0.028 | 7.25 | 7 | 1.469±0.000 | 1.015±0.000 | 0.914±0.000 | 2.313±0.000 | 5.25 | 5 |
| | LP-FT | 2.421±0.060 | 1.395±0.018 | 0.939±0.007 | 2.525±0.013 | 4.50 | 6 | **1.395±0.000** | 0.999±0.000 | 0.907±0.000 | 2.410±0.000 | 3.50 | 1 |
| | WiSE-FT | 2.615±0.072 | 1.391±0.042 | 0.929±0.004 | 2.762±0.053 | 5.50 | 5 | 1.411±0.000 | 1.071±0.000 | **0.905±0.000** | 2.324±0.000 | 3.50 | 4 |
| | L2-SP | 2.393±0.068 | 1.306±0.037 | 0.915±0.002 | **2.497±0.019** | 2.00 | 2 | 1.446±0.055 | 0.997±0.000 | 0.908±0.000 | 2.340±0.020 | 4.25 | 3 |
| | FEATURE-MAP | 2.422±0.021 | 1.327±0.022 | **0.911±0.002** | 2.659±0.021 | 3.75 | 1 | 1.415±0.005 | **0.989±0.027** | 0.921±0.002 | **2.254±0.001** | 3.00 | 2 |
| | BSS | **2.369±0.075** | 1.319±0.050 | 0.925±0.003 | 2.563±0.022 | 2.50 | 3 | 1.499±0.028 | 0.997±0.000 | 0.907±0.000 | 2.381±0.006 | 5.00 | 6 |
| | | **FEWSHOT-100** | | | | | | | | | | | |
| RANDOM | FULL-FT | 1.141±0.030 | 1.141±0.023 | **1.256±0.006** | 2.150±0.021 | 2.00 | 1 | 1.191±0.000 | 1.103±0.000 | 1.258±0.000 | 2.076±0.015 | 5.25 | 4 |
| | LP | 2.273±0.029 | 1.569±0.008 | 1.280±0.003 | 3.235±0.019 | 8.00 | 8 | 1.066±0.000 | 1.045±0.000 | 1.267±0.000 | 2.383±0.000 | 4.75 | 5 |
| | SURGICAL-FT | 1.953±0.039 | 1.281±0.020 | 1.270±0.006 | 3.019±0.047 | 6.75 | 7 | 1.075±0.000 | 1.030±0.000 | 1.266±0.000 | **1.935±0.000** | 2.75 | 2 |
| | LP-FT | 1.244±0.057 | 1.147±0.018 | 1.277±0.003 | 2.156±0.019 | 5.25 | 6 | 1.689±0.000 | 1.097±0.000 | 1.273±0.000 | 2.044±0.015 | 6.25 | 8 |
| | WiSE-FT | 1.189±0.030 | 1.142±0.025 | **1.256±0.006** | 2.211±0.028 | 3.50 | 2 | 1.131±0.000 | 1.078±0.000 | **1.256±0.000** | 2.001±0.071 | 3.75 | 3 |
| | L2-SP | 1.161±0.016 | 1.149±0.007 | 1.260±0.004 | **2.131±0.014** | 3.25 | 4 | 1.098±0.012 | 1.077±0.001 | 1.270±0.000 | 2.261±0.008 | 5.25 | 6 |
| | FEATURE-MAP | **1.120±0.038** | **1.139±0.017** | 1.266±0.004 | 2.283±0.011 | 3.25 | 5 | **0.995±0.018** | **1.025±0.000** | 1.258±0.003 | 1.937±0.023 | 1.75 | 1 |
| | BSS | 1.199±0.033 | 1.149±0.023 | 1.259±0.006 | 2.132±0.019 | 4.00 | 3 | 1.055±0.009 | 1.136±0.000 | 1.274±0.000 | 2.269±0.010 | 6.25 | 7 |
| SCAFFOLD | FULL-FT | 1.436±0.054 | 1.026±0.009 | 1.160±0.011 | 2.198±0.034 | 3.25 | 4 | 1.111±0.000 | 1.037±0.000 | 1.172±0.000 | 1.965±0.023 | 5.00 | 6 |
| | LP | 3.255±0.025 | 1.503±0.008 | 1.154±0.003 | 3.350±0.007 | 7.00 | 8 | 1.228±0.000 | **0.960±0.000** | 1.162±0.000 | 2.423±0.000 | 4.50 | 5 |
| | SURGICAL-FT | 2.587±0.076 | 1.192±0.015 | 1.156±0.003 | 2.914±0.066 | 6.50 | 7 | **1.087±0.000** | 0.966±0.000 | **1.156±0.000** | **1.959±0.000** | 1.25 | 1 |
| | LP-FT | 1.544±0.042 | 1.010±0.011 | 1.163±0.004 | **2.187±0.034** | 4.00 | 6 | 1.111±0.000 | 0.984±0.000 | 1.173±0.000 | 2.149±0.012 | 5.25 | 4 |
| | WiSE-FT | 1.544±0.063 | 1.041±0.017 | 1.151±0.007 | 2.301±0.042 | 4.50 | 3 | 1.110±0.000 | 1.027±0.000 | 1.169±0.000 | 2.013±0.049 | 4.25 | 3 |
| | L2-SP | 1.473±0.009 | 0.961±0.003 | 1.153±0.002 | 2.201±0.038 | 2.75 | 2 | 1.252±0.021 | 0.994±0.013 | 1.163±0.000 | 2.367±0.052 | 5.75 | 7 |
| | FEATURE-MAP | 1.677±0.020 | **0.937±0.008** | 1.149±0.003 | 2.356±0.018 | 3.50 | 1 | 1.158±0.020 | 0.966±0.010 | 1.161±0.000 | 2.024±0.019 | 3.50 | 2 |
| | BSS | **1.463±0.008** | 1.040±0.018 | 1.160±0.006 | 2.253±0.006 | 4.50 | 5 | 1.253±0.027 | 1.033±0.015 | 1.167±0.000 | 2.333±0.022 | 6.50 | 8 |
| SIZE | FULL-FT | 1.889±0.065 | 1.077±0.028 | 0.918±0.005 | **2.425±0.024** | 4.00 | 3 | 1.411±0.000 | **0.962±0.000** | 0.921±0.006 | 2.328±0.015 | 4.75 | 5 |
| | LP | 3.851±0.033 | 1.676±0.025 | 0.911±0.003 | 4.115±0.038 | 6.75 | 8 | 1.253±0.000 | 0.981±0.000 | 0.924±0.000 | 2.635±0.000 | 6.00 | 8 |
| | SURGICAL-FT | 3.237±0.085 | 1.374±0.031 | 0.912±0.002 | 3.174±0.048 | 6.25 | 7 | 1.329±0.000 | 0.965±0.000 | 0.910±0.000 | **2.283±0.000** | 3.25 | 2 |
| | LP-FT | 1.831±0.066 | 1.085±0.014 | 0.920±0.008 | 2.468±0.021 | 4.75 | 4 | **1.242±0.000** | **0.962±0.000** | 0.912±0.000 | 2.375±0.000 | 3.50 | 1 |
| | WiSE-FT | 2.216±0.056 | 1.124±0.031 | 0.917±0.004 | 2.543±0.027 | 5.75 | 5 | 1.398±0.000 | 0.963±0.000 | 0.907±0.002 | 2.319±0.014 | 3.75 | 4 |
| | L2-SP | **1.731±0.071** | **1.025±0.028** | 0.905±0.002 | **2.424±0.024** | 1.25 | 1 | 1.418±0.035 | 0.998±0.038 | **0.906±0.000** | 2.436±0.072 | 5.50 | 6 |
| | FEATURE-MAP | 2.135±0.077 | 1.049±0.013 | **0.898±0.003** | 2.500±0.017 | 3.25 | 2 | 1.335±0.005 | 0.967±0.008 | 0.911±0.001 | **2.265±0.020** | 3.75 | 3 |
| | BSS | 1.734±0.060 | 1.073±0.024 | 0.931±0.008 | 2.439±0.015 | 4.00 | 6 | 1.387±0.039 | 0.998±0.006 | **0.906±0.004** | 2.518±0.137 | 5.50 | 7 |
| | | **FEWSHOT-500** | | | | | | | | | | | |
| RANDOM | FULL-FT | 0.883±0.032 | 0.817±0.012 | 1.194±0.003 | 1.891±0.026 | 2.50 | 3 | 0.753±0.000 | 0.842±0.000 | 1.221±0.012 | 1.806±0.005 | 4.75 | 4 |
| | LP | 1.274±0.011 | 1.036±0.004 | 1.216±0.002 | 2.285±0.004 | 8.00 | 8 | 1.007±0.000 | 0.972±0.000 | 1.223±0.000 | 2.117±0.000 | 7.25 | 8 |
| | SURGICAL-FT | 0.961±0.013 | 0.888±0.005 | 1.201±0.005 | 1.962±0.009 | 5.75 | 6 | 0.748±0.000 | **0.825±0.000** | 1.210±0.000 | 1.795±0.000 | 3.00 | 2 |
| | LP-FT | 0.884±0.035 | 0.842±0.013 | 1.215±0.002 | 1.904±0.011 | 4.75 | 5 | **0.697±0.000** | 0.835±0.016 | **1.210±0.008** | **1.794±0.004** | 2.00 | 3 |
| | WiSE-FT | 0.995±0.010 | 0.855±0.011 | 1.193±0.003 | 1.893±0.021 | 4.00 | 4 | 0.742±0.000 | 0.852±0.001 | 1.228±0.004 | 1.809±0.006 | 5.25 | 5 |
| | L2-SP | **0.878±0.026** | **0.806±0.007** | **1.192±0.004** | 1.893±0.018 | 1.75 | 1 | 0.741±0.029 | 0.907±0.020 | 1.243±0.006 | 1.822±0.003 | 6.00 | 7 |
| | FEATURE-MAP | 1.057±0.008 | 0.894±0.009 | 1.196±0.002 | 2.019±0.004 | 6.50 | 7 | 0.706±0.005 | 0.840±0.013 | **1.200±0.014** | **1.773±0.008** | 1.75 | 1 |
| | BSS | 0.886±0.010 | 0.809±0.005 | 1.194±0.006 | **1.862±0.010** | 2.75 | 2 | 0.715±0.024 | 0.892±0.014 | 1.248±0.006 | 1.824±0.006 | 6.00 | 6 |
| SCAFFOLD | FULL-FT | 1.196±0.013 | 0.819±0.009 | 1.137±0.016 | 1.892±0.017 | 4.25 | 4 | 0.956±0.000 | 0.888±0.011 | 1.149±0.014 | **1.787±0.020** | 4.50 | 5 |
| | LP | 1.867±0.006 | 0.937±0.004 | 1.140±0.002 | 2.338±0.005 | 7.75 | 8 | 1.006±0.000 | 0.921±0.000 | 1.162±0.000 | 2.183±0.000 | 8.00 | 8 |
| | SURGICAL-FT | 1.221±0.011 | 0.883±0.010 | 1.130±0.005 | 1.953±0.007 | 5.75 | 6 | 0.955±0.000 | 0.887±0.000 | 1.138±0.000 | **1.787±0.000** | 3.75 | 3 |
| | LP-FT | **1.112±0.015** | **0.802±0.003** | 1.153±0.005 | 1.895±0.013 | 3.50 | 5 | **0.951±0.000** | 0.889±0.025 | 1.143±0.000 | 1.791±0.008 | 3.50 | 4 |
| | WiSE-FT | 1.388±0.023 | 0.834±0.012 | **1.114±0.002** | 1.936±0.037 | 4.25 | 3 | **0.947±0.000** | 0.893±0.007 | 1.134±0.011 | 1.800±0.006 | 4.00 | 2 |
| | L2-SP | 1.163±0.026 | 0.813±0.010 | 1.126±0.011 | 1.885±0.011 | 2.50 | 2 | 0.991±0.018 | 0.878±0.012 | 1.128±0.002 | 2.017±0.179 | 4.50 | 7 |
| | FEATURE-MAP | 1.495±0.016 | 0.863±0.005 | 1.118±0.001 | 2.008±0.010 | 5.50 | 7 | 0.966±0.014 | **0.826±0.017** | 1.136±0.003 | 1.792±0.011 | 3.50 | 1 |
| | BSS | 1.188±0.026 | 0.814±0.007 | 1.123±0.005 | **1.881±0.014** | 2.50 | 1 | 0.977±0.021 | 0.885±0.014 | **1.126±0.007** | 1.949±0.127 | 4.25 | 6 |
| SIZE | FULL-FT | 1.692±0.070 | 0.838±0.023 | 0.922±0.013 | **2.364±0.030** | 4.00 | 3 | 1.115±0.019 | 0.848±0.038 | 0.915±0.000 | 2.230±0.009 | 5.25 | 5 |
| | LP | 2.290±0.017 | 1.039±0.005 | 0.908±0.022 | 2.749±0.018 | 6.75 | 8 | **1.073±0.000** | 0.871±0.000 | 0.904±0.000 | 2.435±0.000 | 5.25 | 8 |
| | SURGICAL-FT | 1.928±0.039 | 0.895±0.007 | 0.919±0.007 | 2.397±0.014 | 5.50 | 6 | 1.094±0.000 | **0.807±0.000** | 0.904±0.000 | **2.200±0.000** | 2.75 | 1 |
| | LP-FT | 1.674±0.030 | **0.803±0.006** | 0.954±0.011 | **2.328±0.017** | 3.25 | 5 | 1.081±0.024 | 0.842±0.021 | 0.925±0.000 | 2.280±0.000 | 5.25 | 7 |
| | WiSE-FT | 2.071±0.078 | 0.902±0.016 | 0.912±0.003 | 2.379±0.086 | 5.75 | 7 | 1.116±0.023 | **0.805±0.015** | 0.907±0.001 | 2.228±0.010 | 4.00 | 2 |
| | L2-SP | **1.629±0.084** | 0.821±0.011 | 0.904±0.003 | 2.368±0.013 | 2.50 | 1 | 1.183±0.055 | 0.853±0.031 | 0.903±0.004 | 2.227±0.038 | 5.00 | 6 |
| | FEATURE-MAP | 1.963±0.035 | 0.910±0.009 | **0.895±0.002** | 2.366±0.006 | 4.25 | 4 | 1.193±0.058 | 0.850±0.021 | **0.901±0.025** | **2.203±0.023** | 4.50 | 4 |
| | BSS | 1.630±0.035 | 0.818±0.005 | 0.925±0.019 | 2.370±0.013 | 4.00 | 2 | 1.142±0.049 | 0.834±0.018 | **0.900±0.003** | 2.245±0.027 | 4.00 | 3 |

Table 8: Robust fine-tuning performance on 8 **Classification** datasets (AUC metrics) in the **Non-Fewshot** setting, evaluated across 3 dataset splits (RANDOM, SCAFFOLD, SIZE), over **MOLECULESTM** and **GRAPHIUM-LARGE** models. AVG, AVG-F, AVG-R denote the average AUC, average AUC without max and min values, and average rank over all the datasets for each method, respectively. Standard deviations across five replicates are shown in parentheses. We **bold** and underline the best and second-best performances in each scenario.

| SPLIT | METHODS | CLINTOX | BBBP | BACE | HIV | MUV | SIDER | TOX21 | TOXCAST | AVG | AVG-F | AVG-R |
|---|---|---|---|---|---|---|---|---|---|---|---|---|
| | | | | | **SELF-SUPERVISED PRE-TRAINING (MOLECULESTM)** | | | | | | | |
| RANDOM | FULL-FT | 89.90 ± 1.49 | 93.43 ± 0.99 | 89.82 ± 1.08 | 84.72 ± 1.11 | 77.82 ± 3.46 | 62.12 ± 1.15 | 82.49 ± 0.41 | 72.95 ± 0.31 | 81.66 | 82.95 | 3.62 |
| | LP | 74.32 ± 1.90 | 84.76 ± 0.29 | 74.85 ± 0.27 | 74.15 ± 0.69 | 76.86 ± 1.07 | 59.69 ± 0.24 | 73.72 ± 0.20 | 66.19 ± 0.14 | 73.07 | 73.35 | 7.75 |
| | SURGICAL-FT | 86.04 ± 0.89 | 93.68 ± 0.51 | 89.99 ± 0.46 | **85.68 ± 0.84** | 79.59 ± 2.47 | **63.64 ± 0.78** | 81.84 ± 0.66 | 71.83 ± 0.55 | 81.54 | 82.50 | 3.38 |
| | LP-FT | 86.39 ± 1.85 | 93.72 ± 0.93 | 89.82 ± 0.57 | 84.17 ± 1.41 | 76.87 ± 2.38 | 62.19 ± 1.00 | 82.54 ± 0.51 | 72.19 ± 0.52 | 80.99 | 82.00 | 3.75 |
| | WISE-FT | **90.35 ± 1.26** | 92.93 ± 0.80 | **90.41 ± 0.86** | 84.38 ± 1.05 | 77.23 ± 3.08 | 62.17 ± 1.25 | 82.67 ± 0.32 | 73.08 ± 0.32 | 81.65 | 83.02 | 2.88 |
| | $L^2$-SP | 89.69 ± 1.39 | 93.77 ± 0.37 | 89.21 ± 0.92 | 81.94 ± 1.20 | 50.21 ± 4.41 | 61.07 ± 1.22 | 82.97 ± 0.39 | 71.02 ± 0.57 | 77.48 | 79.32 | 5.00 |
| | FEATURE-MAP | 79.93 ± 1.54 | 90.59 ± 0.39 | 83.69 ± 0.24 | 77.66 ± 0.46 | **80.03 ± 1.01** | 59.93 ± 0.14 | 75.32 ± 0.19 | 67.51 ± 0.30 | 76.83 | 77.36 | 6.25 |
| | BSS | 90.17 ± 2.84 | **94.16 ± 0.55** | 89.74 ± 1.12 | 83.96 ± 1.29 | 76.64 ± 1.29 | 61.87 ± 0.69 | **83.26 ± 0.57** | **74.55 ± 0.31** | **81.79** | **83.05** | 3.38 |
| SCAFFOLD | FULL-FT | 74.94 ± 7.23 | 68.62 ± 0.80 | 75.35 ± 2.06 | 76.03 ± 0.91 | 73.43 ± 2.50 | 57.88 ± 1.18 | 76.67 ± 0.68 | 63.62 ± 0.27 | 70.82 | 72.00 | 4.25 |
| | LP | 65.07 ± 1.08 | 59.39 ± 0.35 | 69.24 ± 0.16 | 69.97 ± 0.57 | 71.81 ± 2.40 | 59.93 ± 0.37 | 69.87 ± 0.28 | 60.05 ± 0.25 | 65.67 | 65.69 | 7.00 |
| | SURGICAL-FT | 71.07 ± 4.16 | 67.78 ± 0.60 | 80.16 ± 2.36 | **76.80 ± 1.06** | 75.87 ± 0.82 | 59.24 ± 1.22 | 75.54 ± 0.64 | 63.27 ± 0.70 | 71.22 | 71.72 | 3.75 |
| | LP-FT | 75.07 ± 2.24 | 67.05 ± 1.42 | 75.33 ± 1.14 | 76.68 ± 0.82 | 71.36 ± 1.39 | 58.51 ± 1.15 | 76.85 ± 0.63 | 62.98 ± 0.51 | 70.48 | 71.41 | 4.62 |
| | WISE-FT | **77.27 ± 4.28** | 68.72 ± 0.75 | 77.37 ± 1.44 | 75.91 ± 0.74 | 74.38 ± 2.20 | 58.19 ± 1.26 | **76.89 ± 0.69** | **64.05 ± 0.34** | 71.60 | **72.87** | 3.12 |
| | $L^2$-SP | 74.62 ± 4.99 | 68.30 ± 1.19 | 79.91 ± 2.29 | 73.97 ± 0.78 | 61.62 ± 2.07 | 59.78 ± 0.33 | 75.39 ± 0.51 | 62.34 ± 0.82 | 69.49 | 69.37 | 5.25 |
| | FEATURE-MAP | 61.06 ± 2.00 | 65.12 ± 1.98 | **82.66 ± 0.62** | 74.54 ± 1.00 | 72.81 ± 1.16 | **60.47 ± 0.45** | 70.39 ± 0.11 | 60.10 ± 0.19 | 68.39 | 67.40 | 5.25 |
| | BSS | 73.89 ± 6.04 | **70.04 ± 2.00** | 77.94 ± 2.04 | 76.28 ± 1.28 | **76.20 ± 1.33** | 59.99 ± 1.39 | 75.86 ± 1.08 | 63.62 ± 0.50 | **71.73** | 72.65 | 2.75 |
| SIZE | FULL-FT | 61.94 ± 2.67 | 82.80 ± 2.31 | 63.62 ± 1.19 | 77.81 ± 2.99 | 72.05 ± 2.96 | 54.92 ± 0.79 | 71.08 ± 0.77 | 62.47 ± 0.83 | 68.34 | 68.16 | 5.12 |
| | LP | 55.54 ± 0.65 | 75.89 ± 0.90 | 42.31 ± 0.48 | 67.54 ± 1.27 | 69.87 ± 1.51 | 53.74 ± 0.43 | 68.10 ± 0.39 | 57.50 ± 0.19 | 61.31 | 62.05 | 7.75 |
| | SURGICAL-FT | 64.54 ± 8.03 | **88.90 ± 0.74** | 61.99 ± 2.13 | 78.10 ± 0.96 | **76.07 ± 0.57** | **57.13 ± 1.87** | **72.24 ± 0.28** | 60.52 ± 0.95 | **69.94** | 68.91 | 2.50 |
| | LP-FT | 63.99 ± 3.29 | 83.12 ± 5.20 | **65.48 ± 0.70** | 76.47 ± 3.53 | 72.24 ± 2.79 | 56.31 ± 0.72 | 71.60 ± 0.63 | 62.70 ± 0.87 | 68.97 | 68.72 | 3.75 |
| | WISE-FT | 63.85 ± 3.69 | 81.81 ± 2.80 | 62.71 ± 1.26 | 77.83 ± 2.02 | 73.40 ± 2.08 | 56.63 ± 0.63 | 71.27 ± 0.77 | 62.70 ± 0.87 | 68.78 | 68.63 | 4.00 |
| | $L^2$-SP | 63.67 ± 1.79 | 88.00 ± 1.00 | 63.98 ± 1.51 | 77.38 ± 1.25 | 58.29 ± 3.74 | 56.23 ± 1.70 | 71.93 ± 0.21 | 59.29 ± 0.72 | 67.35 | 65.76 | 4.50 |
| | FEATURE-MAP | 64.41 ± 1.38 | 86.82 ± 0.76 | 59.62 ± 1.17 | 70.71 ± 0.99 | 76.01 ± 0.60 | 55.03 ± 0.30 | 67.98 ± 0.41 | 57.91 ± 0.31 | 67.31 | 66.11 | 5.25 |
| | BSS | **67.80 ± 4.60** | 84.90 ± 2.20 | 62.77 ± 3.69 | **78.13 ± 2.21** | 74.58 ± 1.13 | 54.91 ± 1.34 | 71.40 ± 0.44 | **63.04 ± 0.35** | 69.69 | **69.62** | 3.12 |
| | | | | | | **SUPERVISED PRE-TRAINING (GRAPHIUM-LARGE)** | | | | | | | |
| SCAFFOLD | FULL-FT | 81.27 ± 3.88 | 69.17 ± 1.32 | 79.75 ± 1.07 | 76.42 ± 0.72 | 76.84 ± 1.80 | 63.63 ± 0.06 | 78.12 ± 0.46 | 66.37 ± 0.26 | 73.95 | 74.45 | 3.75 |
| | LP | 80.48 ± 0.00 | 66.90 ± 0.00 | 80.44 ± 0.00 | 75.83 ± 0.00 | 73.35 ± 0.00 | 62.03 ± 0.00 | 79.02 ± 0.00 | 66.09 ± 0.00 | 73.02 | 73.61 | 5.12 |
| | SURGICAL-FT | 86.17 ± 0.00 | **73.71 ± 0.00** | **84.16 ± 0.00** | **77.47 ± 0.00** | **78.87 ± 0.00** | **64.02 ± 0.00** | 78.23 ± 0.00 | **67.34 ± 0.00** | **76.25** | **76.63** | 1.38 |
| | LP-FT | 83.67 ± 3.53 | 69.98 ± 0.83 | 79.28 ± 0.32 | 76.17 ± 2.01 | 77.82 ± 1.15 | 61.20 ± 0.00 | 76.94 ± 0.00 | 66.28 ± 0.00 | 73.92 | 74.41 | 4.62 |
| | WISE-FT | 85.40 ± 1.61 | 71.89 ± 1.79 | 78.13 ± 2.92 | 76.69 ± 1.76 | 74.37 ± 1.79 | 63.58 ± 0.00 | 77.98 ± 0.33 | 66.48 ± 0.43 | 74.31 | 74.26 | 3.62 |
| | $L^2$-SP | 76.83 ± 8.87 | 67.35 ± 0.82 | 78.17 ± 0.02 | 73.69 ± 0.03 | 62.35 ± 0.15 | 62.21 ± 0.45 | 76.27 ± 0.32 | 62.75 ± 0.88 | 69.95 | 69.87 | 6.62 |
| | FEATURE-MAP | **90.13 ± 2.12** | 70.99 ± 0.27 | 83.17 ± 0.49 | 73.61 ± 0.03 | 78.74 ± 0.76 | 62.12 ± 0.02 | **79.99 ± 0.12** | 65.03 ± 0.08 | 75.47 | 75.25 | 3.50 |
| | BSS | 79.99 ± 5.89 | 67.10 ± 0.93 | 78.12 ± 2.32 | 72.50 ± 0.51 | 61.20 ± 0.08 | 61.13 ± 0.95 | 76.69 ± 0.64 | 65.45 ± 0.89 | 70.27 | 70.18 | 7.38 |
| SIZE | FULL-FT | 85.96 ± 4.28 | 87.62 ± 0.90 | 67.41 ± 2.44 | 81.47 ± 1.94 | 72.03 ± 2.55 | 54.72 ± 0.01 | 69.71 ± 0.37 | 61.31 ± 0.37 | 72.53 | 72.98 | 3.88 |
| | LP | 81.84 ± 0.02 | 78.09 ± 0.00 | 58.08 ± 0.01 | 77.48 ± 0.00 | 69.46 ± 0.00 | 53.59 ± 0.00 | 73.65 ± 0.00 | 61.25 ± 0.00 | 69.18 | 69.67 | 5.38 |
| | SURGICAL-FT | 86.59 ± 0.01 | **89.07 ± 0.00** | 70.94 ± 0.01 | 82.50 ± 0.00 | **74.47 ± 0.00** | **56.24 ± 0.00** | 72.30 ± 0.00 | **62.74 ± 0.00** | **74.36** | **74.92** | 1.62 |
| | LP-FT | **86.78 ± 2.69** | 88.02 ± 1.50 | 63.72 ± 1.85 | **82.57 ± 0.46** | 73.51 ± 1.77 | 52.40 ± 0.00 | 68.23 ± 0.87 | 60.85 ± 0.00 | 72.01 | 72.61 | 4.00 |
| | WISE-FT | 82.44 ± 3.02 | 87.76 ± 0.5 | **72.89 ± 0.66** | 81.37 ± 1.07 | 73.67 ± 3.44 | 55.87 ± 0.01 | 68.85 ± 0.84 | 60.61 ± 0.53 | 72.93 | 73.31 | 3.62 |
| | $L^2$-SP | 71.03 ± 3.67 | 81.32 ± 1.51 | 68.82 ± 0.06 | 70.66 ± 0.00 | 64.69 ± 0.32 | 52.08 ± 0.84 | 70.91 ± 0.34 | 56.50 ± 0.01 | 67.00 | 67.10 | 6.88 |
| | FEATURE-MAP | 82.48 ± 3.25 | 87.70 ± 0.64 | 69.56 ± 0.20 | 67.23 ± 1.93 | 71.49 ± 0.13 | 54.43 ± 0.03 | **74.12 ± 0.09** | 58.73 ± 0.04 | 70.72 | 70.60 | 4.38 |
| | BSS | 72.42 ± 0.03 | 82.92 ± 1.60 | 62.76 ± 4.23 | 72.81 ± 0.66 | 65.79 ± 5.31 | 52.89 ± 1.12 | 71.91 ± 0.44 | 57.79 ± 1.80 | 67.41 | 67.25 | 6.25 |

Table 9: Robust fine-tuning performance on 4 **Regression** datasets (RMSE metrics) in the **Non-Fewshot** setting, evaluated across 3 dataset splits (RANDOM, SCAFFOLD, SIZE) over **MOLECULESTM** and **GRAPHIUM-LARGE** models. AVG-R, AVG-R* denote the average rank and the rank based on the average normalized performance over all the datasets for each method, respectively. Standard deviations across five replicates are shown in parentheses. We **bold** and underline the best and second-best performances in each scenario.

| SPLIT | METHODS | SELF-SUPERVISED PRE-TRAINING (MOLECULESTM) | | | | | | SUPERVISED PRE-TRAINING (GRAPHIUM-LARGE) | | | | | |
|---|---|---|---|---|---|---|---|---|---|---|---|---|---|
| | | ESOL | LIPO | MALARIA | CEP | AVG-R | AVG-R* | ESOL | LIPO | MALARIA | CEP | AVG-R | AVG-R* |
| RANDOM | FULL-FT | 0.901 ± 0.063 | 0.660 ± 0.013 | 1.067 ± 0.009 | 1.401 ± 0.035 | 3.00 | 2 | 0.643 ± 0.011 | 0.605 ± 0.011 | 1.085 ± 0.007 | 1.399 ± 0.015 | 4.00 | 4 |
| | LP | 1.374 ± 0.011 | 1.067 ± 0.015 | 1.207 ± 0.004 | 1.999 ± 0.003 | 8.00 | 8 | 0.699 ± 0.000 | 0.672 ± 0.000 | 1.105 ± 0.002 | 1.658 ± 0.011 | 7.75 | 8 |
| | SURGICAL-FT | 1.056 ± 0.028 | 0.724 ± 0.011 | 1.074 ± 0.010 | 1.547 ± 0.011 | 6.00 | 6 | **0.617 ± 0.000** | **0.582 ± 0.000** | **1.047 ± 0.000** | 1.392 ± 0.000 | 1.25 | 1 |
| | LP-FT | 0.922 ± 0.023 | 0.654 ± 0.023 | 1.076 ± 0.014 | 1.365 ± 0.029 | 3.25 | 2 | 0.618 ± 0.023 | 0.591 ± 0.000 | 1.059 ± 0.000 | **1.355 ± 0.008** | 2.00 | 2 |
| | WISE-FT | 0.934 ± 0.061 | 0.662 ± 0.016 | **1.064 ± 0.007** | 1.460 ± 0.042 | 3.75 | 5 | 0.630 ± 0.006 | 0.606 ± 0.008 | 1.086 ± 0.007 | 1.430 ± 0.019 | 5.00 | 3 |
| | $L^2$-SP | **0.884 ± 0.025** | 0.666 ± 0.014 | 1.087 ± 0.011 | 1.385 ± 0.031 | 3.75 | 4 | 0.647 ± 0.028 | 0.602 ± 0.014 | 1.059 ± 0.001 | 1.466 ± 0.050 | 5.75 | 7 |
| | FEATURE-MAP | 1.018 ± 0.024 | 0.789 ± 0.018 | 1.106 ± 0.005 | 1.536 ± 0.008 | 6.50 | 7 | 0.660 ± 0.240 | 0.642 ± 0.009 | 1.059 ± 0.009 | 1.419 ± 0.037 | 5.25 | 5 |
| | BSS | 0.887 ± 0.030 | **0.641 ± 0.014** | 1.070 ± 0.016 | **1.351 ± 0.016** | 1.75 | 1 | 0.619 ± 0.030 | 0.611 ± 0.017 | 1.158 ± 0.041 | 1.404 ± 0.029 | 5.00 | 6 |
| SCAFFOLD | FULL-FT | 1.360 ± 0.049 | 0.752 ± 0.018 | 1.105 ± 0.018 | 1.395 ± 0.041 | 4.50 | 5 | 0.878 ± 0.011 | 0.731 ± 0.003 | 1.107 ± 0.008 | 1.409 ± 0.035 | 4.50 | 5 |
| | LP | 1.608 ± 0.030 | 0.983 ± 0.006 | 1.133 ± 0.002 | 2.009 ± 0.004 | 8.00 | 8 | 0.886 ± 0.005 | 0.772 ± 0.000 | 1.103 ± 0.000 | 1.635 ± 0.017 | 6.25 | 7 |
| | SURGICAL-FT | **1.297 ± 0.044** | 0.765 ± 0.013 | 1.105 ± 0.013 | 1.518 ± 0.010 | 4.50 | 6 | 0.863 ± 0.000 | **0.675 ± 0.000** | 1.090 ± 0.000 | 1.480 ± 0.000 | 2.75 | 3 |
| | LP-FT | 1.331 ± 0.033 | 0.743 ± 0.017 | 1.107 ± 0.011 | 1.356 ± 0.030 | 4.00 | 4 | 0.887 ± 0.002 | 0.709 ± 0.016 | 1.091 ± 0.007 | **1.380 ± 0.005** | 3.75 | 4 |
| | WISE-FT | 1.347 ± 0.036 | **0.740 ± 0.018** | **1.090 ± 0.015** | 1.505 ± 0.045 | 3.00 | 2 | 0.876 ± 0.011 | 0.727 ± 0.004 | 1.120 ± 0.008 | 1.430 ± 0.041 | 4.75 | 6 |
| | $L^2$-SP | 1.300 ± 0.017 | 0.756 ± 0.017 | 1.106 ± 0.005 | 1.347 ± 0.020 | 3.75 | 3 | 0.905 ± 0.022 | 0.778 ± 0.009 | 1.147 ± 0.003 | 1.518 ± 0.011 | 7.50 | 8 |
| | FEATURE-MAP | 1.383 ± 0.008 | 0.824 ± 0.009 | 1.098 ± 0.004 | 1.518 ± 0.003 | 6.00 | 7 | **0.853 ± 0.005** | 0.692 ± 0.002 | 1.149 ± 0.002 | 1.427 ± 0.052 | 3.50 | 2 |
| | BSS | 1.300 ± 0.024 | 0.746 ± 0.010 | 1.097 ± 0.013 | **1.319 ± 0.023** | 2.25 | 1 | 0.873 ± 0.024 | 0.707 ± 0.015 | **0.166 ± 0.000** | 1.431 ± 0.016 | 3.00 | 1 |
| SIZE | FULL-FT | 1.490 ± 0.153 | 0.711 ± 0.017 | **0.883 ± 0.008** | 1.834 ± 0.038 | 3.25 | 2 | 1.020 ± 0.009 | 0.727 ± 0.006 | 0.890 ± 0.013 | **1.847 ± 0.043** | 2.75 | 2 |
| | LP | 2.172 ± 0.065 | 0.935 ± 0.004 | 0.912 ± 0.004 | 2.402 ± 0.018 | 8.00 | 8 | 1.190 ± 0.000 | 0.852 ± 0.000 | 0.912 ± 0.000 | 2.101 ± 0.026 | 7.75 | 8 |
| | SURGICAL-FT | 1.499 ± 0.093 | 0.769 ± 0.013 | 0.889 ± 0.014 | 1.998 ± 0.020 | 5.25 | 6 | 1.105 ± 0.000 | 0.745 ± 0.000 | **0.871 ± 0.000** | 1.902 ± 0.000 | 4.50 | 5 |
| | LP-FT | 1.401 ± 0.053 | 0.703 ± 0.012 | 0.897 ± 0.009 | 1.763 ± 0.037 | 3.25 | 3 | 1.067 ± 0.034 | **0.703 ± 0.016** | 0.892 ± 0.014 | 1.884 ± 0.017 | 3.75 | 4 |
| | WISE-FT | 1.583 ± 0.118 | 0.727 ± 0.018 | 0.889 ± 0.008 | 1.902 ± 0.053 | 5.25 | 5 | 1.026 ± 0.011 | 0.721 ± 0.009 | 0.888 ± 0.011 | 1.848 ± 0.035 | 2.75 | 1 |
| | $L^2$-SP | **1.390 ± 0.115** | 0.725 ± 0.019 | 0.896 ± 0.007 | 1.786 ± 0.022 | 3.25 | 4 | 1.001 ± 0.023 | 0.805 ± 0.010 | 0.903 ± 0.004 | 2.008 ± 0.130 | 5.25 | 6 |
| | FEATURE-MAP | 1.458 ± 0.045 | 0.849 ± 0.012 | 0.896 ± 0.011 | 2.007 ± 0.018 | 6.00 | 7 | 1.028 ± 0.032 | 0.767 ± 0.002 | 0.925 ± 0.002 | 2.079 ± 0.021 | 6.50 | 7 |
| | BSS | 1.408 ± 0.100 | **0.700 ± 0.020** | 0.887 ± 0.011 | **1.725 ± 0.026** | 1.75 | 1 | **0.985 ± 0.040** | 0.720 ± 0.013 | 0.901 ± 0.006 | 1.883 ± 0.023 | 2.75 | 3 |

Table 10: Robust fine-tuning performance on 5 **Classification** datasets (AUC metrics) in the **Fewshot** setting (covering FEWSHOT-50, FEWSHOT-100, FEWSHOT-500), evaluated across 3 dataset splits (RANDOM, SCAFFOLD, SIZE) over **MOLECULESTM** and **GRAPHIUM-LARGE** models. We **bold** and underline the best and second-best performances in each scenario.

| SPLIT | METHODS | SELF-SUPERVISED PRE-TRAINING (MOLECULESTM) | | | | | | | | SUPERVISED PRE-TRAINING (GRAPHIUM-LARGE) | | | | | | | |
|---|---|---|---|---|---|---|---|---|---|---|---|---|---|---|---|---|---|
| | | ClinTox | BBBP | Bace | HIV | Sider | AVG | AVG-F | AVG-R | ClinTox | BBBP | Bace | HIV | Sider | AVG | AVG-F | AVG-R |
| **FEWSHOT-50** | | | | | | | | | | | | | | | | | |
| RANDOM | Full-FT | 49.60±2.85 | 84.86±1.30 | 74.74±1.44 | 60.58±1.47 | 49.47±0.90 | 63.85 | 61.64 | 4.80 | 74.25±0.00 | 82.09±0.77 | 81.04±0.00 | 62.83±0.00 | 52.55±0.00 | 70.55 | 72.71 | 6.00 |
| | LP | 52.66±3.14 | 78.85±1.75 | 58.02±3.19 | 52.39±0.52 | 50.23±0.47 | 58.43 | 54.36 | 6.40 | 64.37±0.00 | 86.23±0.00 | 81.47±0.00 | 60.00±0.00 | 54.28±0.00 | 69.27 | 68.61 | 5.00 |
| | Surgical-FT | 54.43±4.39 | 86.64±0.96 | 74.92±0.95 | 61.71±0.64 | 51.10±0.82 | 65.76 | 63.69 | 2.00 | 76.03±0.00 | 87.04±0.00 | 82.44±0.00 | 62.09±0.00 | 53.09±0.00 | 72.14 | 73.52 | 3.60 |
| | LP-FT | 47.71±2.16 | 84.36±2.65 | 74.92±0.95 | 55.82±1.53 | 51.62±0.37 | 62.89 | 60.79 | 4.60 | 76.40±0.00 | 82.10±0.00 | 73.86±0.00 | 64.86±0.00 | 54.11±0.00 | 70.27 | 71.71 | 4.20 |
| | WiSE-FT | 55.69±5.37 | 84.62±1.45 | 74.02±1.36 | 60.05±1.26 | 49.41±0.89 | 64.76 | 63.25 | 4.60 | 75.77±0.00 | 84.05±0.85 | 81.30±0.00 | 62.46±0.00 | 55.49±0.00 | 71.81 | 73.18 | 3.60 |
| | $L^2$-SP | 50.07±2.37 | 85.69±1.19 | 75.18±1.16 | 58.44±1.98 | 50.58±0.93 | 63.99 | 61.40 | 3.60 | 75.31±2.24 | 84.45±4.02 | 80.56±0.00 | 69.92±0.00 | 53.87±0.00 | 72.82 | 75.26 | 4.60 |
| | Feature-map | 54.09±3.21 | 78.77±4.05 | 67.88±0.54 | 55.43±1.21 | 50.12±0.27 | 61.26 | 59.13 | 6.20 | 71.01±0.00 | 88.81±0.00 | 81.76±0.03 | 61.15±0.00 | 54.47±0.15 | 71.44 | 71.31 | 3.80 |
| | BSS | 52.06±3.58 | 85.62±1.18 | 74.31±1.83 | 58.90±0.76 | 51.18±0.69 | 64.41 | 61.76 | 3.80 | 75.33±0.00 | 81.30±1.08 | 80.98±0.00 | 64.67±0.00 | 53.88±0.51 | 71.23 | 73.66 | 5.20 |
| SCAFFOLD | Full-FT | 45.62±5.48 | 58.05±2.70 | 62.30±1.27 | 48.87±6.91 | 54.88±0.29 | 53.94 | 53.93 | 2.60 | 74.79±0.00 | 61.10±0.00 | 74.43±0.00 | 64.93±0.00 | 54.35±0.00 | 65.92 | 66.82 | 5.60 |
| | LP | 30.76±1.34 | 50.50±1.35 | 56.94±2.34 | 39.19±1.21 | 53.17±0.36 | 46.11 | 47.62 | 7.80 | 67.24±0.00 | 64.31±0.00 | 65.24±0.00 | 50.89±0.00 | 55.24±0.00 | 60.58 | 61.60 | 5.60 |
| | Surgical-FT | 45.60±9.96 | 56.02±1.54 | 63.07±0.78 | 44.00±3.78 | 55.16±0.47 | 52.77 | 52.27 | 3.80 | 71.74±0.00 | 62.43±0.00 | 71.64±0.00 | 65.60±0.00 | 55.55±0.00 | 65.99 | 66.59 | 4.00 |
| | LP-FT | 33.97±3.65 | 55.31±2.06 | 61.87±0.80 | 45.88±1.92 | 55.16±0.46 | 50.44 | 52.12 | 5.20 | 61.66±0.00 | 63.99±0.00 | 76.82±0.00 | 56.11±0.00 | 56.50±0.00 | 62.90 | 60.52 | 4.60 |
| | WiSE-FT | 47.69±5.22 | 57.80±2.92 | 62.06±1.03 | 47.33±5.84 | 55.16±0.57 | 54.01 | 53.55 | 2.60 | 73.93±0.00 | 65.16±0.00 | 74.82±0.00 | 64.36±0.00 | 54.92±0.00 | 66.64 | 67.82 | 3.60 |
| | $L^2$-SP | 45.54±5.40 | 56.06±1.99 | 61.75±1.66 | 45.56±4.10 | 55.29±0.92 | 52.84 | 52.30 | 4.20 | 68.43±0.00 | 64.01±0.93 | 74.63±0.00 | 66.45±0.00 | 56.54±0.00 | 66.01 | 66.30 | 3.20 |
| | Feature-map | 26.69±2.38 | 56.71±1.18 | 61.18±5.30 | 43.71±3.23 | 53.77±0.39 | 48.41 | 51.40 | 6.60 | 65.60±0.03 | 63.73±0.00 | 70.32±0.00 | 70.97±0.00 | 54.72±0.03 | 65.07 | 66.55 | 5.20 |
| | BSS | 42.19±1.78 | 57.09±1.32 | 63.74±2.79 | 50.07±8.79 | 54.75±0.37 | 53.57 | 53.97 | 3.20 | 77.89±0.04 | 61.79±0.00 | 74.27±1.63 | 66.56±0.00 | 55.03±0.01 | 67.11 | 67.54 | 4.20 |
| SIZE | Full-FT | 58.52±2.98 | 58.80±9.95 | 36.17±6.29 | 52.04±2.74 | 51.97±1.34 | 51.50 | 54.18 | 4.20 | 71.15±0.00 | 80.00±0.00 | 59.96±3.09 | 48.05±0.00 | 53.20±0.00 | 62.47 | 61.44 | 4.60 |
| | LP | 57.53±4.82 | 45.54±17.14 | 47.39±1.62 | 48.21±0.61 | 50.89±0.73 | 49.91 | 48.83 | 6.60 | 62.05±0.00 | 72.11±0.00 | 56.89±0.01 | 57.63±0.00 | 49.15±0.00 | 59.57 | 58.86 | 7.20 |
| | Surgical-FT | 61.32±8.19 | 54.19±11.51 | 44.96±7.70 | 51.79±2.35 | 51.41±0.98 | 52.73 | 52.46 | 4.80 | 71.68±0.00 | 83.99±0.00 | 62.17±0.00 | 62.00±0.00 | 54.99±0.00 | 66.97 | 65.28 | 2.40 |
| | LP-FT | 54.70±9.04 | 55.56±3.73 | 43.08±1.91 | 47.90±2.39 | 51.88±0.55 | 50.62 | 51.49 | 5.80 | 70.62±0.00 | 79.53±0.00 | 59.30±0.00 | 58.07±0.00 | 52.10±0.00 | 63.92 | 62.66 | 5.00 |
| | WiSE-FT | 61.60±5.18 | 56.83±9.47 | 42.48±6.40 | 50.61±2.77 | 52.28±1.23 | 52.76 | 53.24 | 3.80 | 70.51±0.00 | 78.10±0.00 | 59.48±3.21 | 54.15±0.00 | 53.24±0.00 | 63.10 | 61.38 | 5.20 |
| | $L^2$-SP | 60.53±2.27 | 62.77±6.52 | 47.51±8.30 | 52.06±2.80 | 51.52±1.67 | 54.88 | 54.71 | 2.60 | 65.70±0.03 | 85.88±0.76 | 56.81±0.04 | 62.79±0.06 | 57.10±0.00 | 65.66 | 61.86 | 3.80 |
| | Feature-map | 59.85±1.06 | 50.21±1.87 | 47.65±3.15 | 44.09±1.27 | 51.48±0.50 | 50.66 | 49.78 | 5.40 | 69.15±0.01 | 65.65±0.01 | 61.95±0.58 | 64.82±0.03 | 50.81±0.01 | 66.48 | 65.31 | 3.60 |
| | BSS | 62.26±1.89 | 60.79±7.04 | 49.70±2.37 | 51.85±3.42 | 51.19±1.56 | 55.16 | 54.61 | 4.20 | 73.63±0.01 | 79.93±2.44 | 56.91±3.73 | 52.67±1.33 | 56.22±0.72 | 63.87 | 62.25 | 4.20 |
| **FEWSHOT-100** | | | | | | | | | | | | | | | | | |
| RANDOM | Full-FT | 73.60±7.53 | 82.09±2.90 | 80.72±1.22 | 61.92±2.62 | 51.58±0.43 | 69.98 | 72.08 | 5.00 | 66.36±0.01 | 86.40±2.10 | 78.44±0.00 | 63.35±0.00 | 56.74±0.00 | 70.26 | 69.38 | 6.20 |
| | LP | 69.43±1.40 | 73.63±0.97 | 60.60±3.89 | 54.74±0.90 | 53.47±0.21 | 62.37 | 61.59 | 6.60 | 65.67±0.00 | 90.26±0.00 | 81.88±0.00 | 61.87±0.00 | 57.00±0.00 | 71.34 | 69.81 | 5.20 |
| | Surgical-FT | 71.20±2.70 | 83.50±0.95 | 80.44±0.62 | 62.65±1.14 | 53.43±0.90 | 70.24 | 71.43 | 4.20 | 71.48±0.00 | 86.23±0.00 | 85.03±0.00 | 63.49±0.00 | 58.53±0.00 | 72.95 | 73.33 | 3.20 |
| | LP-FT | 68.16±1.86 | 84.26±1.37 | 79.93±2.67 | 60.14±3.04 | 52.18±0.81 | 68.93 | 69.41 | 5.20 | 70.77±0.00 | 89.94±0.00 | 77.87±2.04 | 61.52±0.00 | 57.76±0.00 | 71.57 | 70.05 | 5.20 |
| | WiSE-FT | 72.72±8.35 | 83.52±3.24 | 86.26±1.45 | 62.19±2.74 | 51.66±0.43 | 71.67 | 72.81 | 3.80 | 68.92±0.01 | 86.48±0.54 | 79.32±0.00 | 63.14±0.00 | 56.58±0.00 | 70.89 | 70.46 | 6.00 |
| | $L^2$-SP | 73.05±2.80 | 82.49±1.95 | 81.60±1.23 | 63.21±2.21 | 53.92±0.82 | 70.85 | 72.62 | 3.00 | 74.74±1.31 | 86.20±2.30 | 81.62±1.01 | 63.66±0.00 | 57.81±0.00 | 72.81 | 73.34 | 4.00 |
| | Feature-map | 68.01±2.06 | 78.35±0.58 | 69.27±0.87 | 58.07±1.89 | 54.33±0.73 | 65.61 | 65.12 | 6.00 | 75.48±0.27 | 88.54±0.66 | 85.79±0.00 | 64.85±0.00 | 58.22±0.00 | 74.58 | 75.37 | 1.60 |
| | BSS | 76.21±6.50 | 83.52±1.90 | 81.69±0.40 | 63.54±2.05 | 53.26±0.84 | 71.64 | 73.21 | 2.20 | 69.93±3.53 | 86.70±1.52 | 82.64±0.83 | 63.06±1.79 | 57.55±0.48 | 71.98 | 71.88 | 4.60 |
| SCAFFOLD | Full-FT | 54.76±2.86 | 56.25±1.78 | 64.65±1.26 | 56.18±6.68 | 55.07±1.47 | 57.42 | 55.83 | 4.20 | 63.97±0.00 | 62.75±0.00 | 74.88±1.78 | 64.12±0.00 | 55.79±0.00 | 64.30 | 63.61 | 5.40 |
| | LP | 49.89±3.86 | 48.69±1.72 | 60.40±2.76 | 40.97±1.51 | 52.98±0.26 | 50.59 | 50.52 | 7.40 | 70.42±0.00 | 64.36±0.00 | 65.17±0.00 | 54.88±0.00 | 55.06±0.00 | 61.98 | 61.53 | 5.20 |
| | Surgical-FT | 56.64±4.28 | 54.30±2.39 | 66.81±0.67 | 53.60±2.54 | 55.29±0.58 | 57.33 | 55.41 | 4.20 | 75.38±0.00 | 64.72±0.00 | 77.93±0.00 | 69.00±0.00 | 54.35±0.00 | 68.28 | 69.70 | 3.00 |
| | LP-FT | 49.82±6.97 | 52.74±3.13 | 64.81±3.24 | 57.02±4.98 | 57.58±0.29 | 56.39 | 55.78 | 4.40 | 71.26±0.00 | 58.24±0.00 | 79.63±0.00 | 56.16±0.00 | 53.62±0.00 | 63.78 | 61.89 | 5.00 |
| | WiSE-FT | 58.53±5.22 | 56.16±1.85 | 64.17±1.08 | 53.49±6.18 | 55.11±1.23 | 57.49 | 56.60 | 4.40 | 70.33±0.00 | 65.28±0.00 | 75.68±1.80 | 64.47±0.00 | 55.67±0.00 | 66.29 | 66.69 | 3.40 |
| | $L^2$-SP | 57.60±4.63 | 57.53±1.08 | 64.50±1.83 | 59.39±3.16 | 57.05±1.02 | 59.21 | 58.17 | 2.60 | 68.62±4.37 | 60.41±1.38 | 77.83±0.00 | 71.86±0.00 | 54.53±1.21 | 66.65 | 66.96 | 4.60 |
| | Feature-map | 44.86±3.28 | 55.25±0.79 | 57.69±5.35 | 45.60±4.50 | 54.00±0.88 | 51.48 | 51.62 | 7.00 | 70.26±1.11 | 64.93±0.00 | 76.91±0.74 | 63.57±0.00 | 54.78±0.12 | 66.09 | 66.25 | 4.40 |
| | BSS | 58.38±3.59 | 58.27±0.49 | 70.00±2.70 | 58.52±2.49 | 56.50±1.02 | 60.33 | 58.39 | 1.80 | 67.78±0.01 | 64.18±0.07 | 77.20±1.05 | 72.13±0.93 | 53.48±0.88 | 66.95 | 68.03 | 5.00 |
| SIZE | Full-FT | 70.85±5.54 | 75.13±3.96 | 54.43±3.01 | 60.05±6.91 | 52.07±1.73 | 62.51 | 61.78 | 5.20 | 70.63±0.00 | 72.63±0.00 | 52.51±0.01 | 52.48±0.00 | 58.23±0.00 | 61.30 | 60.46 | 5.60 |
| | LP | 58.36±3.23 | 56.25±8.75 | 43.06±1.32 | 45.90±2.48 | 52.35±0.37 | 51.18 | 51.50 | 7.60 | 63.63±0.00 | 70.39±0.00 | 62.63±0.01 | 56.92±0.03 | 51.41±0.00 | 61.00 | 61.06 | 6.80 |
| | Surgical-FT | 67.51±7.23 | 81.75±2.07 | 60.97±1.53 | 62.45±1.60 | 54.19±0.38 | 65.37 | 63.64 | 3.00 | 71.87±0.01 | 83.49±0.00 | 62.88±0.01 | 65.03±0.00 | 55.99±0.00 | 67.85 | 66.59 | 3.40 |
| | LP-FT | 67.07±2.45 | 82.12±3.68 | 57.30±2.65 | 65.84±5.10 | 53.10±0.96 | 65.09 | 63.40 | 3.20 | 69.57±0.01 | 83.67±0.00 | 52.47±0.01 | 61.53±0.00 | 57.53±0.00 | 64.95 | 62.88 | 4.80 |
| | WiSE-FT | 70.06±5.49 | 73.88±4.80 | 52.09±3.06 | 56.91±5.90 | 54.21±0.75 | 61.43 | 60.39 | 4.80 | 72.61±0.03 | 71.35±0.00 | 54.49±0.00 | 63.13±0.00 | 58.37±0.00 | 63.99 | 64.28 | 3.80 |
| | $L^2$-SP | 65.62±4.40 | 79.46±0.79 | 55.84±4.07 | 63.81±7.20 | 53.82±1.27 | 63.71 | 61.76 | 4.40 | 72.35±1.40 | 78.03±1.06 | 54.40±2.22 | 67.14±0.00 | 56.49±0.24 | 65.68 | 65.33 | 3.80 |
| | Feature-map | 65.63±1.73 | 70.03±3.19 | 63.06±1.89 | 45.09±2.28 | 55.32±0.92 | 59.83 | 61.34 | 4.60 | 73.82±0.02 | 84.64±0.07 | 64.80±2.55 | 63.19±1.27 | 52.58±0.10 | 67.81 | 67.27 | 2.60 |
| | BSS | 70.90±2.39 | 77.56±2.51 | 59.84±4.41 | 65.31±6.67 | 52.59±1.16 | 65.24 | 65.35 | 3.20 | 68.24±1.78 | 78.43±2.51 | 56.42±3.96 | 57.58±9.04 | 56.42±1.60 | 63.42 | 60.75 | 5.20 |
| **FEWSHOT-500** | | | | | | | | | | | | | | | | | |
| RANDOM | Full-FT | 85.93±2.06 | 91.93±0.96 | 83.67±0.92 | 69.71±1.63 | 58.42±2.20 | 77.93 | 79.77 | 3.20 | 84.07±1.48 | 90.39±1.55 | 86.30±0.62 | 71.41±0.00 | 57.68±1.79 | 77.97 | 80.59 | 4.60 |
| | LP | 76.92±0.43 | 85.18±0.26 | 70.83±0.51 | 64.43±0.53 | 56.80±0.21 | 70.83 | 70.73 | 8.00 | 82.41±0.00 | 92.73±0.85 | 82.98±0.43 | 69.52±0.00 | 58.71±0.00 | 77.27 | 78.30 | 4.80 |
| | Surgical-FT | 83.62±1.90 | 91.68±0.46 | 86.18±0.83 | 68.37±0.74 | 60.29±0.87 | 78.03 | 79.39 | 3.40 | 83.31±0.00 | 90.31±0.00 | 87.93±0.00 | 70.24±0.00 | 61.57±0.00 | 78.67 | 80.49 | 3.80 |
| | LP-FT | 81.89±2.72 | 90.93±2.04 | 83.92±0.84 | 68.20±1.53 | 58.56±0.71 | 76.70 | 78.00 | 5.80 | 84.29±1.06 | 92.20±0.10 | 87.16±0.23 | 67.84±0.20 | 59.77±0.96 | 78.25 | 79.76 | 3.80 |
| | WiSE-FT | 85.10±2.16 | 91.53±1.15 | 84.19±0.86 | 69.60±1.37 | 58.25±2.04 | 77.73 | 79.63 | 4.20 | 84.69±0.41 | 92.35±0.50 | 86.41±0.64 | 71.40±0.00 | 57.89±0.15 | 78.55 | 80.83 | 3.60 |
| | $L^2$-SP | 84.17±3.97 | 92.19±1.11 | 84.82±0.95 | 70.06±0.93 | 59.31±0.96 | 78.11 | 79.68 | 2.00 | 89.58±2.13 | 88.52±0.72 | 82.31±1.41 | 67.14±0.14 | 57.10±0.75 | 76.93 | 79.32 | 6.00 |
| | Feature-map | 83.37±1.03 | 88.80±0.29 | 79.88±0.14 | 69.38±0.54 | 57.64±0.65 | 75.81 | 77.54 | 6.40 | 85.13±1.33 | 93.41±0.40 | 87.01±0.09 | 66.75±0.60 | 60.05±0.87 | 78.47 | 79.63 | 3.20 |
| | BSS | 85.84±1.94 | 91.81±0.80 | 84.68±0.83 | 69.38±1.98 | 58.85±1.05 | 78.11 | 79.97 | 3.00 | 88.61±0.79 | 89.84±2.25 | 82.33±0.77 | 66.69±0.03 | 56.61±1.58 | 76.82 | 79.21 | 6.40 |
| SCAFFOLD | Full-FT | 63.02±3.19 | 64.84±1.51 | 71.94±2.43 | 68.53±2.78 | 56.27±0.94 | 64.92 | 65.46 | 5.60 | 70.99±6.79 | 67.23±0.00 | 77.62±1.36 | 73.55±0.00 | 55.71±0.00 | 69.02 | 70.59 | 5.00 |
| | LP | 56.80±1.80 | 58.21±0.93 | 67.33±0.37 | 53.12±1.19 | 56.58±0.58 | 58.41 | 57.20 | 7.20 | 64.78±0.07 | 67.80±0.00 | 71.97±0.00 | 59.93±0.73 | | 66.88 | 67.49 | 5.40 |
| | Surgical-FT | 69.47±3.18 | 65.26±0.62 | 76.72±1.60 | 69.94±2.17 | 55.72±0.55 | 67.42 | 68.22 | 3.00 | 70.29±0.00 | 71.03±0.00 | 79.83±0.00 | 75.10±0.00 | 55.85±0.00 | 70.42 | 72.14 | 3.00 |
| | LP-FT | 65.09±3.54 | 64.23±1.67 | 69.36±2.11 | 69.41±1.48 | 57.33±0.44 | 65.08 | 66.23 | 4.60 | 77.43±0.00 | 66.06±0.27 | 78.31±0.00 | 71.62±1.81 | 56.95±0.00 | 70.07 | 71.70 | 4.60 |
| | WiSE-FT | 64.89±4.07 | 64.65±1.47 | 71.94±2.06 | 69.00±2.32 | 56.23±0.76 | 65.38 | 66.25 | 5.00 | 72.63±0.50 | 70.10±0.00 | 77.63±1.40 | 73.58±0.00 | 56.64±0.00 | 69.96 | 71.94 | 3.80 |
| | $L^2$-SP | 69.03±2.49 | 66.06±1.43 | 74.07±1.26 | 67.67±2.21 | 56.42±0.97 | 66.65 | 67.59 | 3.80 | 76.53±1.81 | 66.90±1.99 | 75.37±1.13 | 72.59±1.03 | 55.87±1.39 | 69.45 | 71.62 | 4.80 |
| | Feature-map | 60.04±3.11 | 63.87±0.70 | 75.42±0.70 | 60.08±2.03 | 58.45±0.38 | 63.57 | 61.33 | 4.80 | 68.33±0.25 | 67.77±0.48 | 78.93±0.05 | 73.99±0.07 | 58.79±0.10 | 69.56 | 70.70 | 3.60 |
| | BSS | 68.30±2.86 | 67.26±0.98 | 74.83±2.15 | 69.99±1.80 | 57.43±0.73 | 67.56 | 68.52 | 2.00 | 73.33±3.14 | 66.51±1.39 | 74.82±4.46 | 72.27±0.00 | 55.56±1.49 | 68.50 | 70.70 | 6.20 |
| SIZE | Full-FT | 60.10±5.25 | 76.35±2.26 | 50.25±3.29 | 56.23±5.29 | 54.40±1.70 | 59.47 | 56.91 | 6.00 | 75.35±1.23 | 87.68±0.71 | 54.92±3.57 | 61.36±8.21 | 52.47±0.00 | 67.28 | 66.22 | 5.40 |
| | LP | 55.35±0.51 | 63.98±1.71 | 40.46±4.26 | 58.26±7.53 | 51.43±0.20 | 54.82 | 56.55 | 7.40 | 73.86±1.59 | 85.67±0.00 | 56.69±0.00 | 62.62±0.00 | 52.17±1.22 | 66.20 | 64.89 | 6.20 |
| | Surgical-FT | 61.92±5.41 | 86.02±1.84 | 51.72±2.80 | 58.76±3.21 | 56.61±1.07 | 63.13 | 59.10 | 3.00 | 72.39±0.00 | 87.00±0.00 | 61.62±0.00 | 63.64±0.00 | 53.03±0.00 | 67.54 | 65.88 | 4.60 |
| | LP-FT | 55.39±4.42 | 78.83±7.22 | 53.66±3.35 | 62.85±4.81 | 55.21±1.62 | 61.19 | 57.82 | 4.60 | 79.69±3.59 | 85.97±1.50 | 60.12±2.20 | 60.32±4.95 | 51.27±0.00 | 67.47 | 66.71 | 5.60 |
| | WiSE-FT | 62.14±1.97 | 75.21±2.23 | 48.40±2.94 | 53.63±3.76 | 56.19±1.22 | 59.11 | 57.32 | 5.80 | 80.52±1.79 | 87.80±0.69 | 54.19±3.09 | 70.11±1.43 | 53.01±0.00 | 69.13 | 68.27 | 4.00 |
| | $L^2$-SP | 64.97±0.50 | 83.22±1.87 | 51.14±4.26 | 69.62±3.36 | 56.72±1.04 | 65.13 | 63.77 | 1.80 | 73.19±0.12 | 85.30±2.61 | 59.43±3.01 | 72.12±2.91 | 55.04±0.79 | 69.02 | 68.25 | 4.40 |
| | Feature-map | 63.06±1.12 | 80.15±1.70 | 43.45±0.50 | 66.24±0.37 | 53.29±0.71 | 61.24 | 60.86 | 4.60 | 76.08±0.12 | 90.22±0.04 | 62.58±1.34 | 66.73±0.33 | 53.11±0.07 | 69.74 | 68.46 | 2.20 |
| | BSS | 62.87±5.70 | 80.69±2.55 | 51.61±4.52 | 67.37±4.52 | 56.48±2.00 | 63.80 | 62.24 | 2.80 | 82.39±5.20 | 84.69±1.31 | 60.22±3.85 | 75.06±0.00 | 54.33±0.73 | 71.34 | 72.56 | 3.00 |

Table 11: Robust fine-tuning performance on 4 **Regression** datasets (RMSE metrics) in the **Fewshot** setting (covering FEWSHOT-50, FEWSHOT-100, and FEWSHOT-500), evaluated across 3 dataset splits (RANDOM, SCAFFOLD, SIZE) over **MOLECULESTM** and **GRAPHIUM-LARGE** models. AVG-R, AVG-R* denote the average rank and the rank based on the average normalized performance over all the datasets for each evavluated method, respectively. Standard deviations across five replicates are shown in parentheses. We **bold** and underline the best and second-best performances in each scenario.

| Split | Methods | Self-supervised Pre-training (MoleculeSTM) | | | | | | Supervised Pre-training (Graphium-Large) | | | | | |
|---|---|---|---|---|---|---|---|---|---|---|---|---|---|
| | | Esol | Lipo | Malaria | CEP | Avg-R | Avg-R* | Esol | Lipo | Malaria | CEP | Avg-R | Avg-R* |
| **FEWSHOT-50** | | | | | | | | | | | | | |
| RANDOM | Full-FT | 2.128±0.072 | 1.247±0.031 | 1.310±0.025 | 3.433±0.226 | 5.00 | 6 | 1.125±0.000 | 1.156±0.019 | 1.277±0.000 | 2.198±0.001 | 5.75 | 7 |
| | LP | 2.971±0.017 | 1.638±0.014 | 1.309±0.012 | 3.519±0.052 | 6.75 | 8 | 1.176±0.000 | 1.131±0.000 | 1.294±0.000 | 2.113±0.000 | 6.50 | 8 |
| | Surgical-FT | 2.315±0.081 | 1.327±0.017 | 1.317±0.024 | 3.272±0.199 | 6.50 | 7 | 1.055±0.000 | 1.076±0.000 | 1.283±0.000 | 2.192±0.000 | 4.00 | 4 |
| | LP-FT | 1.600±0.129 | 1.181±0.030 | 1.356±0.011 | 2.358±0.037 | 4.25 | 4 | 1.096±0.000 | 1.032±0.002 | 1.293±0.000 | 2.092±0.002 | 3.00 | 1 |
| | WiSE-FT | 2.135±0.072 | 1.261±0.035 | 1.298±0.023 | 3.576±0.235 | 5.50 | 5 | 1.116±0.000 | 1.151±0.024 | 1.278±0.000 | 2.075±0.004 | 4.00 | 3 |
| | $L^2$-SP | 1.472±0.036 | 1.165±0.037 | 1.297±0.006 | 2.304±0.055 | 1.50 | 1 | 1.161±0.000 | 1.077±0.019 | 1.276±0.000 | 2.127±0.015 | 4.00 | 5 |
| | Feature-map | 1.632±0.028 | 1.257±0.025 | 1.301±0.009 | 2.398±0.037 | 4.00 | 3 | 1.133±0.002 | 1.106±0.003 | 1.277±0.001 | 2.108±0.002 | 3.75 | 2 |
| | BSS | 1.450±0.057 | 1.171±0.021 | 1.314±0.018 | 2.244±0.036 | 2.50 | 2 | 1.188±0.004 | 1.109±0.021 | 1.276±0.000 | 2.108±0.029 | 4.25 | 6 |
| SCAFFOLD | Full-FT | 2.790±0.116 | 1.434±0.072 | 1.195±0.025 | 3.395±0.191 | 5.75 | 6 | 1.237±0.000 | 1.079±0.000 | 1.175±0.000 | 2.051±0.000 | 4.00 | 7 |
| | LP | 3.538±0.075 | 1.755±0.021 | 1.206±0.012 | 3.870±0.038 | 7.75 | 8 | 0.929±0.000 | 1.096±0.000 | 1.170±0.000 | 2.053±0.000 | 3.75 | 1 |
| | Surgical-FT | 3.018±0.118 | 1.491±0.085 | 1.191±0.004 | 3.304±0.347 | 5.75 | 7 | 1.240±0.000 | 1.044±0.000 | 1.180±0.000 | 2.009±0.000 | 4.00 | 2 |
| | LP-FT | 1.636±0.021 | 1.181±0.029 | 1.263±0.009 | 2.294±0.024 | 4.00 | 4 | 1.241±0.000 | 1.085±0.000 | 1.176±0.000 | 2.044±0.000 | 5.00 | 8 |
| | WiSE-FT | 2.762±0.091 | 1.405±0.067 | 1.181±0.008 | 3.496±0.199 | 4.50 | 5 | 1.247±0.000 | 1.099±0.000 | 1.166±0.000 | 2.024±0.000 | 4.25 | 4 |
| | $L^2$-SP | 1.654±0.086 | 1.178±0.022 | 1.185±0.008 | 2.255±0.026 | 2.25 | 2 | 1.280±0.003 | 1.063±0.000 | 1.175±0.000 | 1.997±0.016 | 5.50 | 6 |
| | Feature-map | 1.783±0.034 | 1.252±0.012 | 1.195±0.008 | 2.401±0.028 | 4.50 | 3 | 1.267±0.110 | 1.037±0.006 | 1.170±0.143 | 2.073±0.016 | 4.75 | 5 |
| | BSS | 1.632±0.048 | 1.173±0.022 | 1.182±0.016 | 2.287±0.028 | 1.50 | 1 | 1.159±0.007 | 1.100±0.002 | 1.162±0.000 | 2.060±0.009 | 4.25 | 3 |
| SIZE | Full-FT | 3.457±0.086 | 1.407±0.088 | 1.064±0.067 | 3.311±0.158 | 6.25 | 7 | 1.499±0.000 | 1.108±0.000 | 0.909±0.000 | 2.321±0.000 | 3.50 | 4 |
| | LP | 3.758±0.010 | 1.773±0.025 | 0.990±0.056 | 4.114±0.042 | 6.75 | 8 | 2.025±0.000 | 1.325±0.000 | 0.917±0.000 | 2.358±0.000 | 7.50 | 8 |
| | Surgical-FT | 3.429±0.139 | 1.543±0.083 | 0.990±0.054 | 3.195±0.306 | 5.25 | 6 | 1.675±0.000 | 1.089±0.000 | 0.916±0.000 | 2.271±0.000 | 4.50 | 1 |
| | LP-FT | 2.035±0.080 | 1.208±0.078 | 1.102±0.018 | 2.500±0.045 | 4.00 | 4 | 1.540±0.000 | 1.079±0.001 | 0.994±0.000 | 2.347±0.001 | 5.75 | 7 |
| | WiSE-FT | 3.527±0.112 | 1.392±0.062 | 0.983±0.053 | 3.386±0.142 | 5.00 | 5 | 1.536±0.000 | 1.149±0.000 | 0.911±0.000 | 2.321±0.000 | 4.50 | 5 |
| | $L^2$-SP | 2.111±0.091 | 1.159±0.037 | 0.988±0.032 | 2.421±0.045 | 2.00 | 1 | 1.673±0.000 | 1.072±0.002 | 0.948±0.007 | 2.304±0.022 | 4.25 | 6 |
| | Feature-map | 2.331±0.050 | 1.225±0.049 | 1.000±0.034 | 2.439±0.024 | 4.00 | 3 | 1.594±0.010 | 1.070±0.012 | 0.915±0.001 | 2.306±0.008 | 3.25 | 3 |
| | BSS | 2.197±0.084 | 1.106±0.027 | 1.019±0.033 | 2.419±0.045 | 2.75 | 2 | 1.516±0.008 | 1.076±0.043 | 0.907±0.000 | 2.313±0.049 | 2.50 | 2 |
| **FEWSHOT-100** | | | | | | | | | | | | | |
| RANDOM | Full-FT | 1.842±0.208 | 1.205±0.059 | 1.289±0.032 | 2.784±0.110 | 5.75 | 6 | 1.121±0.000 | 1.187±0.020 | 1.259±0.000 | 1.902±0.011 | 5.00 | 6 |
| | LP | 2.391±0.044 | 1.623±0.011 | 1.279±0.007 | 3.176±0.093 | 7.00 | 8 | 0.912±0.000 | 1.068±0.000 | 1.286±0.000 | 1.920±0.014 | 4.75 | 4 |
| | Surgical-FT | 1.650±0.063 | 1.301±0.037 | 1.277±0.012 | 2.777±0.181 | 5.00 | 4 | 0.952±0.000 | 1.061±0.000 | 1.269±0.000 | 1.881±0.000 | 2.25 | 2 |
| | LP-FT | 1.540±0.123 | 1.234±0.030 | 1.350±0.016 | 2.203±0.030 | 4.50 | 7 | 1.061±0.005 | 1.126±0.000 | 1.290±0.011 | 1.918±0.005 | 6.00 | 7 |
| | WiSE-FT | 1.790±0.147 | 1.207±0.058 | 1.282±0.017 | 2.842±0.123 | 5.50 | 5 | 1.064±0.000 | 1.121±0.050 | 1.258±0.000 | 1.905±0.015 | 3.75 | 3 |
| | $L^2$-SP | 1.486±0.105 | 1.190±0.038 | 1.267±0.007 | 2.207±0.046 | 1.75 | 1 | 1.109±0.082 | 1.094±0.007 | 1.276±0.000 | 1.916±0.022 | 5.00 | 5 |
| | Feature-map | 1.557±0.034 | 1.252±0.007 | 1.269±0.002 | 2.130±0.020 | 3.25 | 2 | 0.897±0.009 | 1.053±0.007 | 1.273±0.000 | 1.881±0.011 | 1.75 | 1 |
| | BSS | 1.543±0.044 | 1.190±0.031 | 1.285±0.011 | 2.170±0.028 | 3.25 | 3 | 1.159±0.012 | 1.129±0.022 | 1.276±0.004 | 2.036±0.139 | 7.00 | 8 |
| SCAFFOLD | Full-FT | 2.036±0.119 | 1.108±0.017 | 1.205±0.050 | 2.942±0.208 | 5.75 | 6 | 1.238±0.000 | 1.027±0.000 | 1.187±0.000 | 1.986±0.019 | 6.75 | 7 |
| | LP | 2.906±0.093 | 1.389±0.008 | 1.180±0.017 | 3.635±0.051 | 6.75 | 8 | 1.184±0.013 | 0.998±0.000 | 1.163±0.000 | 1.935±0.000 | 3.25 | 3 |
| | Surgical-FT | 1.956±0.170 | 1.190±0.027 | 1.183±0.016 | 2.848±0.120 | 5.50 | 5 | 1.121±0.000 | 0.977±0.000 | 1.172±0.000 | 1.914±0.000 | 2.50 | 1 |
| | LP-FT | 1.775±0.178 | 1.103±0.024 | 1.288±0.012 | 2.310±0.034 | 4.75 | 7 | 1.210±0.001 | 1.062±0.003 | 1.206±0.000 | 1.918±0.002 | 6.00 | 8 |
| | WiSE-FT | 2.052±0.082 | 1.112±0.023 | 1.188±0.027 | 3.049±0.246 | 6.25 | 4 | 1.199±0.000 | 1.002±0.000 | 1.160±0.000 | 1.988±0.028 | 4.50 | 5 |
| | $L^2$-SP | 1.559±0.047 | 1.069±0.044 | 1.166±0.004 | 2.227±0.036 | 1.75 | 1 | 1.210±0.030 | 0.999±0.035 | 1.176±0.015 | 2.000±0.009 | 5.75 | 6 |
| | Feature-map | 1.576±0.028 | 1.123±0.009 | 1.181±0.005 | 2.216±0.014 | 3.50 | 3 | 1.106±0.025 | 0.957±0.008 | 1.159±0.003 | 2.047±0.008 | 2.75 | 2 |
| | BSS | 1.680±0.098 | 1.081±0.019 | 1.163±0.004 | 2.212±0.018 | 1.75 | 2 | 1.169±0.035 | 1.025±0.000 | 1.170±0.014 | 1.938±0.030 | 4.25 | 4 |
| SIZE | Full-FT | 2.527±0.152 | 1.113±0.054 | 1.022±0.046 | 2.587±0.100 | 6.25 | 7 | 1.675±0.003 | 1.132±0.000 | 0.909±0.000 | 2.317±0.000 | 5.25 | 6 |
| | LP | 3.020±0.061 | 1.492±0.039 | 0.951±0.011 | 3.408±0.041 | 6.75 | 8 | 1.740±0.000 | 1.245±0.000 | 0.934±0.000 | 2.355±0.000 | 7.75 | 8 |
| | Surgical-FT | 2.435±0.119 | 1.119±0.037 | 0.970±0.020 | 2.607±0.040 | 6.25 | 6 | 1.501±0.000 | 1.091±0.000 | 0.902±0.000 | 2.241±0.000 | 2.50 | 1 |
| | LP-FT | 1.937±0.120 | 1.050±0.052 | 1.045±0.012 | 2.506±0.042 | 4.25 | 5 | 1.662±0.009 | 1.228±0.002 | 0.939±0.003 | 2.310±0.005 | 6.50 | 7 |
| | WiSE-FT | 2.580±0.096 | 1.086±0.051 | 0.962±0.043 | 2.556±0.089 | 5.00 | 4 | 1.605±0.001 | 1.159±0.000 | 0.907±0.000 | 2.300±0.000 | 4.00 | 5 |
| | $L^2$-SP | 1.860±0.183 | 1.063±0.006 | 0.931±0.007 | 2.436±0.043 | 1.75 | 1 | 1.474±0.031 | 1.047±0.097 | 0.915±0.008 | 2.256±0.020 | 2.75 | 3 |
| | Feature-map | 1.921±0.086 | 1.098±0.036 | 0.936±0.009 | 2.374±0.011 | 2.75 | 2 | 1.494±0.038 | 1.085±0.012 | 0.915±0.000 | 2.303±0.006 | 3.75 | 4 |
| | BSS | 1.854±0.109 | 1.075±0.032 | 0.962±0.017 | 2.444±0.014 | 3.00 | 3 | 1.325±0.017 | 1.011±0.045 | 0.909±0.002 | 2.322±0.002 | 3.00 | 2 |
| **FEWSHOT-500** | | | | | | | | | | | | | |
| RANDOM | Full-FT | 1.093±0.085 | 0.834±0.014 | 1.245±0.018 | 1.874±0.042 | 5.00 | 6 | 0.702±0.006 | 0.849±0.006 | 1.217±0.000 | 1.801±0.018 | 5.00 | 5 |
| | LP | 1.542±0.011 | 1.136±0.006 | 1.253±0.003 | 2.435±0.019 | 8.00 | 8 | 0.732±0.000 | 0.829±0.000 | 1.225±0.000 | 1.809±0.011 | 6.25 | 7 |
| | Surgical-FT | 1.177±0.043 | 0.888±0.010 | 1.233±0.009 | 1.948±0.005 | 6.00 | 7 | 0.643±0.000 | 0.800±0.000 | 1.207±0.000 | 1.775±0.000 | 1.50 | 1 |
| | LP-FT | 1.001±0.020 | 0.838±0.020 | 1.244±0.011 | 1.850±0.019 | 4.00 | 5 | 0.664±0.001 | 0.837±0.019 | 1.204±0.000 | 1.809±0.019 | 3.75 | 2 |
| | WiSE-FT | 1.076±0.074 | 0.833±0.007 | 1.236±0.012 | 1.898±0.051 | 4.25 | 4 | 0.661±0.009 | 0.848±0.005 | 1.207±0.000 | 1.802±0.025 | 3.50 | 3 |
| | $L^2$-SP | 0.992±0.034 | 0.838±0.009 | 1.225±0.005 | 1.839±0.024 | 2.75 | 1 | 0.714±0.041 | 0.827±0.011 | 1.223±0.006 | 1.830±0.014 | 5.75 | 8 |
| | Feature-map | 1.070±0.020 | 0.948±0.010 | 1.216±0.002 | 1.904±0.003 | 4.50 | 3 | 0.671±0.014 | 0.791±0.007 | 1.210±0.002 | 1.849±0.002 | 4.25 | 4 |
| | BSS | 0.990±0.046 | 0.829±0.018 | 1.231±0.009 | 1.835±0.023 | 1.50 | 2 | 0.715±0.035 | 0.816±0.015 | 1.228±0.003 | 1.808±0.009 | 5.50 | 6 |
| SCAFFOLD | Full-FT | 1.434±0.044 | 0.885±0.028 | 1.186±0.017 | 1.910±0.022 | 5.00 | 6 | 1.025±0.011 | 0.856±0.016 | 1.125±0.000 | 1.808±0.023 | 5.50 | 6 |
| | LP | 2.047±0.020 | 1.026±0.003 | 1.168±0.005 | 2.572±0.018 | 7.25 | 8 | 0.929±0.003 | 0.841±0.000 | 1.151±0.000 | 1.787±0.000 | 4.50 | 3 |
| | Surgical-FT | 1.323±0.053 | 0.940±0.016 | 1.159±0.014 | 1.920±0.010 | 4.50 | 5 | 0.943±0.000 | 0.812±0.000 | 1.138±0.000 | 1.793±0.000 | 2.00 | 2 |
| | LP-FT | 1.394±0.025 | 0.888±0.017 | 1.204±0.015 | 1.876±0.024 | 5.00 | 7 | 0.962±0.004 | 0.847±0.001 | 1.133±0.003 | 1.809±0.019 | 4.75 | 4 |
| | WiSE-FT | 1.423±0.032 | 0.885±0.023 | 1.170±0.014 | 1.926±0.035 | 5.50 | 4 | 0.995±0.013 | 0.851±0.010 | 1.123±0.000 | 1.807±0.020 | 5.50 | 5 |
| | $L^2$-SP | 1.375±0.030 | 0.879±0.008 | 1.139±0.001 | 1.870±0.032 | 1.75 | 1 | 0.996±0.044 | 0.861±0.012 | 1.122±0.005 | 1.828±0.004 | 5.75 | 7 |
| | Feature-map | 1.453±0.028 | 0.903±0.004 | 1.154±0.003 | 1.913±0.016 | 5.25 | 3 | 0.881±0.005 | 0.808±0.003 | 1.145±0.000 | 1.747±0.014 | 1.00 | 1 |
| | BSS | 1.367±0.043 | 0.881±0.024 | 1.150±0.020 | 1.866±0.018 | 1.75 | 2 | 0.976±0.029 | 0.859±0.004 | 1.158±0.010 | 1.817±0.013 | 3.00 | 8 |
| SIZE | Full-FT | 1.797±0.088 | 0.793±0.019 | 0.997±0.019 | 2.353±0.033 | 5.50 | 7 | 1.198±0.000 | 0.863±0.001 | 0.926±0.000 | 2.235±0.011 | 6.00 | 7 |
| | LP | 2.581±0.049 | 1.030±0.004 | 0.943±0.005 | 2.990±0.030 | 6.75 | 8 | 1.375±0.000 | 0.934±0.000 | 0.938±0.000 | 2.300±0.000 | 8.00 | 8 |
| | Surgical-FT | 1.540±0.078 | 0.846±0.011 | 0.944±0.010 | 2.403±0.038 | 4.50 | 4 | 1.289±0.000 | 0.820±0.000 | 0.917±0.000 | 2.198±0.000 | 3.75 | 4 |
| | LP-FT | 1.717±0.077 | 0.809±0.004 | 0.956±0.014 | 2.287±0.043 | 4.50 | 5 | 1.147±0.000 | 0.855±0.015 | 0.907±0.005 | 2.220±0.031 | 3.25 | 3 |
| | WiSE-FT | 1.874±0.084 | 0.805±0.012 | 0.955±0.019 | 2.363±0.035 | 5.50 | 6 | 1.189±0.000 | 0.873±0.001 | 0.908±0.000 | 2.233±0.007 | 4.75 | 5 |
| | $L^2$-SP | 1.592±0.089 | 0.788±0.014 | 0.930±0.008 | 2.297±0.014 | 2.75 | 2 | 1.114±0.038 | 0.805±0.030 | 0.903±0.009 | 2.220±0.012 | 1.75 | 1 |
| | Feature-map | 1.580±0.070 | 0.873±0.016 | 0.921±0.002 | 2.286±0.036 | 2.75 | 2 | 1.241±0.116 | 0.833±0.010 | 0.917±0.001 | 2.236±0.024 | 5.50 | 6 |
| | BSS | 1.617±0.117 | 0.783±0.018 | 0.957±0.007 | 2.295±0.038 | 3.75 | 3 | 1.189±0.029 | 0.829±0.022 | 0.901±0.007 | 2.219±0.000 | 2.25 | 2 |

Table 12: Robust fine-tuning performance on 8 **Classification** datasets (AUC metrics) in the **Non-Fewshot** setting, evaluated across 3 dataset splits (RANDOM, SCAFFOLD, SIZE), over **GRAPHMAE** and **GRAPHGPS** models. AVG, AVG-F, AVG-R denote the average AUC, average AUC without max and min values, and average rank over all the datasets for each method, respectively. Standard deviations across five replicates are shown in parentheses. We **bold** and underline the best and second-best performances in each scenario.

| SPLIT | METHODS | CLINTOX | BBBP | BACE | HIV | MUV | SIDER | TOX21 | TOXCAST | AVG | AVG-F | AVG-R |
|---|---|---|---|---|---|---|---|---|---|---|---|---|
| | | | | **SELF-SUPERVISED PRE-TRAINING (GRAPHMAE)** | | | | | | | | |
| RANDOM | FULL-FT | 83.22 ± 2.07 | 94.70 ± 0.32 | 89.26 ± 0.40 | 85.31 ± 0.29 | 80.71 ± 0.58 | 61.53 ± 0.48 | 82.35 ± 0.15 | 73.01 ± 0.16 | 81.26 | 82.31 | 4.00 |
| | LP | 78.82 ± 1.55 | 83.16 ± 0.58 | 77.65 ± 1.27 | 74.45 ± 0.31 | 78.54 ± 1.16 | 61.51 ± 0.35 | 73.57 ± 0.16 | 66.96 ± 0.16 | 74.33 | 75.00 | 7.50 |
| | SURGICAL-FT | 83.85 ± 1.52 | 92.11 ± 0.35 | 86.77 ± 0.09 | 84.56 ± 0.30 | **82.71 ± 0.81** | 61.79 ± 0.19 | 79.90 ± 0.14 | 71.51 ± 0.21 | 80.40 | 81.55 | 4.50 |
| | LP-FT | **88.09 ± 1.04** | 94.68 ± 0.19 | 89.58 ± 0.23 | **86.06 ± 0.43** | 80.75 ± 1.53 | 61.69 ± 0.26 | 82.50 ± 0.21 | **73.66 ± 0.07** | 82.13 | 83.44 | 2.25 |
| | WISE-FT | 80.01 ± 4.00 | 93.04 ± 0.46 | 90.15 ± 0.50 | 85.42 ± 0.52 | 82.07 ± 2.10 | 62.18 ± 0.49 | 81.55 ± 0.43 | 72.48 ± 0.26 | 80.86 | 81.95 | 3.38 |
| | L2-SP | 83.39 ± 1.88 | 93.89 ± 0.28 | 88.70 ± 0.10 | 80.22 ± 0.17 | 73.35 ± 1.54 | **62.36 ± 0.43** | 77.45 ± 0.47 | 68.71 ± 0.31 | 78.51 | 78.64 | 5.00 |
| | FEATURE-MAP | 73.08 ± 0.89 | 85.36 ± 0.46 | 75.88 ± 0.75 | 77.04 ± 0.26 | 79.53 ± 1.25 | 62.06 ± 0.32 | 75.36 ± 0.13 | 65.69 ± 0.24 | 74.25 | 74.43 | 6.75 |
| | BSS | 83.98 ± 3.00 | **94.85 ± 0.31** | 89.31 ± 0.21 | 86.05 ± 0.40 | 80.55 ± 0.75 | 61.92 ± 0.21 | 82.48 ± 0.28 | 73.22 ± 0.07 | 81.54 | 82.60 | 2.62 |
| SCAFFOLD | FULL-FT | 74.74 ± 0.93 | 66.35 ± 0.65 | 80.33 ± 0.37 | 77.22 ± 0.38 | 77.47 ± 1.33 | 60.98 ± 0.19 | 76.18 ± 0.31 | 64.27 ± 0.36 | 72.19 | 72.70 | 3.88 |
| | LP | 71.34 ± 1.48 | 64.36 ± 0.24 | 61.70 ± 7.34 | 70.62 ± 0.64 | 79.13 ± 1.20 | 58.23 ± 1.29 | 70.89 ± 0.10 | 60.03 ± 0.13 | 67.04 | 66.49 | 6.75 |
| | SURGICAL-FT | 71.88 ± 1.07 | 66.81 ± 0.29 | 80.24 ± 0.90 | 76.90 ± 0.30 | **79.20 ± 0.50** | **64.00 ± 0.09** | 74.18 ± 0.40 | 62.60 ± 0.27 | 71.98 | 72.16 | 4.12 |
| | LP-FT | 74.88 ± 2.00 | 67.39 ± 0.55 | 80.67 ± 0.57 | **77.97 ± 0.38** | 75.13 ± 1.06 | 60.76 ± 0.45 | 76.18 ± 0.20 | 64.29 ± 0.23 | 72.16 | 72.64 | 3.25 |
| | WISE-FT | **77.30 ± 5.30** | 69.29 ± 2.34 | 82.16 ± 1.50 | 76.75 ± 0.69 | 77.76 ± 1.55 | 59.76 ± 0.86 | 74.99 ± 0.44 | 63.61 ± 0.34 | 72.70 | 73.28 | 3.25 |
| | L2-SP | 73.40 ± 0.45 | 67.39 ± 0.90 | 80.36 ± 0.92 | 74.63 ± 0.44 | 73.20 ± 0.90 | 63.40 ± 0.29 | 73.16 ± 0.14 | 61.29 ± 0.38 | 70.85 | 70.86 | 5.00 |
| | FEATURE-MAP | 64.74 ± 0.62 | 62.46 ± 0.26 | 69.22 ± 2.06 | 72.34 ± 0.58 | 75.63 ± 0.54 | 57.13 ± 1.08 | 71.25 ± 0.13 | 57.78 ± 0.26 | 66.32 | 66.30 | 7.38 |
| | BSS | 75.80 ± 1.11 | 67.46 ± 1.35 | 80.82 ± 0.62 | 77.10 ± 0.77 | 78.53 ± 1.03 | 62.29 ± 0.51 | **76.45 ± 0.24** | 64.03 ± 0.09 | 72.81 | 73.23 | 2.38 |
| SIZE | FULL-FT | 56.52 ± 0.81 | 80.05 ± 2.01 | 59.94 ± 1.83 | 77.21 ± 0.94 | 74.64 ± 1.72 | 53.04 ± 0.74 | 70.87 ± 0.24 | 60.80 ± 0.50 | 66.63 | 66.66 | 4.62 |
| | LP | 57.44 ± 0.94 | 73.52 ± 1.68 | 51.46 ± 0.97 | 73.91 ± 0.89 | 65.97 ± 3.36 | 51.84 ± 0.31 | 67.56 ± 0.10 | 57.49 ± 0.11 | 62.40 | 62.30 | 7.38 |
| | SURGICAL-FT | 57.47 ± 1.16 | 81.96 ± 0.78 | 55.85 ± 2.81 | **80.48 ± 0.18** | 75.86 ± 2.96 | 54.32 ± 0.43 | 71.19 ± 0.30 | 59.45 ± 0.18 | 67.07 | 66.72 | 3.12 |
| | LP-FT | 56.35 ± 0.62 | 76.80 ± 2.24 | 61.61 ± 1.01 | 77.14 ± 0.69 | **79.10 ± 0.89** | 52.69 ± 0.35 | **71.33 ± 0.26** | 60.98 ± 0.27 | 67.00 | 67.37 | 4.00 |
| | WISE-FT | **59.25 ± 3.49** | **82.99 ± 1.91** | 61.16 ± 2.31 | 75.90 ± 1.94 | 75.09 ± 3.95 | **55.74 ± 1.28** | 70.94 ± 0.42 | **61.53 ± 0.56** | 67.83 | 67.31 | 2.50 |
| | L2-SP | 59.11 ± 0.88 | 80.40 ± 1.50 | 61.10 ± 1.54 | 76.67 ± 1.61 | 65.11 ± 0.75 | 53.81 ± 0.72 | 68.96 ± 0.47 | 57.85 ± 0.36 | 65.38 | 64.80 | 4.88 |
| | FEATURE-MAP | 59.02 ± 0.89 | 77.60 ± 0.45 | 43.17 ± 0.32 | 79.17 ± 0.23 | 73.54 ± 0.29 | 52.23 ± 0.32 | 68.74 ± 0.09 | 53.39 ± 0.51 | 63.36 | 64.09 | 5.75 |
| | BSS | 58.58 ± 1.31 | 80.86 ± 1.92 | **61.96 ± 2.00** | 79.14 ± 0.79 | 73.35 ± 1.27 | 53.14 ± 0.63 | 70.76 ± 0.37 | 60.62 ± 0.35 | 67.30 | 67.40 | 3.75 |
| | | | | **SUPERVISED PRE-TRAINING (GRAPHGPS)** | | | | | | | | |
| RANDOM | FULL-FT | 99.77 ± 0.01 | 99.99 ± 0.01 | **100.00 ± 0.00** | 84.80 ± 0.33 | 57.06 ± 0.00 | 87.13 ± 0.39 | 87.17 ± 0.48 | 86.90 ± 0.17 | 87.85 | 90.96 | 4.00 |
| | LP | 99.48 ± 0.04 | 86.96 ± 0.40 | 80.94 ± 0.45 | 86.70 ± 0.42 | 63.97 ± 0.80 | 84.77 ± 0.08 | 82.70 ± 0.14 | 83.93 ± 0.04 | 83.68 | 84.33 | 5.50 |
| | SURGICAL-FT | 99.65 ± 0.05 | 99.16 ± 0.00 | 98.14 ± 0.04 | 86.58 ± 0.03 | 60.52 ± 0.64 | 47.74 ± 0.95 | 51.53 ± 0.00 | 51.71 ± 0.00 | 74.38 | 74.61 | 5.88 |
| | LP-FT | 99.54 ± 0.14 | 89.67 ± 5.14 | 84.88 ± 7.57 | 85.71 ± 1.11 | 63.96 ± 0.80 | 85.97 ± 2.43 | 83.98 ± 2.45 | 84.48 ± 1.09 | 84.77 | 85.78 | 5.12 |
| | WISE-FT | 97.04 ± 1.00 | 58.14 ± 3.98 | 68.29 ± 2.24 | 87.75 ± 0.07 | 67.81 ± 0.32 | 77.59 ± 0.03 | 70.80 | 69.90 | 7.62 |
| | $L^2$-SP | **99.84 ± 0.03** | **100.00 ± 0.00** | **100.00 ± 0.00** | 97.75 ± 0.07 | 74.51 ± 1.12 | 92.16 ± 0.44 | 92.28 ± 0.46 | 89.79 ± 0.07 | 93.29 | 95.30 | 1.25 |
| | FEATURE-MAP | 99.79 ± 0.09 | 100.00 ± 0.00 | 100.00 ± 0.00 | **99.42 ± 0.01** | 53.07 ± 0.82 | 91.64 ± 0.06 | 91.61 ± 0.16 | 89.39 ± 0.06 | 90.62 | 95.31 | 2.62 |
| | BSS | 99.77 ± 0.00 | 100.00 ± 0.00 | 100.00 ± 0.00 | 84.87 ± 0.02 | 58.93 ± 3.25 | 87.61 ± 0.05 | 87.52 ± 0.10 | 86.75 ± 0.05 | 88.18 | 91.09 | 4.00 |
| SCAFFOLD | FULL-FT | 99.76 ± 0.04 | 99.99 ± 0.01 | **100.00 ± 0.00** | 83.67 ± 1.61 | 57.08 ± 1.77 | 87.26 ± 0.15 | 87.16 ± 0.21 | 86.71 ± 0.12 | 87.70 | 90.76 | 4.12 |
| | LP | 99.47 ± 0.04 | 86.84 ± 0.49 | 81.04 ± 0.53 | 86.66 ± 0.44 | 63.98 ± 0.82 | 84.74 ± 0.08 | 82.70 ± 0.14 | 83.93 ± 0.04 | 83.67 | 84.32 | 5.75 |
| | SURGICAL-FT | 99.64 ± 0.08 | 99.33 ± 0.14 | 98.14 ± 0.06 | 87.61 ± 0.63 | 61.75 ± 0.39 | 76.46 ± 1.75 | 72.53 ± 1.99 | 55.58 ± 0.35 | 81.38 | 82.64 | 5.75 |
| | LP-FT | 99.54 ± 0.15 | 89.53 ± 5.24 | 84.35 ± 6.50 | 84.81 ± 2.36 | 62.46 ± 1.48 | 85.96 ± 2.47 | 83.96 ± 2.42 | 84.52 ± 1.17 | 84.39 | 85.52 | 5.38 |
| | WISE-FT | 97.32 ± 0.16 | 64.59 ± 3.69 | 100.00 ± 0.00 | 67.98 ± 4.58 | 49.84 ± 0.72 | 80.53 ± 0.07 | 68.04 ± 0.17 | 77.53 ± 0.02 | 75.73 | 76.00 | 7.00 |
| | $L^2$-SP | 99.83 ± 0.03 | **100.00 ± 0.00** | 100.00 ± 0.00 | 98.35 ± 0.43 | **74.63 ± 0.95** | 92.33 ± 0.21 | 92.43 ± 0.34 | 89.85 ± 0.17 | 93.43 | 95.46 | 1.50 |
| | FEATURE-MAP | 99.85 ± 0.01 | 100.00 ± 0.00 | 100.00 ± 0.00 | **99.26 ± 0.13** | 55.32 ± 0.31 | 91.63 ± 0.04 | 91.61 ± 0.11 | 89.30 ± 0.06 | 90.87 | 95.27 | 2.62 |
| | BSS | 99.81 ± 0.04 | 99.99 ± 0.01 | 100.00 ± 0.00 | 85.03 ± 0.57 | 60.82 ± 4.94 | 89.80 ± 3.20 | 87.36 ± 0.09 | 86.85 ± 0.12 | 88.71 | 91.47 | 3.88 |
| SIZE | FULL-FT | 99.76 ± 0.03 | 99.99 ± 0.01 | **100.00 ± 0.00** | 83.42 ± 1.75 | 56.61 ± 1.51 | 87.41 ± 0.51 | 87.06 ± 0.10 | 86.90 ± 0.13 | 87.64 | 90.76 | 4.12 |
| | LP | 99.47 ± 0.05 | 86.56 ± 0.34 | 80.81 ± 0.52 | 86.66 ± 0.44 | 64.02 ± 0.78 | 84.74 ± 0.08 | 82.38 ± 0.15 | 83.95 ± 0.04 | 83.57 | 84.18 | 5.75 |
| | SURGICAL-FT | 99.21 ± 0.00 | 99.30 ± 0.15 | 98.09 ± 0.07 | 86.08 ± 0.07 | 60.69 ± 0.81 | 76.45 ± 1.71 | 82.17 ± 1.95 | 85.13 ± 0.03 | 85.89 | 87.86 | 5.88 |
| | LP-FT | 99.52 ± 0.14 | 89.35 ± 5.33 | 84.19 ± 7.61 | 84.80 ± 7.61 | 63.79 ± 0.60 | 85.99 ± 2.52 | 83.71 ± 2.54 | 84.49 ± 1.07 | 84.51 | 85.46 | 5.38 |
| | WISE-FT | 96.03 ± 1.22 | 57.52 ± 3.31 | 70.92 ± 2.97 | 66.52 ± 4.13 | 49.80 ± 0.26 | 80.55 ± 0.06 | 67.69 ± 0.21 | 77.52 ± 0.02 | 70.82 | 70.12 | 7.88 |
| | $L^2$-SP | 99.84 ± 0.03 | 99.99 ± 0.01 | 100.00 ± 0.00 | 97.87 ± 0.09 | **75.36 ± 0.79** | 92.22 ± 0.19 | 92.55 ± 0.60 | 90.00 ± 0.07 | 93.48 | 95.41 | 1.88 |
| | FEATURE-MAP | 99.85 ± 0.02 | **100.00 ± 0.00** | 100.00 ± 0.00 | **99.36 ± 0.08** | 65.70 ± 0.23 | 91.61 ± 0.06 | 91.43 ± 0.15 | 89.49 ± 0.03 | 92.18 | 95.29 | 1.75 |
| | BSS | 99.79 ± 0.05 | 100.00 ± 0.00 | 100.00 ± 0.00 | 98.71 ± 0.03 | 59.16 ± 2.37 | 87.40 ± 0.33 | 88.34 ± 0.15 | 86.95 ± 0.14 | 90.04 | 93.53 | 3.38 |

Table 13: Robust fine-tuning performance on 4 **Regression** datasets (RMSE metrics) in the **Non-Fewshot** setting, evaluated across 3 dataset splits (RANDOM, SCAFFOLD, SIZE) over **GRAPHMAE** and **GRAPHGPS** models. AVG-R, AVG-R* denote the average rank and the rank based on the average normalized performance over all the datasets for each method, respectively. Standard deviations across five replicates are shown in parentheses. We **bold** and underline the best and second-best performances in each scenario.

| SPLIT | METHODS | SELF-SUPERVISED PRE-TRAINING (GRAPHMAE) | | | | | | SUPERVISED PRE-TRAINING (GRAPHGPS) | | | | | |
|---|---|---|---|---|---|---|---|---|---|---|---|---|---|
| | | ESOL | LIPO | MALARIA | CEP | AVG-R | AVG-R* | ESOL | LIPO | MALARIA | CEP | AVG-R | AVG-R* |
| RANDOM | FULL-FT | 0.987 ± 0.013 | 0.734 ± 0.007 | 1.109 ± 0.015 | 1.342 ± 0.015 | 3.00 | 3 | 0.191 ± 0.019 | 0.211 ± 0.012 | 0.955 ± 0.008 | 0.587 ± 0.000 | 4.50 | 4 |
| | LP | 1.394 ± 0.012 | 1.156 ± 0.001 | 1.263 ± 0.002 | 3.079 ± 0.105 | 8.00 | 8 | 0.737 ± 0.005 | 0.877 ± 0.004 | 1.031 ± 0.003 | 1.602 ± 0.006 | 6.50 | 6 |
| | SURGICAL-FT | 1.088 ± 0.011 | 0.883 ± 0.007 | 1.120 ± 0.012 | 1.697 ± 0.012 | 6.25 | 6 | 1.565 ± 0.313 | 2.284 ± 0.179 | 0.800 ± 0.022 | 0.881 ± 0.000 | 6.00 | 7 |
| | LP-FT | **0.953 ± 0.009** | 0.743 ± 0.006 | 1.096 ± 0.009 | **1.322 ± 0.025** | 1.75 | 1 | **0.139 ± 0.016** | 0.197 ± 0.003 | 0.925 ± 0.007 | 0.646 ± 0.087 | 3.25 | 3 |
| | WISE-FT | 1.210 ± 0.032 | 0.846 ± 0.023 | **1.060 ± 0.008** | 1.531 ± 0.030 | 4.50 | 5 | 2.488 ± 0.137 | 1.224 ± 0.007 | 1.187 ± 0.001 | 2.574 ± 0.015 | 7.75 | 8 |
| | L2-SP | 0.995 ± 0.024 | 0.787 ± 0.008 | 1.115 ± 0.006 | 1.363 ± 0.040 | 4.25 | 4 | 0.169 ± 0.009 | 0.194 ± 0.010 | 0.559 ± 0.022 | 0.451 ± 0.036 | 2.00 | 2 |
| | FEATURE-MAP | 1.297 ± 0.007 | 1.080 ± 0.002 | 1.115 ± 0.016 | 1.473 ± 0.018 | 6.25 | 7 | **0.187 ± 0.026** | **0.134 ± 0.008** | **0.243 ± 0.009** | **0.215 ± 0.026** | 1.75 | 1 |
| | BSS | 0.975 ± 0.019 | **0.725 ± 0.011** | 1.100 ± 0.004 | **1.334 ± 0.004** | 2.00 | 2 | 0.177 ± 0.013 | 0.213 ± 0.005 | 0.921 ± 0.013 | 0.651 ± 0.079 | 4.25 | 5 |
| SCAFFOLD | FULL-FT | 1.332 ± 0.015 | 0.808 ± 0.008 | 1.104 ± 0.007 | 1.327 ± 0.017 | 3.50 | 3 | 0.218 ± 0.054 | 0.202 ± 0.022 | 0.929 ± 0.011 | 0.528 ± 0.123 | 4.25 | 4 |
| | LP | 1.703 ± 0.016 | 1.043 ± 0.006 | 1.150 ± 0.003 | 3.102 ± 0.136 | 7.50 | 8 | 0.752 ± 0.006 | 0.849 ± 0.005 | 1.008 ± 0.000 | 1.539 ± 0.009 | 6.75 | 7 |
| | SURGICAL-FT | 1.335 ± 0.005 | 0.884 ± 0.007 | 1.111 ± 0.013 | 1.669 ± 0.022 | 5.50 | 6 | 1.574 ± 0.314 | 0.362 ± 0.013 | 0.818 ± 0.007 | 0.917 ± 0.000 | 5.50 | 6 |
| | LP-FT | **1.312 ± 0.024** | **0.788 ± 0.005** | 1.104 ± 0.006 | 1.318 ± 0.017 | 1.75 | 1 | **0.145 ± 0.020** | 0.181 ± 0.012 | 0.944 ± 0.015 | 0.585 ± 0.036 | 3.25 | 3 |
| | WISE-FT | 1.617 ± 0.031 | 0.891 ± 0.009 | **1.077 ± 0.004** | 1.498 ± 0.034 | 5.00 | 5 | 2.338 ± 0.519 | 1.262 ± 0.006 | 1.220 ± 0.017 | 2.610 ± 0.082 | 8.00 | 8 |
| | L2-SP | 1.329 ± 0.030 | 0.835 ± 0.011 | 1.108 ± 0.011 | 1.325 ± 0.021 | 3.50 | 4 | 0.208 ± 0.037 | 0.183 ± 0.004 | 0.733 ± 0.151 | 0.462 ± 0.050 | 2.75 | 2 |
| | FEATURE-MAP | 1.551 ± 0.013 | 0.994 ± 0.004 | 1.097 ± 0.008 | 1.415 ± 0.004 | 5.00 | 5 | 0.194 ± 0.009 | **0.142 ± 0.004** | **0.327 ± 0.034** | **0.232 ± 0.026** | 1.50 | 1 |
| | BSS | 1.326 ± 0.029 | 0.803 ± 0.013 | 1.104 ± 0.009 | **1.302 ± 0.012** | 2.00 | 2 | 0.181 ± 0.008 | 0.206 ± 0.016 | 0.899 ± 0.024 | 0.622 ± 0.021 | 4.00 | 5 |
| SIZE | FULL-FT | 1.822 ± 0.099 | 0.814 ± 0.013 | 0.908 ± 0.005 | 1.722 ± 0.016 | 3.25 | 3 | 0.192 ± 0.022 | 0.221 ± 0.013 | 0.836 ± 0.044 | 0.474 ± 0.042 | 3.75 | 3 |
| | LP | 2.309 ± 0.030 | 1.024 ± 0.014 | 0.927 ± 0.010 | 3.814 ± 0.175 | 7.75 | 8 | 0.752 ± 0.006 | 0.881 ± 0.004 | 0.996 ± 0.005 | 1.540 ± 0.015 | 6.75 | 7 |
| | SURGICAL-FT | 1.915 ± 0.036 | 0.886 ± 0.013 | 0.925 ± 0.003 | 2.135 ± 0.038 | 6.00 | 5 | 1.589 ± 0.314 | 0.353 ± 0.005 | 0.787 ± 0.018 | 0.943 ± 0.000 | 5.25 | 6 |
| | LP-FT | **1.754 ± 0.075** | **0.795 ± 0.005** | 0.907 ± 0.020 | **1.710 ± 0.010** | 1.75 | 1 | **0.145 ± 0.007** | 0.195 ± 0.007 | 0.902 ± 0.067 | 0.575 ± 0.058 | 3.25 | 4 |
| | WISE-FT | 2.323 ± 0.041 | 0.974 ± 0.016 | 0.895 ± 0.011 | 1.982 ± 0.039 | 5.50 | 7 | 2.264 ± 0.336 | 1.226 ± 0.006 | 1.189 ± 0.002 | 2.683 ± 0.151 | 8.00 | 8 |
| | L2-SP | 1.849 ± 0.041 | 0.849 ± 0.025 | 0.911 ± 0.006 | 1.748 ± 0.041 | 4.50 | 4 | 0.192 ± 0.014 | 0.196 ± 0.009 | 0.787 ± 0.029 | 0.456 ± 0.109 | 3.00 | 2 |
| | FEATURE-MAP | 2.136 ± 0.030 | 1.007 ± 0.015 | **0.891 ± 0.012** | 1.947 ± 0.013 | 4.75 | 6 | 0.209 ± 0.014 | **0.153 ± 0.009** | **0.354 ± 0.007** | **0.227 ± 0.048** | 2.00 | 1 |
| | BSS | 1.808 ± 0.039 | 0.818 ± 0.020 | 0.899 ± 0.006 | 1.712 ± 0.021 | 2.50 | 2 | 0.188 ± 0.019 | 0.211 ± 0.006 | 0.946 ± 0.006 | 0.550 ± 0.000 | 4.00 | 5 |

Table 14: Robust fine-tuning performance on 5 **Classification** datasets (AUC metrics) in the **Fewshot** setting (covering FEWSHOT-50, FEWSHOT-100, FEWSHOT-500), evaluated across 3 dataset splits (RANDOM, SCAFFOLD, SIZE) over **GRAPHMAE** and **GRAPHGPS** models. We **bold** and underline the best and second-best performances in each scenario.

| SPLIT | METHODS | Self-supervised Pre-training (GraphMAE) | | | | | | | | Supervised Pre-training (GraphGPS) | | | | | | | |
|---|---|---|---|---|---|---|---|---|---|---|---|---|---|---|---|---|---|
| | | ClinTox | BBBP | BACE | HIV | SIDER | AVG | AVG-F | AVG-R | ClinTox | BBBP | BACE | HIV | SIDER | AVG | AVG-F | AVG-R |
| | | FEWSHOT-50 | | | | | | | | | | | | | | | |
| RANDOM | FULL-FT | 59.67±3.35 | 83.04±0.39 | 74.97±1.30 | 62.63±0.92 | 52.52±0.19 | 66.57 | 65.76 | 4.20 | 98.37±0.28 | 56.80±1.97 | 59.17±2.03 | 72.16±3.04 | 85.04±0.42 | 74.31 | 72.12 | 3.80 |
| | LP | 57.56±4.09 | 71.69±0.89 | 72.96±0.91 | 48.27±4.06 | 55.09±0.22 | 61.11 | 61.45 | 6.20 | 97.63±0.56 | 56.07±1.71 | 56.41±0.20 | 63.26±0.78 | 82.77±2.22 | 71.23 | 67.48 | 6.40 |
| | SURGICAL-FT | 59.83±2.64 | 78.37±1.06 | 75.25±0.92 | 53.35±0.81 | 54.97±0.43 | 64.35 | 63.35 | 4.40 | 97.73±0.49 | 60.73±3.76 | 58.30±0.86 | 72.63±0.13 | 86.59±0.75 | 75.20 | 73.32 | 7.20 |
| | LP-FT | 60.20±2.14 | 84.54±0.41 | 76.82±0.34 | 62.24±0.58 | 54.41±0.32 | 67.64 | 66.42 | 2.60 | 97.37±0.89 | 55.13±0.23 | 56.28±0.08 | 61.68±3.35 | 83.52±1.15 | 70.80 | 67.16 | 7.20 |
| | WiSE-FT | 63.50±7.72 | 70.77±1.42 | 70.57±1.13 | 58.10±2.35 | 51.23±2.01 | 62.83 | 64.06 | 6.00 | 97.59±0.27 | 53.55±1.95 | 53.17±3.12 | 64.90±3.22 | 83.69±0.21 | 70.58 | 67.38 | 6.80 |
| | $L^2$-SP | 61.02±2.03 | 83.79±0.60 | 74.24±0.96 | 61.58±0.81 | 55.34±0.44 | 67.19 | 65.61 | 3.20 | 98.74±0.40 | 58.95±2.37 | 61.20±3.46 | 72.90±2.19 | 85.15±1.21 | 75.39 | 73.08 | 2.20 |
| | FEATURE-MAP | 59.99±3.80 | 73.57±1.12 | 71.18±2.60 | 48.24±4.14 | 55.85±0.10 | 61.77 | 62.34 | 5.20 | 98.10±0.33 | 59.51±0.56 | 61.65±0.88 | 68.77±2.81 | 82.73±0.22 | 74.15 | 71.05 | 4.20 |
| | BSS | 58.86±3.63 | 83.81±0.57 | 74.38±1.20 | 62.06±0.90 | 54.46±0.56 | 66.71 | 65.10 | 4.20 | 98.43±0.09 | 63.68±3.86 | 59.82±3.70 | 73.10±1.05 | 85.03±0.39 | 76.01 | 73.94 | 2.20 |
| SCAFFOLD | FULL-FT | 55.61±2.60 | 58.53±0.58 | 58.21±7.54 | 45.89±4.20 | 54.86±0.67 | 54.62 | 56.23 | 5.60 | 98.29±0.28 | 52.89±0.45 | 64.90±1.55 | 72.07±2.45 | 84.83±0.05 | 74.60 | 73.93 | 3.80 |
| | LP | 62.76±3.66 | 56.21±1.38 | 56.67±6.74 | 52.12±3.82 | 53.39±0.50 | 56.23 | 55.42 | 6.20 | 97.98±0.48 | 56.21±2.18 | 56.28±0.18 | 63.27±0.78 | 82.52±0.30 | 71.25 | 67.36 | 6.40 |
| | SURGICAL-FT | 63.53±3.11 | 59.33±0.82 | 60.97±3.53 | 52.62±1.46 | 54.94±0.39 | 58.28 | 58.41 | 3.00 | 97.72±0.49 | 61.37±2.90 | 58.30±0.86 | 72.63±0.13 | 86.48±0.70 | 71.30 | 66.83 | 4.60 |
| | LP-FT | 60.62±2.83 | 58.45±0.72 | 59.51±1.11 | 51.87±3.30 | 54.67±0.64 | 57.02 | 57.54 | 5.20 | 97.42±0.81 | 55.14±0.44 | 56.41±0.20 | 61.68±3.35 | 83.49±1.19 | 70.83 | 67.19 | 6.60 |
| | WiSE-FT | 55.45±5.80 | 59.33±0.74 | 67.39±2.69 | 58.03±4.66 | 53.77±0.49 | 58.79 | 57.60 | 4.20 | 98.23±0.05 | 50.43±0.95 | 54.67±0.12 | 66.17±5.35 | 83.73±0.00 | 70.65 | 68.19 | 6.20 |
| | $L^2$-SP | 64.76±2.87 | 59.99±0.63 | 61.49±1.47 | 51.94±3.28 | 54.31±0.86 | 58.50 | 58.60 | 3.60 | 98.72±0.47 | 57.64±2.70 | 59.52±3.80 | 72.30±2.10 | 84.94±0.16 | 74.62 | 72.25 | 2.60 |
| | FEATURE-MAP | 68.84±1.77 | 56.59±1.37 | 64.71±2.65 | 43.90±0.98 | 50.07±0.75 | 56.82 | 57.12 | 5.20 | 98.33±0.07 | 58.93±0.76 | 59.64±0.10 | 68.71±3.16 | 82.75±0.19 | 73.67 | 70.37 | 3.80 |
| | BSS | 60.27±3.40 | 60.16±0.57 | 61.83±1.07 | 62.17±1.89 | 54.35±0.96 | 59.76 | 60.75 | 3.00 | 98.55±0.13 | 59.09±2.38 | 59.36±2.79 | 73.24±1.36 | 85.12±0.23 | 75.07 | 72.57 | 2.00 |
| SIZE | FULL-FT | 53.86±4.15 | 58.43±1.97 | 45.83±8.42 | 51.39±8.97 | 52.27±0.60 | 52.36 | 52.51 | 5.40 | 98.34±0.26 | 55.58±1.28 | 60.71±3.03 | 73.23±1.83 | 85.13±0.22 | 74.60 | 73.02 | 3.20 |
| | LP | 52.46±3.47 | 47.60±7.34 | 51.80±9.61 | 46.50±11.95 | 51.79±0.75 | 50.03 | 50.40 | 6.60 | 97.59±0.58 | 55.30±2.06 | 56.73±0.37 | 63.32±0.78 | 82.71±0.22 | 71.13 | 67.59 | 6.80 |
| | SURGICAL-FT | 53.27±3.82 | 48.97±8.11 | 52.03±9.45 | 52.11±9.11 | 53.37±0.34 | 51.95 | 52.47 | 4.40 | 97.70±0.51 | 61.72±5.14 | 59.07±0.65 | 72.65±0.15 | 86.48±0.70 | 75.52 | 73.62 | 3.40 |
| | LP-FT | 54.43±3.19 | 59.46±1.82 | 40.76±2.04 | 57.05±1.85 | 53.41±0.19 | 53.02 | 54.96 | 3.40 | 97.37±0.85 | 53.59±0.16 | 56.52±0.16 | 61.75±3.29 | 83.17±0.75 | 70.48 | 67.15 | 7.20 |
| | WiSE-FT | 56.43±2.94 | 60.62±3.42 | 51.59±4.93 | 66.93±5.90 | 50.96±1.29 | 57.31 | 56.21 | 3.00 | 98.07±0.31 | 52.64±1.65 | 53.03±2.64 | 66.81±3.77 | 83.77±0.34 | 70.86 | 67.87 | 6.20 |
| | $L^2$-SP | 53.09±0.96 | 58.43±4.43 | 45.90±9.25 | 53.69±4.19 | 52.31±0.70 | 52.68 | 53.03 | 5.20 | 98.79±0.45 | 58.86±2.89 | 61.93±3.45 | 71.87±1.73 | 85.12±0.28 | 75.31 | 72.97 | 2.80 |
| | FEATURE-MAP | 53.75±1.04 | 60.21±7.22 | 46.65±1.64 | 53.42±4.82 | 51.88±0.54 | 53.18 | 53.02 | 4.20 | 98.04±0.40 | 59.55±0.79 | 61.46±0.71 | 68.77±3.28 | 82.85±0.21 | 74.13 | 71.03 | 4.40 |
| | BSS | 58.80±1.49 | 59.13±4.12 | 46.62±8.69 | 53.94±4.11 | 51.87±0.64 | 54.07 | 54.87 | 3.80 | 98.56±0.33 | 62.18±3.81 | 60.51±3.78 | 74.09±3.08 | 85.17±0.45 | 76.10 | 73.81 | 2.00 |
| | | FEWSHOT-100 | | | | | | | | | | | | | | | |
| RANDOM | FULL-FT | 67.65±1.95 | 82.80±0.74 | 79.73±0.72 | 62.47±0.47 | 55.03±0.56 | 69.54 | 69.95 | 4.20 | 99.23±0.16 | 56.70±3.43 | 75.07±2.22 | 90.84±0.37 | 78.15 | 78.28 | 3.00 | |
| | LP | 64.03±2.41 | 72.19±1.10 | 75.93±1.12 | 48.46±3.79 | 58.11±0.51 | 63.74 | 64.78 | 6.40 | 98.43±0.42 | 59.75±2.00 | 56.31±1.19 | 62.48±1.27 | 85.10±0.23 | 72.41 | 69.11 | 6.60 |
| | SURGICAL-FT | 66.99±2.08 | 81.07±0.32 | 79.05±0.49 | 54.93±0.64 | 58.16±0.60 | 68.04 | 68.07 | 5.00 | 99.03±1.28 | 68.12±3.40 | 58.41±3.31 | 74.31±0.11 | 66.85±1.09 | 73.14 | 69.76 | 5.00 |
| | LP-FT | 66.54±1.29 | 84.02±0.63 | 81.49±0.40 | 62.60±3.00 | 57.29±0.49 | 70.39 | 70.21 | 2.80 | 98.80±0.54 | 62.10±3.94 | 52.92±3.41 | 63.75±1.89 | 87.56±3.52 | 73.03 | 71.14 | 5.80 |
| | WiSE-FT | 68.92±3.24 | 81.88±3.16 | 71.01±1.00 | 59.41±1.02 | 52.12±1.56 | 66.87 | 66.78 | 5.40 | 97.95±0.61 | 57.91±3.79 | 50.35±3.20 | 71.04±2.44 | 84.13±1.05 | 72.28 | 71.03 | 7.40 |
| | $L^2$-SP | 68.17±0.71 | 83.52±0.97 | 80.29±0.64 | 61.40±0.73 | 58.85±0.38 | 70.45 | 69.95 | 2.80 | 99.34±0.06 | 72.60±1.56 | 59.28±3.92 | 74.20±2.12 | 90.59±0.50 | 79.20 | 79.13 | 2.60 |
| | FEATURE-MAP | 63.25±1.14 | 73.95±1.04 | 74.90±2.19 | 48.29±4.11 | 58.80±0.21 | 63.84 | 65.33 | 6.40 | 99.39±0.15 | 64.63±3.52 | 65.35±0.64 | 75.47±2.26 | 86.79±0.67 | 78.33 | 75.87 | 2.80 |
| | BSS | 68.22±0.52 | 83.55±0.97 | 80.32±0.67 | 62.24±1.89 | 56.13±0.74 | 70.09 | 70.26 | 3.00 | 99.44±0.10 | 69.97±0.57 | 57.79±2.20 | 72.86±1.91 | 90.80±0.54 | 78.17 | 77.88 | 2.80 |
| SCAFFOLD | FULL-FT | 63.22±3.57 | 60.67±0.99 | 65.72±2.20 | 54.23±2.65 | 54.93±0.84 | 59.75 | 59.61 | 4.80 | 99.20±0.19 | 68.92±0.65 | 61.58±0.58 | 74.59±3.75 | 90.59±0.19 | 78.58 | 78.03 | 2.40 |
| | LP | 61.64±3.21 | 53.87±0.93 | 60.85±1.01 | 53.99±1.84 | 53.02±0.35 | 56.67 | 56.24 | 7.40 | 97.91±0.16 | 61.28±1.82 | 55.52±0.78 | 63.49±1.27 | 85.05±0.00 | 72.45 | 69.61 | 7.00 |
| | SURGICAL-FT | 66.38±1.62 | 58.25±0.90 | 62.95±2.47 | 62.20±1.88 | 55.24±0.47 | 61.00 | 61.13 | 4.00 | 98.91±1.29 | 66.99±2.62 | 58.41±3.30 | 68.72±1.09 | | 73.31 | 70.04 | 5.20 |
| | LP-FT | 65.08±3.59 | 60.15±0.20 | 66.58±0.96 | 57.03±3.48 | 54.12±0.52 | 60.59 | 60.75 | 4.60 | 98.72±0.46 | 61.52±2.16 | 52.92±3.41 | 66.12±5.21 | 87.37±3.31 | 73.33 | 71.67 | 5.80 |
| | WiSE-FT | 53.83±2.78 | 64.13±1.64 | 72.12±1.43 | 57.64±4.40 | 55.64±2.15 | 60.67 | 59.14 | 2.80 | 98.25±0.08 | 57.54±5.22 | 50.76±0.29 | 68.45±2.94 | 83.68±0.17 | 71.74 | 69.89 | 7.00 |
| | $L^2$-SP | 66.91±1.79 | 60.77±1.57 | 66.02±1.53 | 54.34±2.25 | 54.72±1.16 | 60.55 | 60.50 | 3.80 | 99.33±0.02 | 69.14±0.93 | 59.04±0.56 | 72.73±4.03 | 90.92±0.50 | 78.23 | 77.60 | 2.60 |
| | FEATURE-MAP | 68.84±1.56 | 55.98±0.58 | 64.15±2.87 | 50.87±2.38 | 49.55±0.88 | 57.88 | 57.00 | 6.00 | 99.38±0.10 | 65.01±1.81 | 64.95±0.51 | 74.25±1.63 | 87.16±0.06 | 78.15 | 75.47 | 3.40 |
| | BSS | 67.11±2.10 | 60.54±1.13 | 66.61±1.12 | 60.74±0.93 | 55.06±1.14 | 62.01 | 62.63 | 2.60 | 99.43±0.08 | 68.86±4.35 | 57.14±1.54 | 75.44±0.60 | 90.53±0.15 | 78.28 | 78.28 | 2.60 |
| SIZE | FULL-FT | 55.01±3.57 | 66.52±1.39 | 51.73±2.47 | 54.13±8.59 | 53.93±0.76 | 56.26 | 54.36 | 3.80 | 99.01±0.18 | 68.31±2.98 | 55.43±1.46 | 75.19±1.58 | 90.82±0.55 | 77.75 | 78.11 | 3.60 |
| | LP | 52.73±3.21 | 49.27±5.99 | 47.22±6.09 | 46.39±11.18 | 51.72±0.76 | 49.47 | 49.40 | 7.40 | 98.11±0.43 | 59.75±2.07 | 56.08±1.29 | 62.54±1.28 | 85.06±0.24 | 72.37 | 69.12 | 6.40 |
| | SURGICAL-FT | 58.30±3.52 | 52.34±6.18 | 49.29±5.93 | 51.50±12.55 | 53.47±0.71 | 52.98 | 52.44 | 5.20 | 98.00±1.31 | 67.07±5.56 | 59.41±3.37 | 74.94±0.41 | 86.72±1.09 | 73.63 | 70.24 | 5.60 |
| | LP-FT | 54.19±2.32 | 67.66±1.06 | 54.39±2.27 | 58.09±1.24 | 55.25±0.33 | 57.92 | 55.91 | 2.60 | 98.71±0.46 | 61.15±2.61 | 52.47±3.57 | 65.82±4.88 | 87.35±3.28 | 73.10 | 71.44 | 5.80 |
| | WiSE-FT | 54.89±5.22 | 65.76±1.41 | 48.32±2.36 | 67.43±6.52 | 54.95±0.94 | 56.69 | 56.32 | 4.80 | 98.24±0.55 | 55.85±5.93 | 50.18±0.23 | 67.31±4.78 | 83.63±0.56 | 71.04 | 68.93 | 7.20 |
| | $L^2$-SP | 53.99±1.00 | 66.39±3.08 | 54.50±3.14 | 54.52±7.69 | 54.34±1.20 | 56.75 | 54.45 | 3.60 | 99.33±0.15 | 64.17±3.16 | 60.98±4.23 | 74.54±1.11 | 90.85±0.25 | 77.97 | 76.52 | 3.20 |
| | FEATURE-MAP | 50.62±1.90 | 58.47±9.57 | 46.18±1.57 | 52.40±5.59 | 51.81±0.64 | 51.90 | 51.61 | 6.80 | 99.35±0.13 | 71.24±3.08 | 64.94±0.87 | 75.86±2.63 | 86.78±0.70 | 79.63 | 77.96 | 2.00 |
| | BSS | 58.71±1.44 | 67.67±2.91 | 54.89±3.17 | 54.60±7.72 | 54.33±1.18 | 58.04 | 56.07 | 1.80 | 99.34±0.15 | 72.34±4.55 | 60.22±3.48 | 75.41±1.92 | 90.76±0.35 | 79.61 | 79.50 | 2.20 |
| | | FEWSHOT-500 | | | | | | | | | | | | | | | |
| RANDOM | FULL-FT | 78.63±0.77 | 91.08±1.35 | 85.62±0.30 | 70.55±0.32 | 59.68±0.36 | 77.11 | 78.27 | 4.40 | 99.82±0.02 | 100.00±0.00 | 100.00±0.00 | 79.94±0.83 | 94.71±0.21 | 94.89 | 98.18 | 3.40 |
| | LP | 72.34±2.23 | 79.79±1.23 | 75.57±1.04 | 54.42±2.54 | 61.10±0.33 | 68.64 | 69.67 | 7.00 | 99.59±0.05 | 90.08±0.21 | 82.16±0.33 | 76.72±0.97 | 86.35±0.15 | 86.98 | 86.20 | 7.00 |
| | SURGICAL-FT | 79.09±0.81 | 85.22±0.36 | 83.73±0.94 | 65.78±0.56 | 61.10±0.47 | 74.99 | 76.21 | 5.00 | 99.68±0.09 | 100.00±0.00 | 99.87±0.07 | 79.91±1.10 | 79.82±1.10 | 91.86 | 93.15 | 5.40 |
| | LP-FT | 80.52±1.76 | 91.82±0.25 | 86.02±0.20 | 69.28±0.65 | 61.10±0.38 | 77.75 | 78.61 | 2.40 | 99.55±0.11 | 100.00±0.00 | 87.85±8.59 | 77.83±1.54 | 89.77±5.03 | 91.03 | 92.43 | 5.20 |
| | WiSE-FT | 78.34±3.82 | 91.54±0.76 | 84.49±0.56 | 61.15±1.37 | 63.77±1.03 | 75.86 | 75.53 | 4.20 | 97.05±0.38 | 73.40±3.17 | 82.91±7.38 | 75.12±4.71 | 80.59±0.10 | 81.81 | 79.54 | 7.60 |
| | $L^2$-SP | 78.56±0.91 | 91.38±0.46 | 85.81±0.40 | 68.73±0.18 | 61.34±0.00 | 77.16 | 77.70 | 3.80 | 99.86±0.02 | 100.00±0.00 | 100.00±0.00 | 80.36±2.11 | 95.12±0.15 | 95.07 | 98.33 | 2.60 |
| | FEATURE-MAP | 69.96±1.65 | 81.31±0.48 | 71.65±0.61 | 58.54±1.57 | 61.40±0.19 | 68.57 | 67.67 | 6.40 | 99.88±0.02 | 100.00±0.00 | 100.00±0.00 | 80.75±0.35 | 92.33±0.07 | 94.59 | 97.40 | 2.60 |
| | BSS | 79.17±0.93 | 91.98±0.48 | 85.85±0.41 | 69.74±0.11 | 60.32±0.51 | 77.41 | 78.25 | 2.40 | 99.83±0.01 | 100.00±0.00 | 100.00±0.00 | 80.79±2.04 | 95.31±0.12 | 95.19 | 98.38 | 2.20 |
| SCAFFOLD | FULL-FT | 68.64±0.79 | 88.65±0.62 | 77.69±0.21 | 66.32±1.81 | 57.55±0.33 | 67.77 | 67.87 | 4.20 | 99.83±0.02 | 100.00±0.00 | 100.00±0.00 | 78.83±1.28 | 94.78±0.08 | 94.69 | 98.20 | 3.80 |
| | LP | 67.38±2.22 | 80.02±0.77 | 72.66±5.53 | 60.14±4.14 | 58.74±1.34 | 61.79 | 60.94 | 6.40 | 99.65±0.03 | 99.96±0.21 | 82.16±0.33 | 89.96±0.21 | 82.16±0.33 | 91.09 | 91.99 | 6.50 |
| | SURGICAL-FT | 70.31±2.21 | 65.27±0.39 | 74.86±1.30 | 70.52±1.05 | 61.99±0.40 | 68.59 | 68.70 | 3.00 | 99.65±0.07 | 100.00±0.00 | 99.86±0.07 | 79.34±0.10 | 79.71±1.12 | 91.71 | 93.03 | 5.80 |
| | LP-FT | 65.58±1.93 | 69.05±0.77 | 78.48±0.58 | 70.22±0.49 | 55.89±0.75 | 67.84 | 68.28 | 4.60 | 99.68±0.12 | 100.00±0.00 | 87.85±8.59 | 75.98±1.99 | 88.57±3.39 | 90.42 | 92.03 | 5.70 |
| | WiSE-FT | 68.48±3.60 | 65.58±1.56 | 82.78±0.77 | 58.90±2.63 | 57.28±0.75 | 66.60 | 64.32 | 5.00 | 97.82±0.01 | 69.60±8.28 | 82.47±6.77 | 77.14±1.59 | 82.47±0.07 | 81.90 | 80.69 | 7.60 |
| | $L^2$-SP | 68.16±0.88 | 59.42±0.29 | 88.25±1.93 | 67.01±2.26 | 56.57±0.43 | 63.88 | 64.86 | 6.20 | 99.89±0.03 | 100.00±0.00 | 100.00±0.00 | 80.44±0.99 | 95.22±0.01 | 94.51 | 97.37 | 2.80 |
| | BSS | 68.59±1.15 | 69.09±0.57 | 78.85±0.93 | 66.05±2.20 | 58.73±0.39 | 68.26 | 67.91 | 3.20 | 99.85±0.04 | 100.00±0.00 | 100.00±0.00 | 100.00±0.00 | 95.77±0.16 | 99.12 | 99.95 | 2.00 |
| SIZE | FULL-FT | 65.78±1.28 | 83.11±0.77 | 49.15±1.50 | 58.35±9.96 | 52.46±1.33 | 61.77 | 59.59 | 4.00 | 99.80±0.02 | 100.00±0.00 | 100.00±0.00 | 79.70±0.81 | 94.74±0.27 | 94.85 | 97.35 | 3.40 |
| | LP | 58.59±2.86 | 60.74±5.06 | 48.22±1.25 | 46.39±11.18 | 51.72±0.76 | 52.94 | 52.63 | 7.40 | 99.69±0.00 | 89.85±0.27 | 82.12±0.35 | 76.74±0.97 | 86.31±0.15 | 86.92 | 86.30 | 6.80 |
| | SURGICAL-FT | 65.88±1.23 | 72.86±1.29 | 47.48±4.55 | 57.44±9.55 | 52.61±0.51 | 59.28 | 58.40 | 4.80 | 99.66±0.10 | 100.00±0.00 | 99.87±0.08 | 79.16±0.08 | 79.72±1.10 | 91.68 | 92.73 | 5.50 |
| | LP-FT | 66.09±1.44 | 82.96±0.52 | 50.17±0.69 | 63.07±0.97 | 52.25±0.55 | 62.91 | 61.08 | 2.60 | 99.66±0.10 | 100.00±0.00 | 87.85±8.59 | 77.93±2.10 | 86.90±1.04 | 90.47 | 91.23 | 5.10 |
| | WiSE-FT | 57.72±2.58 | 77.31±1.56 | 60.42±2.45 | 68.17±2.47 | 51.52±0.50 | 63.03 | 62.33 | 4.60 | 97.38±0.35 | 76.91±5.59 | 77.56±2.56 | 74.71±1.58 | 80.55±0.13 | 81.42 | 79.11 | 7.80 |
| | $L^2$-SP | 65.91±2.13 | 82.22±0.63 | 49.40±0.87 | 60.24±2.10 | 52.79±0.72 | 62.11 | 60.26 | 3.20 | 99.84±0.02 | 100.00±0.00 | 100.00±0.00 | 80.57±2.27 | 95.02±0.14 | 95.09 | 97.49 | 2.60 |
| | FEATURE-MAP | 60.84±1.37 | 63.60±6.18 | 44.07±0.77 | 49.33±7.05 | 51.80±0.59 | 53.93 | 53.97 | 6.80 | 99.86±0.01 | 100.00±0.00 | 100.00±0.00 | 80.74±0.61 | 92.32±0.07 | 94.58 | 96.69 | 2.50 |
| | BSS | 66.64±2.47 | 83.60±0.32 | 49.73±0.59 | 62.63±1.27 | 52.24±0.98 | 62.97 | 61.12 | 2.60 | 99.86±0.03 | 100.00±0.00 | 100.00±0.00 | 80.26±0.75 | 95.81±0.17 | 95.19 | 97.71 | 2.30 |

Table 15: Robust fine-tuning performance on 4 **Regression** datasets (RMSE metrics) in the **Fewshot** setting (covering FEWSHOT-50, FEWSHOT-100, and FEWSHOT-500), evaluated across 3 dataset splits (RANDOM, SCAFFOLD, SIZE) over **GraphMAE** and **GraphGPS** models. AVG-R, AVG-R* denote the average rank and the rank based on the average normalized performance over all the datasets for each evavluated method, respectively. Standard deviations across five replicates are shown in parentheses. We **bold** and underline the best and second-best performances in each scenario.

| SPLIT | METHODS | SELF-SUPERVISED PRE-TRAINING (GraphMAE) | | | | | | SUPERVISED PRE-TRAINING (GraphGPS) | | | | | |
|---|---|---|---|---|---|---|---|---|---|---|---|---|---|
| | | ESOL | LIPO | MALARIA | CEP | AVG-R | AVG-R* | ESOL | LIPO | MALARIA | CEP | AVG-R | AVG-R* |
| **FEWSHOT-50** | | | | | | | | | | | | | |
| RANDOM | FULL-FT | 1.432±0.019 | 1.328±0.051 | 1.297±0.015 | 2.927±0.226 | 4.25 | 6 | 0.896±0.015 | 1.221±0.016 | 1.192±0.017 | 2.072±0.050 | 4.00 | 4 |
| | LP | 1.646±0.027 | 1.395±0.076 | 1.334±0.009 | 4.133±0.372 | 7.50 | 8 | 1.183±0.012 | 1.223±0.007 | 1.193±0.009 | 2.249±0.016 | 6.00 | 6 |
| | SURGICAL-FT | 1.497±0.017 | 1.303±0.051 | 1.309±0.017 | 3.300±0.406 | 5.00 | 7 | 3.573±0.101 | 2.168±0.089 | 1.203±0.010 | 4.263±0.096 | 7.75 | 8 |
| | LP-FT | 1.386±0.022 | 1.217±0.021 | 1.399±0.033 | 2.840±0.226 | 3.75 | 5 | 1.037±0.029 | 1.199±0.041 | 1.178±0.014 | 2.156±0.145 | 3.50 | 5 |
| | WiSE-FT | 1.622±0.053 | 1.343±0.010 | 1.248±0.008 | 2.385±0.026 | 3.75 | 2 | 0.847±0.027 | 1.224±0.007 | 1.180±0.019 | 2.574±0.053 | 6.00 | 7 |
| | L2-SP | 1.444±0.027 | 1.354±0.052 | 1.294±0.005 | 2.315±0.106 | 3.75 | 1 | 0.881±0.037 | 1.203±0.022 | 1.184±0.013 | 2.091±0.049 | 2.50 | 3 |
| | FEATURE-MAP | 1.655±0.027 | 1.312±0.020 | 1.278±0.008 | 2.363±0.127 | 3.75 | 3 | 0.882±0.059 | 1.173±0.013 | 1.193±0.006 | 2.050±0.029 | 2.75 | 2 |
| | BSS | 1.439±0.029 | 1.351±0.051 | 1.294±0.005 | 2.682±0.115 | 4.00 | 4 | 0.822±0.024 | 1.204±0.021 | 1.189±0.011 | 2.109±0.036 | 3.25 | 1 |
| SCAFFOLD | FULL-FT | 1.717±0.028 | 1.214±0.051 | 1.169±0.005 | 2.612±0.178 | 5.25 | 6 | 0.859±0.065 | 1.219±0.025 | 1.426±0.243 | 2.100±0.053 | 4.00 | 3 |
| | LP | 2.209±0.039 | 1.183±0.045 | 1.170±0.004 | 2.656±0.048 | 6.00 | 8 | 1.213±0.015 | 1.223±0.006 | 1.194±0.012 | 2.261±0.019 | 5.50 | 6 |
| | SURGICAL-FT | 1.834±0.031 | 1.198±0.049 | 1.166±0.001 | 3.142±0.589 | 5.25 | 7 | 3.589±0.101 | 2.168±0.089 | 1.204±0.010 | 4.261±0.096 | 7.00 | 8 |
| | LP-FT | 1.642±0.026 | 1.147±0.038 | 1.300±0.061 | 2.879±0.264 | 4.25 | 4 | 1.053±0.180 | 1.198±0.043 | 1.174±0.019 | 2.633±0.625 | 4.00 | 4 |
| | WiSE-FT | 2.221±0.047 | 1.175±0.016 | 1.166±0.002 | 2.326±0.031 | 4.00 | 3 | 1.020±0.045 | 1.259±0.027 | 1.238±0.012 | 2.123±0.035 | 6.00 | 7 |
| | L2-SP | 1.718±0.053 | 1.200±0.053 | 1.202±0.062 | 2.366±0.059 | 5.00 | 5 | 0.897±0.058 | 1.196±0.030 | 1.205±0.032 | 2.105±0.032 | 3.00 | 2 |
| | FEATURE-MAP | 2.197±0.075 | 1.148±0.023 | 1.163±0.003 | 2.400±0.175 | 3.00 | 1 | 0.898±0.010 | 1.200±0.013 | 1.229±0.014 | 2.115±0.031 | 4.25 | 5 |
| | BSS | 1.712±0.056 | 1.168±0.050 | 1.168±0.002 | 2.551±0.121 | 3.25 | 2 | 0.861±0.024 | 1.208±0.016 | 1.186±0.019 | 2.081±0.037 | 2.25 | 1 |
| SIZE | FULL-FT | 2.654±0.075 | 1.557±0.093 | 0.943±0.026 | 2.550±0.053 | 4.25 | 5 | 0.886±0.054 | 1.209±0.011 | 1.173±0.007 | 2.098±0.048 | 3.75 | 4 |
| | LP | 2.818±0.087 | 1.676±0.115 | 0.963±0.030 | 5.414±0.036 | 7.00 | 8 | 1.176±0.011 | 1.232±0.007 | 1.181±0.009 | 2.248±0.016 | 6.75 | 7 |
| | SURGICAL-FT | 2.658±0.088 | 1.641±0.114 | 0.929±0.027 | 3.423±0.550 | 5.75 | 6 | 3.589±0.101 | 2.168±0.089 | 1.204±0.010 | 4.258±0.095 | 8.00 | 8 |
| | LP-FT | 2.440±0.056 | 1.422±0.111 | 1.166±0.053 | 2.339±0.049 | 2.75 | 1 | 1.049±0.186 | 1.204±0.047 | 1.174±0.016 | 2.167±0.129 | 5.25 | 6 |
| | WiSE-FT | 3.050±0.087 | 1.513±0.049 | 0.969±0.001 | 3.223±0.224 | 5.75 | 7 | 1.045±0.054 | 1.230±0.015 | 1.171±0.025 | 2.126±0.039 | 4.50 | 5 |
| | L2-SP | 2.606±0.085 | 1.614±0.112 | 0.914±0.016 | 2.466±0.079 | 3.00 | 2 | 0.851±0.036 | 1.194±0.015 | 1.169±0.005 | 2.101±0.022 | 2.00 | 1 |
| | FEATURE-MAP | 2.630±0.036 | 1.697±0.080 | 0.920±0.007 | 2.408±0.057 | 4.00 | 4 | 0.867±0.035 | 1.180±0.014 | 1.183±0.007 | 2.039±0.022 | 3.00 | 3 |
| | BSS | 2.579±0.066 | 1.613±0.110 | 0.926±0.018 | 2.580±0.157 | 3.50 | 3 | 0.844±0.007 | 1.200±0.028 | 1.171±0.033 | 2.104±0.032 | 2.75 | 2 |
| **FEWSHOT-100** | | | | | | | | | | | | | |
| RANDOM | FULL-FT | 1.304±0.041 | 1.239±0.032 | 1.289±0.003 | 3.028±0.310 | 3.25 | 1 | 0.412±0.016 | 1.061±0.017 | 1.140±0.016 | 1.976±0.031 | 3.00 | 3 |
| | LP | 1.609±0.032 | 1.285±0.043 | 1.334±0.009 | 4.562±0.047 | 7.50 | 8 | 0.902±0.037 | 1.185±0.007 | 1.174±0.004 | 2.239±0.010 | 7.25 | 7 |
| | SURGICAL-FT | 1.356±0.022 | 1.219±0.016 | 1.298±0.008 | 3.100±0.805 | 4.50 | 5 | 3.371±0.120 | 1.925±0.045 | 1.162±0.013 | 4.076±0.046 | 7.50 | 8 |
| | LP-FT | 1.310±0.021 | 1.226±0.021 | 1.374±0.045 | 3.241±0.438 | 4.75 | 6 | 0.735±0.230 | 1.144±0.049 | 1.153±0.030 | 2.158±0.100 | 5.50 | 6 |
| | WiSE-FT | 1.600±0.051 | 1.324±0.013 | 1.245±0.017 | 2.294±0.024 | 4.75 | 7 | 0.671±0.104 | 1.068±0.049 | 1.159±0.036 | 2.017±0.095 | 5.00 | 5 |
| | L2-SP | 1.323±0.034 | 1.253±0.029 | 1.276±0.014 | 2.271±0.065 | 3.25 | 2 | 0.405±0.034 | 1.055±0.022 | 1.129±0.016 | 1.951±0.045 | 1.75 | 1 |
| | FEATURE-MAP | 1.526±0.030 | 1.243±0.027 | 1.276±0.004 | 2.271±0.116 | 3.75 | 3 | 0.422±0.021 | 1.014±0.006 | 1.170±0.013 | 1.883±0.012 | 3.25 | 4 |
| | BSS | 1.322±0.033 | 1.251±0.028 | 1.293±0.006 | 2.541±0.128 | 4.25 | 4 | 0.405±0.060 | 1.050±0.003 | 1.147±0.014 | 1.980±0.018 | 2.75 | 2 |
| SCAFFOLD | FULL-FT | 1.695±0.045 | 1.168±0.030 | 1.167±0.003 | 3.087±0.765 | 4.50 | 2 | 0.497±0.045 | 1.125±0.034 | 1.215±0.015 | 2.036±0.073 | 4.75 | 6 |
| | LP | 2.045±0.044 | 1.211±0.064 | 1.173±0.004 | 4.579±0.037 | 7.50 | 8 | 0.971±0.036 | 1.185±0.008 | 1.174±0.004 | 2.247±0.005 | 6.25 | 5 |
| | SURGICAL-FT | 1.693±0.019 | 1.146±0.017 | 1.169±0.003 | 3.226±0.563 | 4.50 | 1 | 3.386±0.120 | 1.927±0.041 | 1.162±0.013 | 4.073±0.048 | 7.00 | 8 |
| | LP-FT | 1.626±0.016 | 1.123±0.011 | 1.312±0.023 | 2.782±0.364 | 3.75 | 5 | 0.730±0.236 | 1.136±0.029 | 1.154±0.029 | 2.167±0.117 | 4.75 | 3 |
| | WiSE-FT | 2.069±0.066 | 1.205±0.014 | 1.158±0.008 | 2.244±0.068 | 4.25 | 7 | 1.069±0.332 | 1.124±0.023 | 1.228±0.016 | 2.143±0.115 | 6.00 | 7 |
| | L2-SP | 1.679±0.045 | 1.201±0.048 | 1.168±0.003 | 2.327±0.030 | 3.50 | 4 | 0.497±0.060 | 1.098±0.015 | 1.155±0.022 | 2.031±0.061 | 3.25 | 2 |
| | FEATURE-MAP | 1.964±0.034 | 1.164±0.029 | 1.164±0.001 | 2.341±0.095 | 3.50 | 6 | 0.489±0.040 | 1.039±0.014 | 1.185±0.010 | 2.008±0.022 | 2.75 | 4 |
| | BSS | 1.681±0.043 | 1.191±0.046 | 1.169±0.004 | 2.566±0.149 | 4.50 | 3 | 0.396±0.010 | 1.054±0.033 | 1.139±0.005 | 1.972±0.010 | 1.25 | 1 |
| SIZE | FULL-FT | 2.414±0.081 | 1.283±0.070 | 0.911±0.008 | 2.677±0.139 | 3.00 | 1 | 0.431±0.059 | 1.039±0.026 | 1.118±0.014 | 1.968±0.056 | 3.25 | 4 |
| | LP | 2.859±0.078 | 1.493±0.115 | 0.951±0.030 | 5.420±0.033 | 7.50 | 8 | 0.901±0.043 | 1.192±0.007 | 1.163±0.004 | 2.236±0.011 | 7.25 | 7 |
| | SURGICAL-FT | 2.537±0.059 | 1.301±0.074 | 0.909±0.003 | 3.707±0.589 | 4.75 | 6 | 3.386±0.120 | 1.933±0.038 | 1.151±0.012 | 4.077±0.040 | 7.50 | 8 |
| | LP-FT | 2.217±0.047 | 1.146±0.022 | 1.065±0.020 | 2.562±0.076 | 3.50 | 2 | 0.733±0.232 | 1.166±0.029 | 1.147±0.022 | 2.138±0.123 | 5.50 | 6 |
| | WiSE-FT | 2.507±0.098 | 1.297±0.038 | 0.904±0.002 | 2.823±0.031 | 3.75 | 3 | 0.708±0.099 | 1.079±0.040 | 1.147±0.040 | 1.987±0.050 | 4.75 | 5 |
| | L2-SP | 2.442±0.047 | 1.362±0.082 | 0.916±0.009 | 2.451±0.093 | 4.50 | 5 | 0.409±0.024 | 1.037±0.030 | 1.125±0.016 | 1.942±0.032 | 2.00 | 1 |
| | FEATURE-MAP | 2.716±0.026 | 1.551±0.085 | 0.912±0.003 | 2.424±0.039 | 5.00 | 7 | 0.419±0.016 | 1.009±0.013 | 1.160±0.010 | 1.886±0.031 | 3.00 | 3 |
| | BSS | 2.434±0.046 | 1.358±0.084 | 0.912±0.005 | 2.533±0.103 | 3.75 | 3 | 0.387±0.020 | 1.038±0.021 | 1.136±0.013 | 1.967±0.023 | 2.50 | 2 |
| **FEWSHOT-500** | | | | | | | | | | | | | |
| RANDOM | FULL-FT | 1.042±0.017 | 1.023±0.022 | 1.290±0.004 | 1.958±0.038 | 4.00 | 5 | 0.135±0.019 | 0.070±0.005 | 0.787±0.009 | 1.554±0.044 | 3.25 | 3 |
| | LP | 1.487±0.011 | 1.233±0.019 | 1.331±0.012 | 4.602±0.019 | 8.00 | 8 | 0.769±0.018 | 0.854±0.008 | 1.035±0.001 | 1.941±0.004 | 6.00 | 6 |
| | SURGICAL-FT | 1.164±0.010 | 1.127±0.007 | 1.240±0.011 | 3.577±0.498 | 5.00 | 7 | 2.376±0.207 | 0.806±0.037 | 0.803±0.010 | 3.058±0.054 | 6.75 | 7 |
| | LP-FT | 0.995±0.010 | 0.975±0.007 | 1.310±0.019 | 2.004±0.056 | 3.75 | 4 | 0.545±0.293 | 0.605±0.352 | 0.793±0.018 | 1.566±0.027 | 5.50 | 5 |
| | WiSE-FT | 1.251±0.029 | 0.976±0.010 | 1.231±0.016 | 1.975±0.017 | 3.25 | 2 | 2.512±0.245 | 1.563±0.200 | 1.197±0.017 | 2.177±0.063 | 7.75 | 8 |
| | L2-SP | 1.048±0.043 | 1.036±0.009 | 1.241±0.007 | 1.886±0.032 | 3.75 | 1 | 0.141±0.043 | 0.080±0.026 | 0.781±0.010 | 1.549±0.022 | 2.75 | 2 |
| | FEATURE-MAP | 1.340±0.007 | 1.202±0.014 | 1.241±0.007 | 1.992±0.013 | 5.75 | 6 | 0.155±0.021 | 0.104±0.005 | 0.778±0.004 | 1.565±0.028 | 3.25 | 4 |
| | BSS | 1.031±0.013 | 1.020±0.006 | 1.272±0.007 | 1.896±0.034 | 3.00 | 3 | 0.129±0.013 | 0.018±0.004 | 0.779±0.007 | 1.543±0.028 | 1.25 | 1 |
| SCAFFOLD | FULL-FT | 1.406±0.016 | 0.945±0.021 | 1.199±0.025 | 2.057±0.072 | 4.75 | 5 | 0.145±0.023 | 0.072±0.005 | 0.776±0.006 | 1.564±0.033 | 2.50 | 3 |
| | LP | 1.849±0.028 | 1.102±0.019 | 1.182±0.007 | 4.607±0.020 | 7.00 | 8 | 0.771±0.018 | 0.854±0.008 | 1.035±0.001 | 1.941±0.004 | 6.50 | 6 |
| | SURGICAL-FT | 1.436±0.010 | 1.020±0.006 | 1.156±0.010 | 2.874±0.652 | 5.00 | 6 | 2.377±0.207 | 0.805±0.041 | 0.802±0.011 | 3.053±0.051 | 6.50 | 6 |
| | LP-FT | 1.354±0.011 | 0.940±0.012 | 1.278±0.044 | 2.052±0.053 | 3.75 | 4 | 0.544±0.293 | 0.605±0.352 | 0.793±0.010 | 1.803±0.196 | 5.25 | 5 |
| | WiSE-FT | 1.707±0.029 | 1.028±0.025 | 1.125±0.008 | 1.906±0.020 | 3.50 | 3 | 2.476±0.626 | 1.459±0.258 | 1.207±0.030 | 2.173±0.061 | 7.75 | 8 |
| | L2-SP | 1.413±0.045 | 0.943±0.022 | 1.156±0.012 | 1.931±0.054 | 3.25 | 2 | 0.137±0.017 | 0.070±0.009 | 0.782±0.005 | 1.524±0.014 | 2.25 | 2 |
| | FEATURE-MAP | 1.880±0.021 | 1.081±0.006 | 1.129±0.006 | 1.992±0.008 | 5.25 | 7 | 0.163±0.010 | 0.111±0.002 | 0.786±0.005 | 1.592±0.013 | 4.00 | 4 |
| | BSS | 1.404±0.042 | 0.941±0.019 | 1.199±0.029 | 1.926±0.041 | 3.00 | 1 | 0.127±0.015 | 0.068±0.004 | 0.777±0.008 | 1.513±0.007 | 1.25 | 1 |
| SIZE | FULL-FT | 2.102±0.080 | 0.968±0.032 | 0.955±0.031 | 2.283±0.060 | 3.50 | 4 | 0.142±0.049 | 0.070±0.003 | 0.723±0.008 | 1.548±0.011 | 3.00 | 3 |
| | LP | 2.486±0.040 | 1.140±0.046 | 0.968±0.027 | 5.452±0.018 | 7.50 | 8 | 0.771±0.018 | 0.855±0.009 | 1.008±0.004 | 1.938±0.004 | 6.50 | 6 |
| | SURGICAL-FT | 2.142±0.062 | 0.982±0.014 | 0.949±0.032 | 3.765±0.499 | 4.50 | 7 | 2.384±0.212 | 0.812±0.042 | 0.745±0.011 | 3.070±0.035 | 6.50 | 7 |
| | LP-FT | 2.003±0.037 | 0.889±0.017 | 0.985±0.033 | 2.339±0.049 | 3.75 | 3 | 0.550±0.287 | 0.812±0.042 | 0.740±0.027 | 1.533±0.003 | 4.25 | 5 |
| | WiSE-FT | 2.302±0.057 | 1.040±0.015 | 0.906±0.003 | 2.437±0.032 | 3.75 | 5 | 2.559±0.295 | 1.599±0.242 | 1.196±0.027 | 2.189±0.053 | 7.75 | 8 |
| | L2-SP | 2.030±0.059 | 1.012±0.030 | 0.951±0.030 | 2.208±0.030 | 3.25 | 2 | 0.124±0.013 | 0.073±0.006 | 0.705±0.008 | 1.533±0.014 | 2.25 | 2 |
| | FEATURE-MAP | 2.253±0.017 | 1.174±0.023 | 0.908±0.001 | 2.341±0.027 | 5.25 | 5 | 0.157±0.020 | 0.103±0.005 | 0.705±0.008 | 1.553±0.014 | 3.50 | 4 |
| | BSS | 1.980±0.051 | 0.989±0.025 | 0.956±0.041 | 2.237±0.058 | 3.25 | 1 | 0.126±0.013 | 0.064±0.004 | 0.710±0.014 | 1.551±0.028 | 2.25 | 2 |

Table 16: DWiSE-FT performance on 4 **Regression** datasets (RMSE metrics) in the **Fewshot** setting with 50,100, 500 samples, evaluated across 3 dataset splits (RANDOM, SCAFFOLD, SIZE) given **MOLE-BERT** model. AVG-R denote the average rank. Standard deviations across five replicates are shown in parentheses. We **bold** and underline the best and second-best performances in each scenario.

| SPLIT | METHODS | FEWSHOT 50 | | | | | FEWSHOT 100 | | | | | FEWSHOT 500 | | | | |
|---|---|---|---|---|---|---|---|---|---|---|---|---|---|---|---|---|
| | | ESOL | LIPO | MALARIA | CEP | AVG | ESOL | LIPO | MALARIA | CEP | AVG | ESOL | LIPO | MALARIA | CEP | AVG |
| RANDOM | WiSE-FT | 1.384±0.047 | 1.212±0.020 | 1.276±0.007 | 2.410±0.051 | 3.75 | 1.189±0.030 | 1.142±0.025 | 1.256±0.006 | 2.211±0.028 | 3.00 | 0.995±0.010 | 0.855±0.011 | 1.193±0.003 | 1.893±0.021 | 3.75 |
| | L²-SP | 1.372±0.029 | 1.196±0.019 | 1.277±0.006 | 2.280±0.031 | 3.00 | 1.161±0.016 | 1.149±0.007 | 1.260±0.004 | 2.131±0.014 | 3.25 | 0.878±0.026 | 0.806±0.007 | 1.192±0.004 | 1.893±0.018 | 1.50 |
| | Top | 1.329±0.021 | 1.164±0.010 | 1.271±0.007 | 2.275±0.022 | 1.25 | 1.120±0.038 | 1.139±0.017 | 1.256±0.004 | 2.131±0.011 | 1.00 | 0.878±0.026 | 0.806±0.007 | 1.192±0.004 | 1.862±0.010 | 1.00 |
| | DWiSE-FT | 1.378±0.055 | 1.189±0.020 | 1.273±0.009 | 2.222±0.059 | 2.00 | 1.132±0.025 | 1.138±0.028 | 1.256±0.004 | 2.129±0.020 | 1.25 | 0.918±0.012 | 0.818±0.013 | 1.192±0.004 | 1.865±0.030 | 2.25 |
| SCAFFOLD | WiSE-FT | 1.842±0.056 | 1.177±0.009 | 1.162±0.004 | 2.454±0.043 | 3.50 | 1.544±0.063 | 1.041±0.017 | 1.151±0.007 | 2.301±0.042 | 3.50 | 1.388±0.023 | 0.834±0.012 | 1.114±0.002 | 1.936±0.037 | 3.25 |
| | L²-SP | 1.699±0.049 | 1.086±0.019 | 1.162±0.002 | 2.331±0.024 | 2.50 | 1.473±0.009 | 0.961±0.003 | 1.153±0.002 | 2.201±0.038 | 2.50 | 1.163±0.026 | 0.813±0.010 | 1.126±0.011 | 1.885±0.011 | 2.50 |
| | Top | 1.680±0.042 | 1.036±0.007 | 1.159±0.000 | 2.292±0.026 | 1.25 | 1.436±0.054 | 0.937±0.008 | 1.149±0.003 | 2.187±0.034 | 1.25 | 1.112±0.015 | 0.802±0.003 | 1.114±0.002 | 1.881±0.010 | 1.00 |
| | DWiSE-FT | 1.616±0.047 | 1.110±0.013 | 1.173±0.005 | 2.306±0.030 | 2.50 | 1.485±0.041 | 0.979±0.014 | 1.158±0.009 | 2.149±0.040 | 2.75 | 1.266±0.021 | 0.823±0.010 | 1.121±0.004 | 1.900±0.019 | 3.00 |
| SIZE | WiSE-FT | 2.615±0.072 | 1.391±0.042 | 0.929±0.004 | 2.762±0.053 | 4.00 | 2.216±0.056 | 1.124±0.031 | 0.917±0.004 | 2.543±0.027 | 3.75 | 2.071±0.078 | 0.902±0.016 | 0.912±0.003 | 2.379±0.086 | 3.75 |
| | L²-SP | 2.393±0.068 | 1.306±0.037 | 0.915±0.002 | 2.497±0.019 | 2.50 | 1.731±0.071 | 1.025±0.028 | 0.905±0.002 | 2.424±0.024 | 1.75 | 1.629±0.084 | 0.821±0.011 | 0.904±0.005 | 2.368±0.013 | 2.50 |
| | Top | 2.369±0.075 | 1.297±0.040 | 0.911±0.002 | 2.497±0.019 | 1.50 | 1.731±0.071 | 1.025±0.028 | 0.898±0.003 | 2.424±0.024 | 1.50 | 1.629±0.084 | 0.803±0.006 | 0.895±0.002 | 2.328±0.017 | 1.50 |
| | DWiSE-FT | 1.488±0.101 | 1.113±0.021 | 0.913±0.007 | 2.539±0.023 | 1.75 | 1.469±0.052 | 1.031±0.022 | 0.920±0.006 | 2.390±0.025 | 2.25 | 1.466±0.040 | 0.816±0.022 | 0.915±0.003 | 2.322±0.031 | 2.00 |

Table 17: DWiSE-FT performance on 2 **Regression** datasets (RMSE metrics) and 2 **Classification** datasets (AUC) in the **Fewshot** setting with 50 samples, evaluated across dataset splits (SCAFFOLD, SIZE) given **GRAPHGPS** model. AVG-R denote the average rank. Standard deviations across five replicates are shown in parentheses. We **bold** and underline the best and second-best performances in each scenario.

| Method | Few-Shot 50 (Scaffold Split) | | | | | | Few-Shot 50 (Size Split) | | | | | |
|---|---|---|---|---|---|---|---|---|---|---|---|---|
| | BACE (AUC) | SIDER (AUC) | Avg AUC | ESOL (RMSE) | LIPO (RMSE) | Avg R | BACE (AUC) | SIDER (AUC) | Avg AUC | ESOL (RMSE) | LIPO (RMSE) | Avg R |
| WiSE-FT | 54.67 ± 0.12 | 83.73 ± 0.00 | 69.23 | 1.020 ± 0.045 | 1.259 ± 0.027 | 3 | 53.03 ± 2.64 | 83.77 ± 0.34 | 68.40 | 1.045 ± 0.054 | 1.230 ± 0.015 | 4 |
| L2-SP | 59.52 ± 3.80 | 84.94 ± 0.16 | 72.23 | 0.897 ± 0.058 | 1.196 ± 0.030 | 2 | 61.93 ± 3.45 | 85.12 ± 0.28 | 73.53 | 0.851 ± 0.036 | 1.194 ± 0.015 | 2.5 |
| TOP | 64.90 ± 1.55 | 85.12 ± 0.23 | 75.01 | 0.859 ± 0.065 | 1.196 ± 0.030 | 1.5 | 61.93 ± 3.45 | 86.48 ± 0.70 | 74.21 | 0.844 ± 0.007 | 1.180 ± 0.014 | 1.5 |
| DWiSE-FT | 64.82 ± 1.53 | 85.23 ± 0.02 | 75.03 | 0.859 ± 0.071 | 1.190 ± 0.016 | 1 | 61.46 ± 0.57 | 85.39 ± 0.23 | 73.43 | 0.868 ± 0.041 | 1.167 ± 0.016 | 2 |

Table 18: XGBoost performance on both regression and classification datasets in the Fewshot setting across 3 dataset splits

**Classification tasks**

| #Shots | Split | Dataset | | | |
|---|---|---|---|---|---|
| | | Clintox | BBBP | BACE | HIV |
| 50 | Random | 50.00 | 75.25 | 75.13 | 47.75 |
| | Scaffold | 68.21 | 57.32 | 58.04 | 50.00 |
| | Size | 50.00 | 62.98 | 61.68 | 52.48 |
| 100 | Random | 68.95 | 70.39 | 82.02 | 47.51 |
| | Scaffold | 82.53 | 58.59 | 65.59 | 56.51 |
| | Size | 62.09 | 63.60 | 63.96 | 52.31 |
| 500 | Random | 87.24 | 86.14 | 83.20 | 63.54 |
| | Scaffold | 86.06 | 64.43 | 69.26 | 66.03 |
| | Size | 71.75 | 80.51 | 53.16 | 65.41 |

**Regression tasks**

| #Shots | Split | Dataset | | | |
|---|---|---|---|---|---|
| | | ESOL | LIPO | Malaria | CEP |
| 50 | Random | 2.1118 | 1.3447 | 1.4396 | 2.3080 |
| | Scaffold | 2.3763 | 1.2556 | 1.3096 | 2.6531 |
| | Size | 3.3287 | 1.5481 | 1.2063 | 2.3934 |
| 100 | Random | 2.0708 | 1.2751 | 1.3917 | 2.2813 |
| | Scaffold | 2.1859 | 1.2160 | 1.2721 | 2.2624 |
| | Size | 2.8140 | 1.3235 | 1.2349 | 2.4970 |
| 500 | Random | 1.3626 | 1.0906 | 1.3015 | 1.8142 |
| | Scaffold | 1.9525 | 1.1078 | 1.2221 | 1.8396 |
| | Size | 2.4934 | 1.0358 | 1.1975 | 2.1820 |

Table 19: LoRA Performance under few-shot and non-fewshot settings across classification and regression datasets with pretrained model GraphGPS.

| Scaffold Split | clintox | bbbp | bace | hiv | sider | esol | lipo | malaria | cep |
|---|---|---|---|---|---|---|---|---|---|
| fewshot 50 | 97.77 ± 0.21 | 60.15 ± 3.83 | 57.99 ± 3.48 | 67.95 ± 4.21 | 83.02 ± 0.24 | 0.796 ± 0.032 | 1.232 ± 0.039 | 1.188 ± 0.008 | 2.081 ± 0.082 |
| fewshot 500 | 99.79 ± 0.02 | 100.00 ± 0.00 | 99.99 ± 0.02 | 80.19 ± 1.52 | 92.16 ± 0.25 | 0.354 ± 0.009 | 0.260 ± 0.010 | 0.872 ± 0.006 | 1.569 ± 0.040 |
| non-fewshot | 99.80 ± 0.02 | 99.84 ± 0.06 | 99.54 ± 0.21 | 94.98 ± 0.66 | 90.84 ± 0.11 | 0.375 ± 0.033 | 0.318 ± 0.009 | 0.737 ± 0.015 | 0.632 ± 0.017 |

