# OpenReview forum: "RoFt-Mol: Benchmarking Robust Fine-tuning with Molecular Graph Foundation Models"
_NeurIPS.cc/2025/Datasets_and_Benchmarks_Track — NeurIPS 2025 Datasets and Benchmarks Track spotlight_

### Official Review · Reviewer_1iJG · 2025-06-24

**Rating:** 5
**Confidence:** 3

**Summary:**

In this work, the authors systematically explore various fine-tuning strategies for molecular datasets. They employ several representative backbone models—including GNNs and graph transformers—along with established benchmark datasets from the chemistry domain. The study investigates (1) full fine-tuning, (2) partial fine-tuning, (3) weight-based fine-tuning, and (4) representation-based fine-tuning approaches.

Through this comprehensive analysis, the authors highlight several key findings, such as the importance of task alignment and the effectiveness of supervised fine-tuning.

Building on these insights, they propose a novel fine-tuning strategy, which demonstrates competitive performance compared to existing methods.

**Additional Feedback:**

Please refer to the weakness and dataset&code comments

**Dataset Code Accessibility:**

Partly

**Dataset Code Comments:**

Please make the code provided as the local folder in the GitHub format to make users better leverage the work.

**Ethical Considerations:**

No, there are no or only very minor ethics concerns

**Final Justification:**

The authors have addressed my comments.

**Limitations Weaknesses:**

**[Lack of representative PEFT strategies]** In NLP, parameter-efficient fine-tuning (PEFT) methods such as LoRA [1], Prefix Tuning [2], and IA3 [3] are widely adopted. To my understanding, the authors do not address these approaches in their study. Including an analysis of these strategies would enhance the work.

**[Unclear motivation presentation]** In Lines 32–42, the authors argue that molecular datasets are significantly smaller than language and image datasets, which motivates the need for specialized analysis. While this is a valid point, I believe the core issue is not merely the size of the data, but its 'representativeness.'

For instance, many large language models are trained on vast portions of web data, enabling them to cover a broad and representative spectrum of human language as it appears online and in real life. This means LLMs are exposed to most of the essential text patterns used across the web.

In contrast, the real-world chemical space is far more complex, and the molecular datasets available to us likely capture only a small and potentially unrepresentative subset of that space. The real challenge, therefore, lies in the limited representativeness of available molecular data, rather than just its scale. I believe this perspective should be reflected in the discussion.

**[Weak user friendliness]** Since this is part of the D&B track, I believe the code and datasets should be made publicly available in the GitHub format (instead of the local folder format, which is provided in the Supplementary material) to ensure reproducibility.

**[References]**

- [1] LoRA: Low-Rank Adaptation of Large Language Models
- [2] Prefix-Tuning: Optimizing Continuous Prompts for Generation
- [3] Few-Shot Parameter-Efficient Fine-Tuning is Better and Cheaper than In-Context Learning

**Strengths Contributions:**

**[Timely work]** I think the recent trend in machine learning is using a foundation model and fine-tuning it to perform the downstream task. In light of this, I think this benchmark is a timely work that contributes to the representation learning in the chemistry domain.

**[Comprehensive analysis]** In my view, the experimental design and results are quite comprehensive, covering a wide range of representative strategies. Although some methods are not included, this is understandable given practical constraints.

---

> ### Author Rebuttal · Authors · 2025-07-31
>
> We thank the reviewer for pointing out our categorization of fine-tuning methods, the inclusion of various pre-trained models with different backbones and our key findings. We will respond to the listed limitations as follows.
>
> > 1). Lack of representative PEFT strategies
>
> We thank the reviewer for raising this insightful point regarding the connection to PEFT methods. We clarify below why these techniques were not included in our current benchmark and how our study relates to them:
>
> First, the **primary goal of our benchmark differs from the original motivation of most PEFT methods**. Our focus is on exploring robust fine-tuning strategies for MGFMs particularly under **downstream data scarcity and distribution shift**, a setting where **model generalization** and **resilience to overfitting and forgetting** are critical. To that end, we evaluate fine-tuning strategies that directly address these robustness challenges, categorizing them into weight-based, representation-based, and partial fine-tuning approaches.
>
> In contrast, **PEFT methods are primarily motivated by computational efficiency**, aiming to reduce the number of trainable parameters when adapting very large models, such as LLMs. While these methods may also offer regularization benefits, **our benchmark also includes partial fine-tuning techniques** such as Linear Probing and Surgical-FT, which similarly restrict parameter updates and serve as lightweight PEFT variants within the context of MGFMs.
>
> Second, from a **practical standpoint**, the **scale of MGFMs is modest** compared to LLMs. For example, the large pre-trained model GraphGPS has approximately **14 million parameters**, which can be fine-tuned end-to-end on standard GPUs (e.g., 48GB) without memory constraints. Therefore, the urgency for computationally motivated PEFT approaches is currently lower in this domain.
>
> We will include the additional discussion of PEFT and how our work relates to those methods in the appendix of the revised camera-ready version.
>
> > 2). Unclear motivation presentation
>
> We thank the reviewer for this thoughtful and nuanced observation. We **fully agree** that the **representativeness and diversity of pre-training datasets** are just as critical as their scale, particularly in the molecular domain.
>
> **We observe this effect empirically in our benchmark**. As discussed in Appendix G.1, **GraphGPS**, which is pre-trained on **PCQM4MV2**—a relatively homogeneous dataset focused on quantum properties—shows reduced performance when using Feature-map, a method that enforces strong alignment between pre-trained and fine-tuned representations. In contrast, BSS, another representation-based method that adds regularization over less transferable feature directions, performs better in this setting.
>
> Meanwhile, **Graphium-Large** and **Graphium-Toy**, pre-trained on **LargeMix** and **SmallMix** respectively—both of which include a broader range of bioactivity and physical property datasets—demonstrate stronger generalization when fine-tuned with Feature-map. These results reinforce the reviewer’s point that dataset diversity and task alignment significantly influence the transferability of molecular representations.
>
> We will revise the introduction to explicitly incorporate this important perspective, emphasizing that the core challenge in molecular pre-training lies not only in dataset scale but also in its representativeness.
>
> > 3). Weak user friendly
>
> We thank the reviewer for pointing out the importance of user accessibility and reproducibility. In the camera-ready version, we will release a **well-organized and publicly accessible GitHub repository** containing all code, configuration files, and instructions for reproducing the benchmark and experiments.

---

> > ### Comment · Reviewer_1iJG · 2025-08-01
> >
> > Thank you for your response.
> >
> > Regarding Response 1, while I agree that the original motivation behind LoRA (and related techniques) is to facilitate scalable fine-tuning of large models, I see no practical limitation to applying these techniques to the GNNs and Graph Transformers discussed in the paper.
> >
> > Moreover, to my understanding, LoRA shares structural similarities with the weight-based fine-tuning method already explored by the authors. In this context, the additional parameters introduced could potentially be replaced with a LoRA structure, which (1) is more parameter-efficient and (2) may better preserve pre-trained knowledge by updating fewer parameters.
> >
> > Given that this work serves as a benchmark study evaluating existing fine-tuning techniques, **I believe it would be valuable to include at least a simple empirical analysis** of widely-used methods such as LoRA.

---

> > > ### Author Response · Authors · 2025-08-06
> > >
> > > Thank you for the suggestion. We have now incorporated **LoRA results for GraphGPS** under the scaffold split across three regimes: Fewshot-50, Fewshot-500, and non-fewshot. We will add more results with LoRA in the later camera-ready version.
> > >
> > > Across both classification and regression tasks, LoRA falls short of full fine-tuning in roughly two-thirds of cases, with the gap widening for regression when more samples are available. This pattern is unsurprising since more challenging tasks and larger downstream datasets generally require updating a greater number of parameters. In the instances where LoRA does outperform full-FT, its results typically lie between standard full fine-tuning and the strongest fine-tuning baselines. Notably, under the Fewshot-50 regression setting, LoRA occasionally matches or even exceeds the best benchmarked fine-tuning methods, highlighting its potential in extremely low-data scenarios.
> > >
> > > | scaffold split | clintox       | bbbp          | bace          | hiv           | sider         |
> > > | -------------- | ------------- | ------------- | ------------- | ------------- | ------------- |
> > > | fewshot 50     | 97.77 (0.21)  | 60.15 (3.83)  | 57.99 (3.48)  | 67.95 (4.21)  | 83.02 (0.24)  |
> > > | fewshot 500    | 99.79 (0.02)  | 100 (0.00)    | 99.99 (0.02)  | 80.19 (1.52)  | 92.16 (0.25)  |
> > > | non-fewshot    | 99.80 (0.02)  | 99.84 (0.06)  | 99.54 (0.21)  | 94.98 (0.66)  | 90.84 (0.11)  |
> > >
> > > | scaffold split | esol           | lipo           | malaria        | cep            |
> > > | -------------- | -------------- | -------------- | -------------- | -------------- |
> > > | fewshot 50     | 0.796 (0.032)  | 1.232 (0.039)  | 1.188 (0.008)  | 2.081 (0.082)  |
> > > | fewshot 500    | 0.354 (0.009)  | 0.260 (0.010)  | 0.872 (0.006)  | 1.569 (0.040)  |
> > > | non-fewshot    | 0.375 (0.033)  | 0.318 (0.009)  | 0.737 (0.015)  | 0.632 (0.017)  |

---

> > > > ### Comment · Reviewer_1iJG · 2025-08-07
> > > >
> > > > Thank you for your additional response.
> > > >
> > > > I think the new results are insightful, and therefore, I have increased the score to accept.

---

### Official Review · Reviewer_8grM · 2025-06-30

**Rating:** 5
**Confidence:** 3

**Summary:**

This paper benchmarks the fine-tuning methods for molecular graph foundation models (MGFMs). Due to the data sparsity issue, the fine-tuning of MGFMs plays a crucial role for diverse downstream tasks. This paper explores eight fine-tuning methods and divides them into three categories. Through extensive experiments, this paper draws several important conclusions about fine-tuning MGFMs, which are meaningful for researchers in this field.

**Dataset Code Accessibility:**

Yes

**Dataset Code Comments:**

This paper makes extensive experiments with detailed descriptions.

**Ethical Considerations:**

No, there are no or only very minor ethics concerns

**Final Justification:**

This paper makes a good attempt at fine-tuning the molecular foundation models. The motivation is clear.

This paper also makes extensive experiments, e.g., 17 quantitative tables, to benchmark existing methods.

In the rebuttal, the authors clearly address my concerns and promise to polish their presentations.

Based on the comments from other reviewers, I think this paper is ready to publish. Therefore, I increase my score to accept.

**Limitations Weaknesses:**

1. My major concern about this paper is whether we should use **smaller pre-training datasets** as benchmarks. This paper states that MGFMs are limited by smaller pre-training datasets and more severe data scarcity for downstream tasks. However, there are several large-scale pre-training datasets, e.g., PCQM4MV2. However, this paper does not fully explore the influence of these datasets.

  - GraphGPS surpasses other models by a large margin and achieves nearly 100% classification accuracy in the few-shot setting. This implies that models pre-trained on PCQM4MV2 can generalize to downstream tasks.

  - Tables 12 & 13 show that Feature-map is the most suitable fine-tuning method for GraphGPS, and WiSE-FT does not work well on GraphGPS. In this case, researchers may doubt the effectiveness of DWiSE-FT.

  - This paper only evaluates the performance of DWiSE-FT on the  Mole-BERT model, which is not convincing.

2. The emerging graph prompt learning framework can also be viewed as a parameter-efficient fine-tuning method, which should also be useful for the molecular tasks. It would be better if the authors could discuss the related methods and evaluate some representative works [1, 2].

3. The conclusions in Section 4 are analyzed based on the results from Mole-BERT and Graphium-Toy models. However, similar to weakness 1, both Mole-BERT and Graphium-Toy are pre-trained on small-scale datasets, which may have inconsistent results with GraphGPS. It would be better to analyze the models pre-trained on large-scale datasets.

4. The presentation of this paper should be carefully polished. Here are some suggestions:
  - In the abstract, "informs the design of a refined robust fine-tuning method, ROFT-MOL". In my understanding, ROFT-MOL is the evaluation framework, and DWiSE-FT should be the proposed fine-tuning method. The description in the abstract is confusing.
  - In line 95, "reconstruction methods and contrastive methods. The generative-based strategies". What does the " generative-based strategies" mean? There are only reconstruction and contrastive methods.
  - Some abbreviations are not defined, e.g., CV and NLP.
  - Some symbols are confusing, e.g., Q1 and Q2 in the introduction. Better to add a link.

[1] Towards Graph Prompt Learning: A Survey and Beyond.
[2] ProG: A Graph Prompt Learning Benchmark.

**Strengths Contributions:**

1. Instead of focusing on the pretraining stage of MGFMs, this paper explores the role of fine-tuning, which is overlooked by previous methods but is also important for molecular tasks. The motivation is clear.

2. This paper gives a clear taxonomy of existing fine-tuning methods, including weight-based FT, representation-based FT, and partial FT.

3. This paper makes extensive experiments, e.g., 17 quantitative tables, based on which the authors draw a series of conclusions. All of them are meaningful for molecular tasks.

4. This paper proposes a new fine-tuning method, called DWiSE-FT, which combines the strengths of both post-hoc weight interpolation and weight ensemble, consistently improving the fine-tuning performance on diverse molecular tasks.

---

> ### Author Rebuttal · Authors · 2025-07-31
>
> We thank the reviewer for appreciating the motivation and extensiveness of our work and the proposed DWiSE-FT method. We will address the concerns as below.
>
> > 1). Smaller pre-training datasets as benchmark
>
> We appreciate the reviewer’s important observation. We would like to clarify that our benchmark **does include pre-trained models based on large and diverse datasets**. Specifically, we benchmark GraphGPS, which is pre-trained on **PCQM4MV2**, as well as Graphium-Large, which is trained on **LargeMix**, a dataset containing approximately **~343M graph-level and ~198M node-level** samples. Notably, LargeMix includes the PCQM4M_G25_N4 subset mentioned by the reviewer. When we refer to “smaller pre-training datasets,” we do so in contrast to the scale of language or vision foundation models, which are often trained on billions of examples. Also, we fully agree that pre-training on large datasets like PCQM4MV2 significantly improves downstream generalization, as reflected in the strong performance of GraphGPS.
>
> > 2). Performance of DWiSE-FT on other pre-trained models
>
> We thank the reviewer for pointing this out. We would like to emphasize three clarifications:
>
> - First, the main goal of this work is to **provide systematic benchmarking and insights** into fine-tuning strategies across a range of pre-training paradigms. DWiSE-FT is introduced as a **proof of concept demonstrating the potential of weight-based strategies**, particularly, layer-adaptive interpolation can be extended beyond existing methods like WiSE-FT.
>
> - Second, while WiSE-FT underperforms on GraphGPS, DWiSE-FT is not equivalent to WiSE-FT. As shown in Table 4, DWiSE-FT consistently outperforms L2-SP in self-supervised settings by learning layer-wise adaptive mixing coefficients, thus mitigating limitations of fixed-weight interpolation. Therefore, we expect DWiSE-FT to perform comparably to L2-SP even on GraphGPS due to its adaptive design, despite the underperformance of WiSE-FT. To address this concern, we conducted a **preliminary evaluation of DWiSE-FT on the supervised GraphGPS model**. Below we summarize the Few-shot 50 results with scaffold and size splits. These results show that **DWiSE-FT not only significantly improves over WiSE-FT and L2-SP, but also matches or exceeds the best-performing method in some cases**. This demonstrates that DWiSE-FT remains effective even under supervised pre-training, including on models like GraphGPS where traditional weight-based methods struggle.
>
> ### Few-Shot 50 (Scaffold Split)
>
> | Method     | BACE (AUC)       | SIDER (AUC)      | Avg AUC | ESOL (RMSE)      | LIPO (RMSE)      | Avg Rank |
> |------------|------------------|------------------|---------|------------------|------------------|-----------|
> | WiSE-FT    | 54.67 (±0.12)    | 83.73 (±0.00)    | 69.23  | 1.020 (±0.045)   | 1.259 (±0.027)   | 3         |
> | L2-SP      | 59.52 (±3.80)    | 84.94 (±0.16)    | 72.23   | 0.897 (±0.058)   | 1.196 (±0.030)   | 2         |
> | TOP        | 64.90 (±1.55)    | 85.12 (±0.23)    | 75.01   | 0.859 (±0.065)   | 1.196 (±0.030)   | 1.5       |
> | DWiSE-FT   | 64.82 (±1.53)    | 85.23 (±0.02)    | 75.03  | 0.859 (±0.071)   | 1.190 (±0.016)   | 1         |
>
> ### Few-Shot 50 (Size Split)
>
> | Method     | BACE (AUC)       | SIDER (AUC)      | Avg AUC | ESOL (RMSE)      | LIPO (RMSE)      | Avg Rank |
> |------------|------------------|------------------|---------|------------------|------------------|-----------|
> | WiSE-FT    | 53.03 (±2.64)    | 83.77 (±0.34)    | 68.40   | 1.045 (±0.054)   | 1.230 (±0.015)   | 4         |
> | L2-SP      | 61.93 (±3.45)    | 85.12 (±0.28)    | 73.53  | 0.851 (±0.036)   | 1.194 (±0.015)   | 2.5       |
> | TOP        | 61.93 (±3.45)    | 86.48 (±0.70)    | 74.21  | 0.844 (±0.007)   | 1.180 (±0.014)   | 1.5       |
> | DWiSE-FT   | 61.46 (±0.57)    | 85.39 (±0.23)    | 73.43  | 0.868 (±0.041)   | 1.167 (±0.016)   | 2         |
>
> - Third, DWiSE-FT is appealing not only for its empirical results but also for its **efficiency and practical usability**. As a post-hoc method, it requires no additional training, incurs minimal computational overhead, and is easily applicable across models. Even when its performance is on par with other methods, its **plug-and-play nature** makes it a valuable option in practice.
>
> We will revise the final version to better highlight these clarifications and include broader results for DWiSE-FT on supervised models like GraphGPS.
>
> > 3). Whether the conclusions from pre-trained models on larger datasets are consistent with the ones discussed in section 4
>
> We thank the reviewer for raising this important point. We would like to clarify that the conclusions presented in Section 4 **generalize well to models pre-trained on large-scale datasets**, such as **GraphGPS** (pre-trained on PCQM4MV2) and **Graphium-Large** (trained on the LargeMix dataset containing hundreds of millions of labeled molecular graphs).
> In Section 4.3 and Appendix G.1, we analyze the consistency of trends across all six pre-trained models in our benchmark. Below, we summarize key observations that hold true for models trained on large-scale data:
>
> - **Supervised pre-training on large datasets leads to stronger downstream performance**, particularly in few-shot settings. This aligns with our **main conclusion in Section 4.2 (Q2)**, where we compare supervised and self-supervised pre-training. Models like GraphGPS and Graphium-Large consistently outperform self-supervised models such as Mole-BERT and GraphMAE under the same fine-tuning protocols.
>
> - **Representation-based fine-tuning methods (e.g., Feature-map and BSS)** remain top-performing strategies for supervised pre-trained models on large-scale datasets, consistent with **Finding 3**. This trend holds across both classification and regression tasks, and across different dataset splits.
>
> - **Partial fine-tuning methods (e.g., LP and Surgical-FT)** continue to underperform in few-shot settings. This observation supports **Finding 2**, and reflects their tendency to underfit in data-scarce regimes, even when the underlying pre-trained model is strong.
>
> We will make these observations more explicit in the final version by highlighting relevant comparisons in the main text, along with clearer references to the corresponding sections and appendix materials.
>
> > 4). Discussion of graph prompt learning framework
>
> We thank the reviewer for pointing out the relevance of graph prompt learning as a PEFT strategy. We agree that this is a promising and emerging direction for adapting pre-trained graph models to downstream tasks with minimal parameter updates.
>
> Our work primarily focuses on evaluating **robust fine-tuning strategies** for molecular graph foundation models (MGFMs) under challenging conditions, such as **limited labeled downstream data and distribution shifts**. The **partial fine-tuning techniques** such as Linear Probing and Surgical-FT we benchmarked can be viewed as instances of PEFT, as they update only a subset of model parameters and implicitly act as a form of regularization against overfitting.
>
> While graph prompt learning methods offer an orthogonal and potentially complementary approach to robust fine-tuning, while integrating and benchmarking these methods would require a more substantial methodological adaptation to the molecular graph domain, which we consider an exciting direction for future work.
>
> In the final version, we will include a **discussion on graph prompt learning**, its relevance to molecular tasks, and its connection to PEFT. We will also **cite the representative works suggested** by the reviewer.
>
> > 5). Paper presentation
>
> We greatly appreciate the reviewer’s attention to detail and helpful suggestions regarding the clarity and presentation of the paper. We will carefully polish the final version to improve readability and eliminate the ambiguities.

---

> > ### Comment · Reviewer_8grM · 2025-08-04
> > **Responce to Author Rebuttal**
> >
> > Thanks for the detailed rebuttal and most of my concerns have been addressed. I will increase my score to accept.
> >
> > It would be great if the author could revise the presentation to make this paper clearer and more complete.

---

### Official Review · Reviewer_D7uA · 2025-06-30

**Rating:** 5
**Confidence:** 4

**Summary:**

Motivated by recent advances in Molecular Graph Foundation Models (MGFMs), this work systematically benchmarks fine-tuning techniques for pre-trained MGFMs, evaluating both supervised and self-supervised pre-trained models on a range of downstream classification and regression tasks.

The proposed benchmarking is non-trivial given the unique characteristics of MGFMs. Through comprehensive benchmarking, the authors uncover valuable insights and accordingly propose a refined robust fine-tuning method for MGFMs.

**Additional Feedback:**

1. Tox21 does not have a citation in Table 5.
2. The font size in the tables and figures is too small, making it challenging to read and interpret the results. Rearranging the tables—such as splitting them into several smaller, more focused tables—would greatly improve readability.

**Dataset Code Accessibility:**

Yes

**Dataset Code Comments:**

The code is provided and well-documented.

**Ethical Considerations:**

No, there are no or only very minor ethics concerns

**Final Justification:**

The author's response provides a proper explanation of the usage of existing datasets and agrees to improve the format of the paper.

Considering that my original rating is high, I will retain my rating.

**Limitations Weaknesses:**

1. This work proposes a benchmarking framework for fine-tuning techniques on pre-trained MGFMs, primarily utilizing existing datasets and tasks from prior studies, which are already familiar to the community.
While this may somewhat limit the novelty of the contributions, I consider it a minor issue given that the benchmarking yields valuable and insightful findings.
2. The formatting of the paper could be improved to enhance readability, especially when connecting the findings to the information presented in the two large tables. The tables and figures currently use a font size that is too small, and each table contains an overwhelming amount of information. While these choices are understandable given the page limitations, the final version would benefit from more reader-friendly, findings-driven result tables, with the larger tables moved to the appendix for reference.

**Strengths Contributions:**

1. This work systematically analyzes the characteristics of pre-training and fine-tuning of MGFMs (e.g., limited training and fine-tuning examples, higher demands on generalization), and accordingly categorizes both pre-training and fine-tuning methods.
2. The overall benchmarking framework is thoughtfully designed, and the insights derived from the benchmarking are valuable and informative.
3. The proposed refined fine-tuning method is well-motivated and demonstrates promising empirical performance.

---

> ### Author Rebuttal · Authors · 2025-07-31
>
> We appreciate the reviewer for acknowledging our systematic analysis of pre-training and fine-tuning of MGFMs and the proposed refined DWiSE-FT method. We will respond to the listed comments below.
>
> > 1). Usage of existing datasets:
>
> We appreciate the reviewer for understanding the primary contribution of our work. Indeed, instead of aiming to introduce new benchmark datasets, our main goal is to reveal insightful findings in selecting appropriate robust fine-tuning methods for MGFM given different pre-training mechanisms and under limited downstream data samples.
>
> We thank the reviewer for recognizing the value of our benchmarking efforts. Indeed, the primary objective of our work is not to introduce new datasets, but to **systematically evaluate and distill practical insights** into the selection of robust fine-tuning strategies for MGFMs across diverse pre-training paradigms and downstream task sample availability.
>
> By leveraging widely used datasets from prior studies, we ensure comparability, reproducibility, and relevance of our findings to the broader community. The novelty of our work lies in the **categorization of fine-tuning methods, the unified evaluation under both few-shot and out-of-distribution settings**, and the actionable recommendations derived from extensive empirical analysis.
>
> > 2). Paper formatting:
>
> We thank the reviewer for carefully reviewing our paper and for the constructive feedback on improving readability. In the final version, we will **replace the large tables with more concise, findings-focused tables and plots** that highlight the key trends. The complete numerical results will be moved to the appendix for reference, ensuring both clarity in the main text and accessibility to the full results.

---

> > ### Comment · Reviewer_D7uA · 2025-08-04
> >
> > Thank you for the rebuttal.
> >
> > As stated in my initial review, I am satisfied with the core contributions of this work. With improvements to the clarity and readability of the paper, along with thoughtful consideration of the remaining concerns raised by other reviewers, I believe the paper is well-positioned for acceptance.
> >
> > I do have one minor comment regarding the choice of representative graph Transformer models. Specifically, GraphGPS, while included as a graph Transformer baseline, relies heavily on a message-passing component, which—based on its own ablation study—contributes more to performance than the global attention module. In this sense, it behaves more like a message-passing neural network (MPNN) than a pure Transformer.
> >
> > To more faithfully represent the class of pure graph Transformer architectures, the authors might consider including alternatives such as Graphormer (Ying et al., 2021), GD-Graphormer (Zhang et al., 2023), or GRIT (Ma et al., 2023), which do not incorporate MPNNs.
> >
> > That said, this is a relatively minor, architecture-driven point and does not detract from the overall contribution of the paper.
> >
> > I will maintain my current score, which already reflects a recommendation for acceptance.
> >
> >
> > ----
> > - Ying et al., "Do Transformers Really Perform Bad for Graph Representation?", NeurIPS 2021
> > - Zhang et al., "Rethinking the Expressive Power of GNNs via Graph Biconnectivity", ICLR 2023
> > - Ma et al., "Graph Inductive Biases in Transformers without Message Passing", ICML 2023

---

### Official Review · Reviewer_Z9wV · 2025-07-02

**Rating:** 5
**Confidence:** 3

**Summary:**

The submission introduces ROFT‑MOL, a benchmark aimed at the systematic evaluation of robust fine‑tuning techniques for molecular graph foundation models (MGFMs). The authors use eight representative fine‑tuning (FT) methods that are grouped into three mechanisms: weight‑based, representation‑based, and partial FT. The methods are tested across six publicly available MGFMs (both self‑supervised and supervised), twelve downstream molecular property datasets (eight classification, four regression), three data‑split regimes (random, scaffold, size), and both non‑few‑shot and few‑shot settings.

The paper, based on the results, suggests design rules summarising when each FT mechanism works best and, motivated by those findings, proposes DWiSE‑FT, a lightweight layer‑wise weight‑interpolation strategy that marries WiSE‑FT and L2‑SP. Extensive tables (e.g., Tables 2–3) and rank‑plots (Figure 2) substantiate three core insights: (i) supervised pre‑training helps most in few‑shot scenarios; (ii) regression tasks are less prone to over‑fitting than classification tasks; and (iii) weight‑based FT dominates for self‑supervised checkpoints, whereas representation‑regularised FT dominates for supervised ones.

Lastly, and importantly for the DB track, the code, configuration files, and data loaders are released to reproduce every experiment.

**Dataset Code Accessibility:**

Yes

**Dataset Code Comments:**

The authors provide their source codes, and they seem reproducible, but I have not tried to run them myself.

**Ethical Considerations:**

No, there are no or only very minor ethics concerns

**Final Justification:**

The authors have addressed all my comments, so I keep my original positive rating of the paper.

**Limitations Weaknesses:**

* To my understanding, all downstream tasks originate from MoleculeNet or similar small‑molecule datasets. Results may not generalize to macromolecules or reaction graphs.

* Only property‑prediction tasks are studied. Generative objectives or reaction‑outcome prediction remain unexplored.

* To the best of my understanding, DWiSE‑FT is evaluated only on self‑supervised checkpoints in few‑shot regression/classification. A head‑to‑head against Feature‑Map or Surgical‑FT on supervised checkpoints would strengthen the claim of universality.

* Proposition 1 is proven in the appendix but its practical implication (why alpha becomes layer‑specific in practice) could be expanded in the main text for better clarity.

**Strengths Contributions:**

* The paper offers rigorously designed test‑bed datasets and benchmarks that reflect real‑world constraints such as extreme label scarcity and out‑of‑distribution evaluation.

* The authors perform an analysis of the results. That is, the paper goes beyond raw numbers, articulating high‑level design rules (Section 4) and supporting them with concrete examples (e.g., classification vs. regression performance disparities in Figure 2)

* Appendix F details hyper‑parameters, compute budgets, and data splits; the supplementary checklist affirms open‑sourcing of code and data.

* The manuscript is clearly written. It is very information heavy so it requires effort to read it, but given that this is a rather unestablished field, I am fine with that.

---

> ### Author Rebuttal · Authors · 2025-07-31
>
> We thank the reviewer for recognizing the design and insights of our benchmark and for providing valuable suggestions to enhance the paper. Below, we address the noted weaknesses:
>
> > 1). Limited studies domain and downstream tasks (Weakness 1 & 2):
>
> We thank the reviewer for the insightful suggestion. Indeed, extending the benchmark to include macromolecules, reaction graphs, and generative or reaction-outcome prediction tasks is a valuable direction. Our current study focuses on small-molecule property prediction, a setting that is widely studied and where pre-trained molecular graph models are most mature. It also presents well-established challenges such as data scarcity and distribution shift, which are central to our goal of evaluating robust fine-tuning strategies. We agree that expanding to more complex molecular domains is an exciting direction for future work.
>
> > 2). DWISE-FT only evaluated on self-supervised checkpoints.
>
> We thank the reviewer for the thoughtful suggestion to evaluate DWiSE-FT more broadly across different types of pre-trained models. We respond to this concern from the following perspectives:
>
> First, we would like to clarify that the primary focus of this paper is on benchmarking and providing actionable insights into selecting robust fine-tuning strategies for downstream tasks, rather than proposing novel methodological contributions. Therefore, DWiSE-FT serves as a proof of concept **demonstrating the potential for enhancing the benefit from weight-based methods** under self-supervised settings.
>
> Second, we acknowledge the importance of evaluating DWiSE-FT on supervised pre-trained models. To this end, we have conducted a **preliminary study on the GraphGPS** model, which is trained under a supervised setting. Below we summarize the Few-shot 50 results with scaffold and size splits. These results show that **DWiSE-FT not only significantly improves over WiSE-FT and L2-SP, but also matches or exceeds the best-performing method in some cases**. This demonstrates that DWiSE-FT remains effective even under supervised pre-training, including on models like GraphGPS where traditional weight-based methods struggle.
>
> ### Few-Shot 50 (Scaffold Split)
>
> | Method     | BACE (AUC)       | SIDER (AUC)      | Avg AUC | ESOL (RMSE)      | LIPO (RMSE)      | Avg Rank |
> |------------|------------------|------------------|---------|------------------|------------------|-----------|
> | WiSE-FT    | 54.67 (±0.12)    | 83.73 (±0.00)    | 69.23  | 1.020 (±0.045)   | 1.259 (±0.027)   | 3         |
> | L2-SP      | 59.52 (±3.80)    | 84.94 (±0.16)    | 72.23   | 0.897 (±0.058)   | 1.196 (±0.030)   | 2         |
> | TOP        | 64.90 (±1.55)    | 85.12 (±0.23)    | 75.01   | 0.859 (±0.065)   | 1.196 (±0.030)   | 1.5       |
> | DWiSE-FT   | 64.82 (±1.53)    | 85.23 (±0.02)    | 75.03  | 0.859 (±0.071)   | 1.190 (±0.016)   | 1         |
>
> ### Few-Shot 50 (Size Split)
>
> | Method     | BACE (AUC)       | SIDER (AUC)      | Avg AUC | ESOL (RMSE)      | LIPO (RMSE)      | Avg Rank |
> |------------|------------------|------------------|---------|------------------|------------------|-----------|
> | WiSE-FT    | 53.03 (±2.64)    | 83.77 (±0.34)    | 68.40   | 1.045 (±0.054)   | 1.230 (±0.015)   | 4         |
> | L2-SP      | 61.93 (±3.45)    | 85.12 (±0.28)    | 73.53  | 0.851 (±0.036)   | 1.194 (±0.015)   | 2.5       |
> | TOP        | 61.93 (±3.45)    | 86.48 (±0.70)    | 74.21  | 0.844 (±0.007)   | 1.180 (±0.014)   | 1.5       |
> | DWiSE-FT   | 61.46 (±0.57)    | 85.39 (±0.23)    | 73.43  | 0.868 (±0.041)   | 1.167 (±0.016)   | 2         |
>
> Lastly, DWISE-FT is not only compelling due to its empirical performance, but also because of its **efficiency and plug-and-play nature**. As a post-hoc interpolation method, it introduces minimal overhead while offering meaningful gains, making it both practical and broadly applicable. Consequently, it still offers some practical computational benefit if it has comparable performance to representation-based fine-tuning under supervised pre-trained models.
>
> We will clarify these points and include the preliminary results in the revision to provide a more balanced evaluation.
>
> > 3). Clarification of proposition 1
>
> We thank the reviewer for highlighting the need to better connect Proposition 1 to its practical implications.
>
> Proposition 1 demonstrates that the optimal solution of the L2-SP objective corresponds to a **direction-wise interpolation** between the pre-trained weights $\theta_{\text{pre}}$ and the fine-tuned weights $\theta_{\text{ft}}$, modulated by the curvature (eigenvalues) of the loss landscape. Specifically, it shows that in **high-curvature directions**, the solution favors $\theta_{\text{ft}}$, while in **flatter directions**, it leans more towards $\theta_{\text{pre}}$. This insight contrasts with the global interpolation of WiSE-FT and reveals that L2-SP inherently performs **adaptive interpolation**.
>
> In practice, since **different layers are typically associated with parameters of varying curvature and semantic roles**, they should be interpolated differently. For example, deeper layers, which capture task-specific information, often exhibit sharper curvature and thus should remain closer to the fine-tuned weights. In contrast, earlier layers, which encode more general-purpose features, benefit from retaining more of the pre-trained initialization. This motivates our **layer-wise extension of interpolation coefficients ($\alpha_i$)** in DWiSE-FT.
>
> We will incorporate this explanation into the main text to better connect the theoretical motivation with the practical design of our method.

---

> > ### Comment · Reviewer_Z9wV · 2025-08-04
> >
> > I thank the authors for the clarifications and added experiments, which further validate the merit of the work. I keep my score and recommend acceptance.

---

### Decision · Program_Chairs · 2025-09-18

**Decision:**

Accept (spotlight)

**Comment:**

Wide consensus on acceptance. Agree.

===== FINAL UPDATE FROM DB Track PCs ====

The final decision for this paper has been taken by the program chairs after consultation with the SACs. All Senior Area Chairs have ranked papers according to the feedback from the AC during the review process. We decided to leave the original meta-review to reflect the opinion of the AC in light of the initial discussions with reviewers and SAC.